# EFFICIENT SYNTHETIC NETWORK GENERATION VIA LATENT EMBEDDING RECONSTRUCTION

## ABSTRACT

Network data are ubiquitous across the social sciences, biology, and information systems. Generating realistic synthetic network data has broad applications from network simulation to scientific discovery. However, many existing black-box approaches for network generation tend to overfit observed data while overlooking characteristic network structure, and incur substantial computational overhead at scale. These practical challenges call for synthetic network generation methods that are both efficient and capable of capturing structural properties of networks. In this paper, we introduce Synthetic Network Generation via Latent Embedding Reconstruction (SyNGLER), a general and efficient framework for synthetic network generation that builds on latent space network models. Given an observed network, SyNGLER first learns low-dimensional latent node embeddings via a latent space network model and then reconstructs the latent space by building a distribution-free generator over these embeddings. For generation, SyNGLER first samples (or resamples) node embeddings from the generator in the latent space and then produces synthetic networks using the latent space network model. Through the latent space framework, SyNGLER preserves unique characteristics in networks such as sparsity and node degree heterogeneity, while allowing for efficient training with lower computational cost than many existing deep architectures. We provide theoretical guarantees by developing consistency results regarding the distance between the true and synthetic edge distributions. Empirical studies further demonstrate the effectiveness of SyNGLER, where SyNGLER efficiently produces networks that better preserve key network characteristics such as network moments and degree distributions compared with existing approaches.

## 1 INTRODUCTION

Graph network data capture interactions among entities in complex systems. Examples include social networks (Traud et al., 2012), molecular interaction networks (Gómez-Bombarelli et al., 2018), and brain connectivity networks (Bullmore & Sporns, 2009). Generating realistic synthetic network data (Zhu et al., 2022) has broad applications, spanning drug discovery (Li et al., 2018a), material discovery (Merchant et al., 2023), and image recognition (Xie et al., 2019). Designing efficient generative models that produce realistic network data while preserving characteristic structural network properties remains a long-standing and active research challenge.

Recent years have witnessed a growing line of work on data-driven graph network generation using deep learning. For example, Li et al. (2018b) proposed an autoregressive generation scheme, in which a graph neural network (GNN; Scarselli et al. (2008)) sequentially adds nodes and edges based on the current graph. You et al. (2018) later adopted recurrent neural networks (Schmidt, 2019) that summarize nodes and edges and generate, at each step, the next node and its associated edges. Liao et al. (2019) introduced a block-wise autoregressive model with graph attention mechanism (Veličković et al., 2017), reducing serial computation while preserving long-range dependencies. Nevertheless, for large graphs, training and sampling in deep autoregressive models remain computationally heavy due to sequential modeling of a large graph (Salha et al., 2021). Another line of research develops diffusion models (Sohl-Dickstein et al., 2015; Ho et al., 2020; Song et al., 2020) for graphs. Early methods (Niu et al., 2020; Jo et al., 2022) applied continuous diffusion processes directly in adjacency-matrix space, which neglected discreteness in graphs. Vignac et al. (2022) and Haefeli et al. (2022) studied discrete Markov processes over adjacency matrices. However, while operating

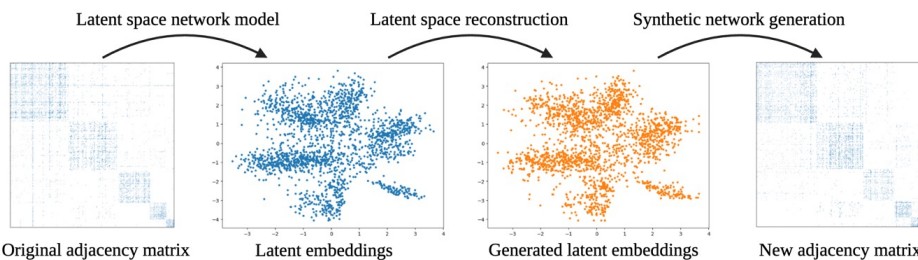

Figure 1: An illustrative SyNGLER pipeline using the YOUTUBE dataset (Yang & Leskovec, 2012) with a two-dimensional latent space. From left to right: observed network in the form of an adjacency matrix; learned latent embeddings; synthetic embeddings from the generator in the latent space; synthetic network.

on a discrete state space, applying diffusion directly in adjacency-matrix space still overlooks the low-rank structure often present in large-scale network data (Luo et al., 2023). Vahdat et al. (2021) and Rombach et al. (2022) combined diffusion modeling with encoder-decoder architectures by applying diffusion in a continuous latent space. The resulting latent-diffusion approach has been used for molecular and protein graph generation (Xu et al., 2023; Fu et al., 2024) and extended to general graph generation (Zhou et al., 2024), covering both conditional and unconditional settings. Nevertheless, most of the methods typically rely on variational training to connect the graphs and the latent space, which becomes computationally demanding for large-scale networks. Overall, existing methods tend to face computational challenges on large graphs due to deep neural network training on large-dimensional data and/or variational procedures, while characteristic network structure is often neglected. A recent work Li et al. (2023) uses a message-passing neural network (MPNN) as the encoder and is able to efficiently generate networks from a single large observation, but a theoretical analysis of the algorithm is lacking. These limitations underscore the need for graph generation models that can capture complex network structure while staying computationally tractable and scalable to large graphs, with accompanying theoretical guarantees.

In this work, we introduce Synthetic Network Generation via Latent Embedding Reconstruction (SyNGLER), an efficient synthetic network generation framework that leverages latent space network models (Hoff et al., 2002; Ma et al., 2020) to address these challenges. During training, SyNGLER first fits a likelihood-based latent space model to the observed network to learn a set of low-dimensional node embeddings. Using these embeddings, it then trains a distribution-free generator in the latent space. For generation, SyNGLER samples (or resamples) node embeddings from the generator and produces synthetic networks from these node embeddings via the latent space model. An illustrative pipeline is in Figure 1. Via the latent space approach, SyNGLER avoids training deep models directly on the high-dimensional network space by learning low-dimensional node embeddings with flexibly chosen likelihood models and requiring only lightweight generative model training in the latent space, thereby reducing computational cost. Moreover, the geometry of in the latent space enables SyNGLER to preserve key structural properties of the network that reflect latent node-node interactions. We provide a theoretical analysis of SyNGLER and establish its consistency and generalization guarantees under a hierarchical latent space network model. Extensive experiments on synthetic and real-world datasets further demonstrate its strong performance in comparison to existing approaches, with significantly reduced computational cost.

The remainder of the paper is organized as follows. In Section 2, we formally introduce the SyNGLER framework. Section 3 presents the theoretical results. Section 4 reports empirical results on simulated and real-world data. Section 5 concludes the paper with a discussion. Additional numerical results, experimental details, and proofs are provided in the Appendix.

## 2 SYNTHETIC NETWORK GENERATION VIA LATENT EMBEDDING RECONSTRUCTION

Given an observed network with $n$ nodes, our goal is to train a generative model that can produce networks that preserves key structural properties of the original network. The synthetic node set may

comprise the original nodes, newly generated nodes with distributional characteristics similar to the originals, or any mixture of the two. Specifically, we represent the observed network by an adjacency matrix $A \in \mathbb{R}^{n \times n}$ with $A_{ij} = A_{ji}$ for $i \neq j$ and $A_{ii} = 0$ for all $i \in [n]$, where the observation $A_{ij}$ can be binary observations in $\{0, 1\}$, count-valued observations in $\mathbb{N}$, or general continuous observations in $\mathbb{R}$. Given the observation $A$, our goal is to generate a synthetic network $\widetilde{A} \in \mathbb{R}^{n \times n}$.

We introduce Synthetic Network Generation via Latent Embedding Reconstruction (SyNGLER) to achieve this goal. During training, SyNGLER first learns a set of node embeddings from the observed network using a likelihood model $p(\cdot \mid \cdot)$ compatible with edge types of $A_{ij}$, and then reconstructs the latent space by training a distribution-free generator over the learned embeddings. For data generation, SyNGLER samples a set of node embeddings from the latent generator and produces synthetic networks using these embeddings via $p(\cdot \mid \cdot)$. Algorithm 1 summarizes the procedure.

---

**Algorithm 1** Synthetic Network Generation via Latent Embedding Reconstruction

---

1: **Input:** Latent dimension $r$, input network $A \in \{0, 1\}^{n \times n}$.
2: Fit the likelihood model to get $(\hat{Z}, \hat{\alpha}) \in \mathbb{R}^{n \times r} \times \mathbb{R}^n$;
3: Train a generative model using the fitted data: `Sampler = GenModel`$(\{(\hat{z}_i, \hat{\alpha}_i)\}_{i=1}^n)$;
4: **for** each $i = 1, \ldots, n$ **do**
5:      Sample $(\tilde{z}_i, \tilde{\alpha}_i) \in \mathbb{R}^{r+1}$ from `Sampler`.
6: **end for**
7: **for** each pair of nodes $(i, j)$ with $1 \leq i < j \leq n$ **do**
8:      Independently generate the edge observation $\widetilde{A}_{ij} = \widetilde{A}_{ji}$ from the conditional model $p(\cdot \mid \tilde{z}_i^\top \tilde{z}_j + \tilde{\alpha}_i + \tilde{\alpha}_j)$
9: **end for**
10: **Output:** Generated network $\widetilde{A}$.

---

In Algorithm 1, $\hat{Z} = (\hat{z}_1, \ldots, \hat{z}_n)^\top$ denotes the learned latent node embeddings and $\hat{\alpha} = (\hat{\alpha}_1, \ldots, \hat{\alpha}_n)^\top$ the learned node degree parameters. `GenModel` denotes the generator architecture in latent space. We specify the likelihood $p(\cdot \mid \cdot)$ and formally introduce the model parameters in Section 2.1, and we detail `GenModel` in Section 2.2.

## 2.1 LATENT SPACE NETWORK MODELS AND NETWORK EMBEDDING

Latent space network models (Hoff et al., 2002; Ma et al., 2020; Li et al., 2025) provide a flexible and efficient network embedding framework. Associate each node $i$ with a latent position $z_i \in \mathbb{R}^r$ and a degree parameter $\alpha_i \in \mathbb{R}$. Let $\alpha = (\alpha_1, \alpha_2, \ldots, \alpha_n) \in \mathbb{R}^n$ and define $\Phi \in \mathbb{R}^{n \times (r+1)}$, where each row of $\Phi$ is given by $\Phi_i = (z_i^\top, \alpha_i)^\top \in \mathbb{R}^{r+1}$, $i = 1, \ldots, n$. Latent space network models specify that given $\Phi$, each edge observation $A_{ij} = A_{ji}$ for $1 \leq i < j \leq n$ is independently generated from $\mathbb{P}(A_{ij} \in \mathcal{A} \mid \Phi) = \int_{\mathcal{A}} p(a \mid z_i^\top z_j + \alpha_i + \alpha_j) \, d\mu(a)$, where $p(\cdot \mid \cdot)$ is the conditional density of $A_{ij}$ given $\Phi$, $\mu$ is some measure on $\mathbb{R}$ (e.g., counting measures for discrete-valued edges or Lebesgue measures for continuous-valued edges), and $\mathcal{A}$ is the corresponding Borel set. Given $\Phi$, we use $\mathbb{P}_{A|\Phi}$ to denote the conditional distribution of $A$ given $\Phi$.

**Assumption 2.1** (embedding distribution). $\{(z_i, \alpha_i)\}_{i=1}^n$ *are independently generated from some distribution* $\mathbb{P}_0$ *on* $\mathbb{R}^{r+1}$ *such that* $\mathbb{E}_{\mathbb{P}_0}[z_i] = 0_r$. *Moreover, there exists* $R > 0$ *such that* $\|z\|_2 \leq R$ *for all* $(z, \alpha) \in \text{supp}(\mathbb{P}_0)$.

Assumption 2.1 guarantees the boundedness of the embedding space and the identifiability of the node degree distribution, since joint shifts in $\alpha$ and $z$ can leave the edge distribution unchanged. Specifically, in (2.1), for all $i \in [n]$, let $\check{z}_i = z_i + a$ and $\check{\alpha}_i = \alpha_i - a^\top z_i - a^\top a / 2$ for some $a \in \mathbb{R}^r$. Then $\check{z}_i^\top \check{z}_j + \check{\alpha}_i + \check{\alpha}_j = z_i^\top z_j + \alpha_i + \alpha_j$ for all $i, j \in [n]$, and the likelihood remains unchanged. Assumption 2.1 thus fixes this shift as $a = -\mathbb{E}_{\mathbb{P}_0}[z_i]$ to ensure node-degree identifiability. Similar conditions have been commonly considered in the latent space literature for node degree identifiability, for example, in Ma et al. (2020) and Zhang et al. (2022).

**Network embedding for general edge types.** Based on the observed edge type of $A_{ij}$, the likelihood model $p(\cdot|\cdot)$ can be chosen flexibly. Define $\pi_{ij} = z_i^\top z_j + \alpha_i + \alpha_j$, which determines the distribution of $A_{ij}$. If $A_{ij}$'s are binary, one can choose the Bernoulli model as further introduced

in the next paragraph. If $A_{ij}$'s are continuously measured, one can use a Gaussian noise model $p(a_{ij} \mid \pi_{ij}) = (2\pi\sigma^2)^{-1/2} \exp\{-(a_{ij} - \pi_{ij})^2/(2\sigma^2)\}$ for some $\sigma^2 > 0$ (Sun & Li, 2017; Wang & Guo, 2023). Once the conditional model is determined, degree parameters and latent embeddings can be estimated by maximizing the following likelihood function over the parameters $(Z, \alpha)$, where $Z = (z_1^\top, z_2^\top, \ldots, z_n^\top)^\top$ is the latent position matrix with rows $z_1, z_2, \ldots, z_n$ and $\alpha = (\alpha_1, \ldots, \alpha_n)$ is the vector of degree parameters:

$$(\hat{Z}, \hat{\alpha}) = \underset{(Z,\alpha) \in \mathbb{R}^{n \times (r+1)}: Z^\top \mathbf{1}_n = \mathbf{0}_r}{\arg\max} \sum_{1 \le i < j \le n} \log p(A_{ij} \mid z_i^\top z_j + \alpha_i + \alpha_j). \tag{1}$$

The constraint in Eq. (1) ensures identifiability of the degree parameters and is designed based on Assumption 2.1. A projected gradient descent algorithm can be employed to efficiently solve Eq. (1) with provable convergence guarantees (Ma et al., 2020). We leave the details of the optimization algorithm to Appendix C.1.

**Sparse networks for binary edges.** For binary networks, let $\mu = \delta_0 + \delta_1$, where $\delta_0$ and $\delta_1$ are Dirac measures at 0 and 1, and specify $p(a_{ij} \mid \pi_{ij}) = \exp\{a_{ij}\pi_{ij}\}(1 + \exp\{\pi_{ij}\})^{-1}$, i.e., $\mathbb{P}(A_{ij} = 1 \mid \pi_{ij}) = \sigma(\pi_{ij})$, where $\sigma(\cdot) = \exp(\cdot)/(1 + \exp(\cdot))$. Recall that $\pi_{ij} = z_i^\top z_j + \alpha_i + \alpha_j$ with $\mathbb{E}_{\mathbb{P}_0}[z_i] = 0$. We further define $\rho_n = 2\mathbb{E}_{\mathbb{P}_0}[\alpha_i]$ and let $w_n = \exp(\rho_n)$. Then we can reparameterize $\pi_{ij} = z_i^\top z_j + \alpha_i + \alpha_j + \rho_n$, where the $\alpha_i$ have been centered by their expectation $\mathbb{E}_{\mathbb{P}_0}[\alpha_i]$. Accordingly, Eq. (1) is modified by introducing an additional parameter $\rho$ and an additional constraint $\alpha^\top \mathbf{1}_n = 0$, which we empirically find to facilitate the embedding process. Under the logistic link, note that $\sigma(\pi_{ij}) \asymp w_n \to 0$ as $\rho_n \to -\infty$, assuming the variance of $\alpha_i$ is bounded. The quantity $w_n$ thus serves to quantify the global edge sparsity, which is commonly observed in large networks. In our theoretical analysis in Section 3, we provide a more detailed discussion of sparsity.

**Remark 1.** *The latent space network model also covers a wide class of classical network reconstruction models given appropriate choices of link functions and parametrizations of the node embeddings, including the Erdős–Rényi graph (Erdos & Rényi, 1960), the Chung–Lu graph (Chung & Lu, 2002), and the stochastic block model and its mixed-membership variants (Holland et al., 1983; Karrer & Newman, 2011; Airoldi et al., 2008). Further discussion is provided in Appendix B.*

## 2.2 LATENT EMBEDDING RECONSTRUCTION

In this section, we introduce two implementations of GenModel in Algorithm 1. Both implementations reconstruct the latent space from the learned embeddings. The first approach resamples from the empirical distribution on $\hat{\Phi}$, which is suitable when the synthetic network should include some original nodes from the observed network. The second approach trains a score-based generator using the learned embeddings and is appropriate when new nodes need to be generated in the network. The final GenModel can be a mixture of the two, producing networks that include both original nodes and new nodes. This choice is flexible and can be adapted to practitioners' needs.

**Resampling based latent space reconstruction.** Given the learned embeddings $\hat{\Phi}$, we set GenModel to be the uniform distribution over the discrete set $\hat{\Phi}$. At each call, this sampler returns one of the row vectors of $\hat{\Phi}$, with replacement. The resampled latent embeddings, denoted $\tilde{\Phi}$, are used later to construct the synthetic network. The idea of resampling latent embeddings in networks has been used for bootstrap inference of network statistics (see Levin & Levina, 2019), but it has not been systematically studied for network generation tasks. In scenarios where duplicate embeddings exist in $\tilde{\Phi}$ due to sampling with replacement, we suggest removing the duplicate embeddings. Meanwhile, any nodes that must remain in the network can have their embeddings preserved in $\tilde{\Phi}$ as needed.

**Score-based generator in the latent space.** To sample novel node embeddings that are close in distribution to the learned embeddings, we consider a score-based generative model formulated via stochastic differential equations (Song et al., 2020), in which a forward noising process gradually adds noise to the training samples and a backward denoising process recovers the distribution of the original samples from pure noise using information from the forward process. In our setup, a key difference from existing work is that we train the score network based on the learned embeddings. Specifically, let $s_\theta(x, t): \mathbb{R}^d \times [0, 1] \to \mathbb{R}^d$ be a prediction model parameterized by $\theta$. We use both

the XGBoost, following ForestDiffusion (Jolicoeur-Martineau et al., 2024) and multilayer perceptrons (MLP), to approximate the score function. Consider a variance-preserving Ornstein-Uhlenbeck (OU) (Maller et al., 2009) forward process. Given the fitted embeddings $\hat{\Phi}$, the forward process $\phi_t$ follows $d\phi_t = -\phi_t \, dt + \sqrt{2} \, dB_t$, where $\phi_0$ is randomly sampled from the row vectors of $\hat{\Phi}$ and $B_t$ is a standard Wiener process. Given the forward process, the parameter $\theta$ in $s_\theta(x, t)$ is optimized by minimizing the denoising score-matching objective (Vincent, 2011) constructed using the learned embeddings (Wu et al., 2025):

$$\hat{\theta} = \arg\min_{\theta} \mathbb{E}_{t \sim \mathcal{U}[0,1], \, z \sim \mathcal{N}(0, I_{r+1})} \left[ \frac{1}{n} \sum_{i=1}^{n} \left\| s_\theta \big( e^{-t} \hat{\phi}_i + \sqrt{1 - e^{-2t}} \, z, \, t \big) + \frac{z}{\sqrt{1 - e^{-2t}}} \right\|^2 \right].$$

To sample from the generator, we simulate the following process initialized at $\tilde{\phi}^{(0)} \sim \mathcal{N}(0, I_{r+1})$, using the trained network $s_{\hat{\theta}}$:

$$\tilde{\phi}^{(k+1)} = \tilde{\phi}^{(k)} + h \left( \tilde{\phi}^{(k)} + 2 \, s_{\hat{\theta}}(\tilde{\phi}^{(k)}, 1 - kh) \right) + \sqrt{2h} \, \xi_k, \quad \xi_k \overset{\text{i.i.d.}}{\sim} \mathcal{N}(0, I_{r+1}).$$

When $T$ is large (equivalently, $h$ is small) and $s_\theta$ sufficiently approximates the true score, the distribution of $\tilde{\phi}^{(\lceil 1/h \rceil)}$ is close to the distribution of the learned embeddings. This approach enables us to sample novel node embeddings, thereby including novel nodes in the synthetic network.

## 2.3 SYNGLER-ATTR: SYNTHETIC NETWORK GENERATION VIA LATENT EMBEDDING RECONSTRUCTION FOR ATTRIBUTED NETWORK

In the main body of this work, we study the task of generating networks when only network structures are observed and needed in downstream tasks, using the vanilla SyNGLER. Meanwhile, generating attributed networks that preserve feature-network interactions is also an important problem. The classical VGAE (Kipf & Welling, 2016) approaches this problem in the setting of fixed attributes, where the attributes of the generated graph remain the same as those of the input graph. Using continuous diffusion models, Jo et al. (2022) developed a pipeline that successfully generates attributed graphs with new features, although its design and training are tailored to multiple small-scale graphs. Li et al. (2023) uses a message-passing neural network (MPNN) as the encoder and is able to efficiently generate attributed graphs from a single large observation. In our paper, to address this task, we introduce SyNGLER-Attr, a generalization of SyNGLER for generating attributed networks. Specifically, SyNGLER-Attr jointly embeds and reconstructs the latent embeddings of the network and its associated attributes, allowing the synthetic data to preserve the joint latent structures of the network and its attributes. Due to page limits, we present the detailed algorithmic description and implementation, along with an empirical study of SyNGLER-Attr to assess its utility in downstream machine learning tasks in Appendix E.

## 3 THEORETICAL ANALYSIS

In this section, we study generation consistency of SyNGLER via the distance between the distributions of the synthetic and original networks. We consider the logistic link model for sparse binary networks introduced in Section 2.1 with logsitic link . Throughout this section, we treat the global sparsity parameter $\rho_n^* = \mathbb{E}_{\mathbb{P}_0}[\alpha_1]/2$ as a fixed parameter, where a superscript $*$ denotes the true parameters, and we assume that the observed network $A$ is generated from the latent space model in Section 2.1 with global sparsity parameter $\rho_n^*$ and some $\Phi^*$ whose rows are independent realizations from $\mathbb{P}_0$ in Assumption 2.1. In this section, we study generation consistency of SyNGLER via the distance between the distributions of the synthetic and original networks. We consider the model for sparse binary networks introduced in Section 2.1 with logistic link. Throughout this section, we treat the global sparsity parameter $\rho_n^* = \mathbb{E}_{\mathbb{P}_0}[\alpha_1]/2$ as a fixed parameter, where a superscript $*$ denotes the true parameters, and we assume that the observed network $A$ is generated from the latent space model in Section 2.1 with global sparsity parameter $\rho_n^*$ and some $\Phi^*$ whose rows are independent realizations from $\mathbb{P}_0$ in Assumption 2.1. Note that the distribution of the synthetic network $\tilde{A}$ is based on the model trained on the observed network $A$. We denote the distribution of $\tilde{A}$ given $A$ as $\mathbb{P}_{\tilde{A}}^A$, which can be viewed as a random measure where the randomness comes from the observed network $A$ generated under model Section 2.1 with embeddings $\Phi^*$ and sparsity parameter $\rho_n^*$. Similarly, we

define $\mathbb{P}_{\tilde{\Phi}}^A$ as the random measure for the group of latent embeddings on $\mathbb{R}^{n \times (r+1)}$. We also denote the individual distribution of $\tilde{\phi}$ given $A$ as $\mathbb{P}_{\tilde{\phi}}^A$. Similarly, we denote the marginal distribution of each $\hat{\phi}_i$ as $\mathbb{P}_{\hat{\phi}}$, where the subscript $i$ in $\tilde{\phi}_i$ is omitted due to the exchangeability of the nodes. Our first theorem decomposes the Kullback-Leibler (KL) divergence between $\mathbb{P}_A$ and $\mathbb{P}_{\tilde{A}}^A$.

**Theorem 3.1.** *Under Assumption 2.1, the average KL divergence between the distribution of $A$ and $\tilde{A}$ given $A$ admits the following decomposition:*

$$\frac{1}{n^2} d_{\mathrm{KL}}(\mathbb{P}_A \parallel \mathbb{P}_{\tilde{A}}^A) = E_\rho + E_\Phi + E_{gen}. \tag{2}$$

*where the three error terms are defined as follows:*

$$E_\rho = \mathbb{E}_{\mathbb{P}_{A|\Phi} \times \mathbb{P}_0^{\otimes n}} \Big[ \log \frac{p(A_{12} \mid z_1^\top z_2 + \alpha_1 + \alpha_2 + \rho_n^*)}{p(A_{12} \mid z_1^\top z_2 + \alpha_1 + \alpha_2 + \hat{\rho})} \Big];$$

$$E_\Phi = \min_{\mathcal{T}} \frac{1}{n} \Big( \mathbb{E}_{\mathbb{P}_0^{\mathcal{T}}} \Big[ \log \frac{\mathbb{P}_0^{\mathcal{T}}}{\mathbb{P}_{\hat{\phi}}}(\phi) \Big] + \mathbb{E}_{\mathbb{P}_0^{\mathcal{T}}} \Big[ \log \frac{\mathbb{P}_{\hat{\phi}}}{\mathbb{P}_{\tilde{\phi}}^A}(\phi) \Big] - \mathbb{E}_{\mathbb{P}_{\hat{\phi}}} \Big[ \log \frac{\mathbb{P}_{\hat{\phi}}}{\mathbb{P}_{\tilde{\phi}}^A}(\phi) \Big] \Big); \tag{3}$$

$$E_{gen} = \frac{1}{n} d_{\mathrm{KL}}(\mathbb{P}_{\hat{\phi}} \parallel \mathbb{P}_{\tilde{\phi}}^A).$$

*Here the minimization in Eq. (3) is over all orthogonal transforms $\mathcal{T} : \Phi = (Z, \alpha) \mapsto (ZU, \alpha)$, where $U$ is a $r$-dimensional rotation matrix, and $\mathbb{P}_0^{\mathcal{T}}$ is the distribution of $\mathcal{T}(\phi)$ when $\phi \sim \mathbb{P}_0$.*

The first term in Theorem 3.1 is referred to as $E_\rho$ since $p(A_{ij} \mid z_i^\top z_j + \alpha_i + \alpha_j + \rho_n^*)$ differs from $p(A_{ij} \mid z_i^\top z_j + \alpha_i + \alpha_j + \hat{\rho})$ only in the global sparsity parameter $\rho_n$. The second term is denoted as $E_\Phi$, since it primarily depends on the distance between $\mathbb{P}_{\hat{\phi}}$ and $\mathbb{P}_0$. The third term $E_{\mathrm{gen}}$ denotes the KL divergence between the conditional distributions of $\hat{\phi}$ and $\tilde{\phi}$ given $A$, and is determined by the generative model in Algorithm 1. Error analysis of such terms has been considered in the literature, for example, Chen et al. (2022; 2023a) for score-based generative models. In this analysis, we focus on characterizing the first two terms in Eq. (2), while noting that the third term concerns the generative error in a low-dimensional space rather than in the original large-scale network space.

We first consider the sparsity regime in the following assumption.

**Assumption 3.1** (network sparsity). $w_n = \exp(\rho_n^*) = \Omega(\log n/n)$.

Assumption 3.1 states that the edge density is bounded below by $\Omega(\log n/n)$, and accordingly the expected node degrees are at least of order $\log n$ as $n$ grows. Such a sparsity regime is consistent with the network analysis literature; see, for example, Athreya et al. (2018) and Ma et al. (2020).

In the sequel, we analyze $E_\rho$ and $E_\Phi$ under the asymptotic regime where $n \to \infty$. We first have the following theorem on $E_\rho$.

**Theorem 3.2.** *Under Assumptions 2.1 and 3.1, it holds that $E_\rho = O_p((w_n \cdot n)^{-1/2} \log n)$.*

Theorem 3.2 demonstrates that as long as $w_n \gg (\log n)^2/n$, the first error term satisfies $E_\rho = o_p(1)$. This requirement on $w_n$ is consistent with Assumption 3.1 up to a logrithmic factor. Analyzing the second term is challenging since it involves the marginal distribution of the estimated latent positions, which is practically hard to be compared with the true distribution $\mathbb{P}_0$. Following Wu et al. (2025), we use the technique of discretizing the underlying distribution to understand the approximation error between $\mathbb{P}_{\hat{\phi}}$ and $\mathbb{P}_0^{\mathcal{T}}$ for some transform $\mathcal{T}$. Suppose that $\mathbb{P}_0$ is a continuous distribution of the latent embeddings with density $p_0$ that follows Assumption 2.1. Since the support of $\mathbb{P}_0$ is bounded, we discretize the support of $\mathbb{P}_0$ into the following grid: $\mathcal{G}_{\gamma_n} = \{\phi \in \mathbb{R}^{r+1} : \phi_i/\gamma_n \in \mathbb{Z}, |\phi_i| \leq R + \gamma_n\}$, where $\{\gamma_n\}$ is a sequence of discretization scales that converge to zero. Then for any $\phi \in \mathcal{G}_{\gamma_n}$, we define the following mass function $q_{\gamma_n}(\phi) = \int_{\mathbb{R}^{r+1}} \prod_{i \leq r+1} \mathbb{1}\{\phi_i' \in [\phi_i - \gamma_n/2, \phi_i + \gamma_n/2]\} p_0(\phi') d\phi'$. Using the linearity of expectation, we conclude that $\sum_{\phi \in \mathcal{G}_{\gamma_n}} q_{\gamma_n}(\phi) = 1$. Therefore, $q_{\gamma_n}$ is a probability mass function. We claim that, as long as the original density function is sufficiently smooth, this discretized mass function is able to capture the structure of the original density function well. To this end, we define the projection operator associated with the grid as $\mathrm{proj}_{\mathcal{G}_{\gamma_n}}(\phi) = \arg\min_{\phi' \in \mathcal{G}_{\gamma_n}} \|\phi' - \phi\|_2$, and consider $p_{\gamma_n}(\phi) = q_{\gamma_n}(\mathrm{proj}_{\mathcal{G}_{\gamma_n}}(\phi)) \gamma_n^{-(r+1)}$. Then we have the following theorem.

**Theorem 3.3.** *Suppose that $p_0 : \mathbb{R}^{r+1} \to \mathbb{R}^+$ is L-Lipschitz, then it holds that $|p_0(\phi) - p_{\gamma_n}(\phi)| \leq L\gamma_n\sqrt{r+1}$.*

This theorem indicates that sampling from the original distribution is almost the same as sampling from $q_{\gamma_n}$. We consider $\Phi^\dagger = (\phi_1^\dagger, \ldots, \phi_n^\dagger)^\top \in \mathbb{R}^{n \times (r+1)}$ where the rows are independent realizations from $q_{\gamma_n}$. We denote the corresponding empirical mass function as $\hat{q}_{\gamma_n}(\phi) = n^{-1}\sum_{i \leq n} \mathbb{1}\{\phi_i^\dagger = \phi\}$ for $\phi \in \mathcal{G}_{\gamma_n}$. Then the following result indicates that $\hat{q}_{\gamma_n}$ is close to $q_{\gamma_n}$.

**Lemma 3.1.** *For any small $\delta > 0$, it holds that $\max_{\phi \in \mathcal{G}_{\gamma_n}} |q_{\gamma_n}(\phi) - \hat{q}_{\gamma_n}(\phi)| \leq \sqrt{\log(1/\delta) + \log(R/\gamma_n)}/\sqrt{n}$ with probability $1 - \delta$.*

When $\Phi^*$ is replaced by $\Phi^\dagger$, we need to modify the estimators in Eq. (1) accordingly using the projection operator $\text{proj}_{\mathcal{G}_{\gamma_n}}$. Given $\widehat{\Phi}$ and a rotation transform $\mathcal{T}$, we define the corresponding empirical distribution on the grid $\mathcal{G}_{\gamma_n}$ as follows: $\check{q}_{\gamma_n}(\phi) = \frac{1}{n}\sum_{i=1}^{n} \mathbb{1}\{\phi_i = \text{proj}_{\mathcal{G}_{\gamma_n}}(\mathcal{T}\hat{\phi}_i)\}$, $\phi \in \mathcal{G}_{\gamma_n}$. Next theorem shows that $\check{q}_{\gamma_n}$ is close to $\hat{q}_{\gamma_n}$, which is a direct consequence of the uniform consistency of the estimated latent embeddings.

**Theorem 3.4.** *Suppose that each $\phi$ is sampled from a discrete mass function $q_{\gamma_n}$ on $\mathcal{G}_{\gamma_n}$. Fix a sequence of discretization levels $\gamma_n = \Omega\big((w_n \cdot n)^{-1/2+\epsilon}\big)$ where $\epsilon > 0$ is fixed. Under Assumption 2.1 and Assumption 3.1, there exists a transform $\mathcal{T} : \phi = (Z, \alpha) \mapsto (ZU, \alpha)$ for some rotation matrix $U$, such that $\max_{\phi \in \mathcal{G}_\gamma} |\hat{q}_{\gamma_n}(\phi) - \check{q}_{\gamma_n}(\phi)| \to 0$, in probability as $n \to \infty$. Here, the randomness comes from the realizations of $A$.*

This result shows that the marginal distribution of any single point from $\{\text{proj}_{\gamma_n}(\hat{\phi}_i)\}_{i \leq n}$ is close to the discretized distribution $q_{\gamma_n}$, up to a rotation, and consequently controls $E_\Phi$. Proofs of all results in this section are provided in Appendix A.

## 4 EXPERIMENTS

In this section, we empirically evaluate the effectiveness and efficiency of the proposed SyNGLER framework using both simulated and real-world network datasets.

**Simulated network dataset.** We consider simulated sparse networks with sizes and latent dimensions $(n, r) \in \{500, 1000, 1500\} \times \{2, 3, 4\}$. For each $(n, r)$, we independently sample latent node embeddings $\{z_i^*\}_{i=1}^n$ from a truncated Gaussian mixture in $\mathbb{R}^r$ and degree parameters $\{\alpha_i^*\}_{i=1}^n$ from a uniform distribution. We adopt a logarithmic sparsity $\rho_n = -0.4\log(n)$, which results in an edge density that scales as $O(n^{-0.4})$. Such sparse configuration is able to closely characterize real-world network dataset. Based on the sampled latent embeddings, we generate a network from the binary logistic network model in Section 2.1. Each result is based on 200 Monte Carlo repetitions. Please see more details in Appendix C.2.

**Real-world datasets.** We use four large real-world networks: (i) the user-user friendship network from the Yelp Open Dataset (Yelp, 2024); (ii) the YouTube social network dataset (Yang & Leskovec, 2012); (iii) the DBLP co-authorship network (Yang & Leskovec, 2012); and (iv) the PolBlogs network (Adamic & Glance, 2005). Details regarding the preprocessing of these datasets are in Appendix C.4.

**Baselines and implementations.** We consider two diffusion-based approaches, SyNG-D and SyNG-D (MLP), where SyNG-D (MLP) uses a multilayer perceptron to approximate the score instead of the tree-based estimator in SyNG-D, and a resampling-based approach, SyNG-R, as defined in Section 2.2. SyNG-D and SyNG-R are included in all experiments, whereas SyNG-D (MLP) is evaluated only on real-world datasets. For baselines, VGAE (Kipf & Welling, 2016) is compared across all experiments evaluating structural recovery and efficiency. In the real-data evaluations, we also compare to GRAN (Liao et al., 2019), EDGE (Chen et al., 2023b), GraphMaker (Li et al., 2023), and the classical Erdős–Rényi model (Erdos & Rényi, 1960). Implementation details for our methods are provided in Appendix C.4. We also cover mKPGM (Moreno et al., 2013) and BTER (Kolda et al., 2014) for baselines. The implementations of the baselines on simulated and real-world datasets are provided separately in Appendix C.5 and Appendix C.7, respectively.

Table 1: Simulated sparse networks. Eigenvalues and degree centrality are evaluated by energy (En.) and Kolmogorov–Smirnov (KS) distances. Best results are in **bold**.

| $(n, r)$ | Method | Tri. ($\times 10^{-4}$) | | Clus. ($\times 10^{-3}$) | | Eig. ($\times 10^{-2}$) | | DegC. ($\times 10^{-2}$) | |
| --- | --- | --- | --- | --- | --- | --- | --- | --- | --- |
| | | RMSE | Bias | RMSE | Bias | En. | KS | En. | KS |
| | SyNG-D | **4.60** | **4.37** | **15.7** | **15.2** | **5.88** | **9.07** | **3.34** | **11.07** |
| $(500, 2)$ | SyNG-R | 5.37 | 5.11 | 18.1 | 17.7 | 6.38 | 9.78 | 3.69 | 12.45 |
| | VGAE | 9.70 | -9.16 | 53.1 | -50.2 | 10.18 | 15.67 | 10.69 | 44.45 |
| | SyNG-D | **0.63** | **0.57** | **4.58** | **4.07** | **7.09** | **12.09** | **1.59** | **6.42** |
| $(1500, 4)$ | SyNG-R | 1.35 | 1.26 | 9.80 | 9.65 | 9.41 | 14.45 | 2.52 | 9.58 |
| | VGAE | 3.05 | -2.78 | 37.4 | -35.0 | 17.26 | 32.46 | 9.25 | 48.18 |

## 4.1 STRUCTURAL PROPERTY RECOVERY

To evaluate the quality of the synthetic networks, we consider the distance between network statistics of the synthetic and the observed networks. Specifically, we evaluate triangle density (Tri., measuring the prevalence of triangle motifs / three-node interactions), clustering coefficient (Clus., summarizing local transitivity), eigenvalue distributions (Eig., reflecting global spectral structure), and degree centrality (DegC., describing node connectivity). For Tri. and Clus., which are single values for each network, we compute the root-mean-square error (RMSE) and bias relative to the observed network. For Eig. and DegC., which are vectors for each network, we compute the maximum mean discrepancy (MMD), the Kolmogorov-Smirnov (KS) statistic, and the energy distance. In addition, we also include the 4-graphlet frequency distance in the real-data experiment. More details are in Appendix C.3.

Table 1 summarizes the results for simulated networks. Our methods exhibit overall superior performance compared to the baseline VGAE model, reasonably due to the fact that the data are simulated from sparse latent space network models with a mixture distribution over latent embeddings. In the relatively large-scale setting with $(n, r) = (1500, 4)$, we note that SyNG-D outperforms SyNG-R on all metrics, indicating the effectiveness of generating novel node embeddings while capturing network structures. Here, we report only the results for $(n, r) \in \{(500, 2), (1500, 4)\}$ due to page constraints. More results on simulated networks are provided in Appendix C.5.

Real-world datasets allow a fairer comparison. For each method, we select the configuration that yields the best average performance across all four metrics. The results are summarized in Table 2. EDGE and GRAN ran out of memory on the Yelp dataset on a single NVIDIA GeForce RTX 4090 with memory of 24GB, and are marked "-" at the corresponding entries. This reflects that these methods are computationally expensive at scale and pose challenges when computational resources are limited. For most metrics, SyNG-D and SyNG-R produce networks with better-preserved characteristics. In Figure 2, we visualize the YouTube network alongside synthetic networks produced by different methods using the Fruchterman-Reingold force-directed algorithm. We also provide additional visualization results obtained using alternative tools in Appendix G. SyNG-D and SyNG-R preserve the clustering patterns in the observed network evidently better in comparison with other methods. More results and details are in Appendix C.7. Our proposed approach shows superior performances on all four networks. The improvements on Yelp and PolBlogs networks are clear.

## 4.2 ML UTILITY EVALUATION

We also evaluate the machine learning utility of the generated graphs, that is, whether synthetic graphs can effectively support downstream predictive tasks. Following the evaluation protocol proposed by Li et al. (2023), we adopt a discriminative-model-based framework to quantify utility. In this setting, a predictive model is first trained on the original graph $G$ and then separately on a generated graph $\hat{G}$. Both models are then evaluated on the test set of the original graph to obtain performance measures $\text{ACC}(G \mid G)$ and $\text{ACC}(G \mid \hat{G})$ (Li et al., 2023). A generated graph is regarded as having high ML utility if $\frac{\text{ACC}(G|\hat{G})}{\text{ACC}(G|G)} \approx 1$, indicating that training on the synthetic graph yields predictive performance comparable to training on the real graph. We consider the link prediction task, which aims to infer missing edges in a partially observed graph and may utilize node features and node labels where available. For this task, we adopt the Graph Autoencoder (GAE) model of Kipf & Welling (2016),

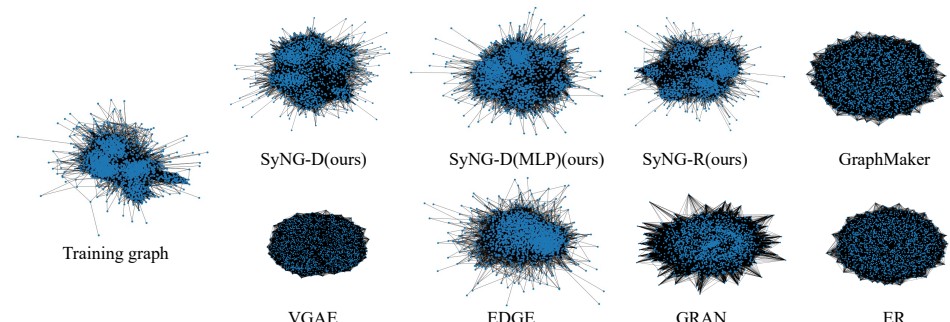

Figure 2: Visualization of the YouTube network and synthetic networks by different methods.

Table 2: Results on real-world networks. Eigenvalues and degree centrality are evaluated using MMD. Each entry is the average over 200 generated networks. Best performances are in **bold**.

(a) YouTube

| Method | Tri ($\times 10^{-4}$) | Clus ($\times 10^{-2}$) | Eig ($\times 10^{-2}$) | DegC ($\times 10^{-2}$) |
|---|---|---|---|---|
| E-R | 1.05 | 16.3 | 25.96 | 69.22 |
| BTER | **0.02** | 9.7 | 6.42 | **0.00** |
| mKPGM | 1.03 | 16.0 | 22.38 | 44.86 |
| VGAE | 0.559 | 8.0 | 17.20 | 27.83 |
| GRAN | 1.13 | 10.1 | 7.88 | 7.86 |
| EDGE | 0.785 | 14.4 | 4.44 | 7.84 |
| GraphMaker | 1.12 | 16.7 | 23.13 | 64.99 |
| SyNG-D | 0.967 | 2.28 | 2.61 | 2.13 |
| SyNG-D(MLP) | 1.62 | 1.58 | **0.378** | 3.89 |
| SyNG-R | 1.11 | **2.23** | 4.47 | 1.10 |

(b) DBLP

| Method | Tri ($\times 10^{-4}$) | Clus ($\times 10^{-1}$) | Eig ($\times 10^{-1}$) | DegC ($\times 10^{-1}$) |
|---|---|---|---|---|
| E-R | 7.96 | 8.94 | 4.409 | 8.716 |
| BTER | 6.09 | 6.78 | 1.484 | **0.462** |
| mKPGM | 8.00 | 9.00 | 3.003 | 4.568 |
| VGAE | **0.578** | **0.136** | 1.196 | 2.797 |
| GRAN | 7.99 | 8.92 | 1.757 | 2.973 |
| EDGE | 2.20 | 1.33 | 1.387 | 0.992 |
| GraphMaker | 7.98 | 8.98 | 3.931 | 7.512 |
| SyNG-D | 1.58 | 0.622 | 0.879 | 0.926 |
| SyNG-D(MLP) | 3.63 | 1.01 | 1.116 | 1.401 |
| SyNG-R | 1.43 | 0.378 | **0.799** | 0.748 |

(c) Yelp

| Method | Tri ($\times 10^{-4}$) | Clus ($\times 10^{-2}$) | Eig ($\times 10^{-2}$) | DegC ($\times 10^{-2}$) |
|---|---|---|---|---|
| E-R | 8.12 | 14.5 | 24.53 | 77.36 |
| BTER | 2.27 | 4.45 | 8.26 | **0.00** |
| mKPGM | 9.35 | 16.0 | 9.26 | 49.97 |
| VGAE | 7.41 | 10.3 | 17.58 | 33.74 |
| GRAN | - | - | - | - |
| EDGE | - | - | - | - |
| GraphMaker | 8.82 | 15.5 | 20.70 | 69.26 |
| SyNG-D | 2.00 | 1.77 | **2.50** | 6.72 |
| SyNG-D(MLP) | **0.745** | 2.35 | 9.62 | 3.26 |
| SyNG-R | 0.778 | **0.756** | 4.69 | 0.62 |

(d) PolBlogs

| Method | Tri ($\times 10^{-4}$) | Clus ($\times 10^{-2}$) | Eig ($\times 10^{-2}$) | DegC ($\times 10^{-2}$) |
|---|---|---|---|---|
| E-R | 3.22 | 20.4 | 40.10 | 79.24 |
| BTER | 0.88 | 5.27 | 3.04 | **0.00** |
| mKPGM | 3.32 | 21.4 | 31.15 | 50.67 |
| VGAE | 2.07 | 3.26 | 31.10 | 35.33 |
| GRAN | 1.57 | 11.4 | 9.04 | 13.86 |
| EDGE | 0.790 | 5.69 | 4.69 | **0.00** |
| GraphMaker | 3.27 | 20.8 | 38.28 | 74.15 |
| SyNG-D | 0.710 | **1.90** | **2.58** | 0.97 |
| SyNG-D(MLP) | 0.594 | 2.23 | 5.27 | 3.05 |
| SyNG-R | 0.830 | 2.45 | 3.01 | 0.92 |

and use AUROC as the performance measure ACC. Additional implementation details are provided in Appendix F.

Table 3 summarizes the results for ML utility ratios. Our SyNGLER-based methods consistently achieve ratios close to one, indicating strong preservation of the learning signal necessary for downstream ML tasks. These results suggest that SyNGLER not only captures structural characteristics but also maintains the discriminative information required for effective model training.

### 4.3 EFFICIENCY

**Evaluation metrics and configuration.** We compare training efficiency between SyNG-D and the baseline methods through a computational workload metric we define as the equivalent floating-point operations (e-FLOPs). To evaluate the training efficiency of different methods, the e-FLOPs metric

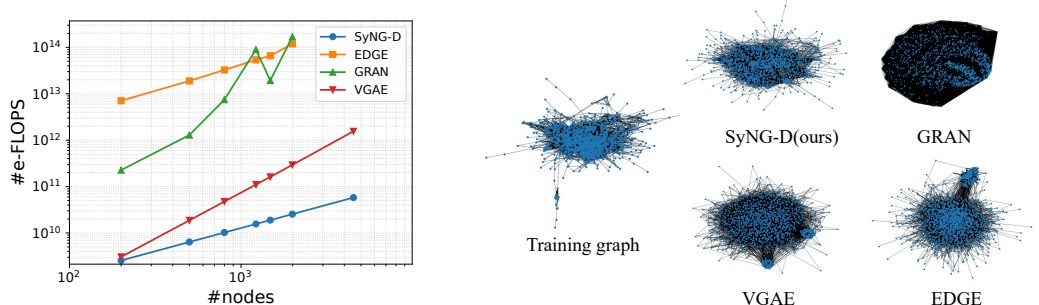

Figure 3: Efficiency comparison. Left: number of e-FLOPS versus the number of nodes in the observed graph. Right: Synthetic networks from different methods on the DBLP dataset.

counts the total number of floating-point operations or their approximate equivalents during the training process.

Note that SyNG-D and the baseline methods consist of neural nets and tree structures in their model architecture. For neural nets, e-FLOPs counts floating-point operations, whereas for tree structures, e-FLOPs counts node visits. We use e-FLOPs to evaluate the efficiency of different methods, since they are less dependent on the hardware environment. For the more straightforward wall-clock training time, we present the results along with the hardware environment used by each method in Appendix C.8. In addition to four real-world datasets, we also

Table 3: ML utility evaluation. "–": out-of-memory.

| Method | DBLP | PolBlogs | YouTube | Yelp |
|---|---|---|---|---|
| ER | $0.90 \pm 0.03$ | $\mathbf{1.00 \pm 0.00}$ | $0.99 \pm 0.00$ | $0.80 \pm 0.01$ |
| BTER | $0.85 \pm 0.05$ | $0.95 \pm 0.01$ | $0.91 \pm 0.01$ | $0.94 \pm 0.01$ |
| mKPGM | $0.96 \pm 0.02$ | $1.01 \pm 0.00$ | $0.92 \pm 0.01$ | $0.89 \pm 0.02$ |
| VGAE | $\mathbf{1.00 \pm 0.00}$ | $1.01 \pm 0.00$ | $\mathbf{1.00 \pm 0.00}$ | $\mathbf{1.00 \pm 0.00}$ |
| GRAN | $0.98 \pm 0.08$ | $0.92 \pm 0.08$ | $\mathbf{1.00 \pm 0.00}$ | - |
| EDGE | $\mathbf{1.00 \pm 0.00}$ | $0.98 \pm 0.01$ | $\mathbf{1.00 \pm 0.00}$ | - |
| GraphMaker | $0.95 \pm 0.02$ | $\mathbf{1.00 \pm 0.00}$ | $0.99 \pm 0.00$ | $0.83 \pm 0.01$ |
| SyNG-D | $\mathbf{1.00 \pm 0.00}$ | $0.99 \pm 0.00$ | $0.99 \pm 0.01$ | $0.99 \pm 0.00$ |
| SyNG-D(MLP) | $\mathbf{1.00 \pm 0.00}$ | $0.97 \pm 0.01$ | $0.99 \pm 0.00$ | $0.99 \pm 0.00$ |
| SyNG-R | $\mathbf{1.00 \pm 0.00}$ | $0.99 \pm 0.01$ | $0.99 \pm 0.01$ | $\mathbf{1.00 \pm 0.00}$ |

compare these methods on a group of simulated networks with sizes of 200, 500, and 800. For a fair comparison, we keep the latent dimension of all models at 4. Specifically, for SyNG-D and VGAE, the latent dimension corresponds to the dimension of the latent space. For GRAN, it is the output dimension of the attention layers. For EDGE, it is the dimension of the hidden layer in the score network.

**Results.** Figure 3 summarizes the results. We have several observations. In terms of e-FLOPs, SyNG-D attains the lowest cost across all settings and remains stable as network size grows. In terms of synthetic network quality, SyNG-D best preserves the overall structure of the observed networks; VGAE, while computationally comparable to SyNG-D, does not match its quality. Third, GRAN and EDGE require substantially longer training times, yet their performance is not as satisfactory, especially for GRAN. Overall, SyNG-D is both effective and efficient for this task, producing realistic synthetic networks with a computationally lightweight training process. Additional results regarding efficiency comparisons, along with further details, are provided in Appendix C.8.

## 5 CONCLUSION

In this work, we address the challenge of synthesizing realistic networks at scale while preserving salient structural properties using Synthetic Network Generation via Latent Embedding Reconstruction (SyNGLER), a general and efficient framework that learns low-dimensional latent node embeddings from a single observed network and then trains a distribution-free generator in the learned latent space. By separating representation learning via a likelihood-based latent space approach from generative modeling, SyNGLER preserves structural information with latent space geometry where lightweight generators suffice, enabling fast training and sampling. Theoretical and empirical results both demonstrate the effectiveness of SyNGLER. Future research directions include incorporating richer supervision for conditional generation (e.g., node/edge attributes and constraints), extending to directed, dynamic, and multilayer networks, and developing rigorous privacy-preserving training and release mechanisms.

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

## APPENDIX

We provide experimental details, additional numerical results, and proofs of the theoretical results in the appendix. Appendix A collects proofs of the theoretical results in Section 3. Appendix B then discusses the connections between latent space models and several classcial network models. Appendix C contains complete experimental details. Specifically, Appendix C.1 covers the details of the estimation algorithms. Appendix C.2 specifies the setup for the simulated networks and provides details on the real-world networks, as well as their preprocessing pipelines. Appendix C.3 includes the detailed description of the evaluation metrics in our experiments. Appendix C.4 contains implementation details for both SyNGLER and the baselines, including the device environments on which they are implemented and the hyperparameter configurations used. Appendix C.5 and Appendix C.7 collect additional results for the simulated networks and the real-world networks, respectively. Appendix C.8 provides additional numerical results on the efficiency analysis of different methods. We also provide a heuristic analysis on the sample complexities of different methods in Appendix C.9 The performance of the proposed method on large scale dataset is included in Appendix D. In Appendix E, we illustrate how to extend the SyNGLER to the attributed networks with demonstrations on the real-world datasets. In Appendix F, we introduce the pipeline of assessing ML utilities of the generated networks with experimental results. Finally, in Appendix G, we present a comprehensive visualization comparison of the generated networks from Section 4.

## A  PROOFS IN SECTION 3

### A.1  PROOF OF THEOREM 3.1

*Proof of Theorem 3.1.* Note that the marginal distribution of the network $A$ can be expressed as the product of the conditional distribution and the marginal distribution of the latent positions. Because of the identifiability issue of $Z$ in model Section 2.1, we consider any transform $\mathcal{T} : \Phi = (Z, \alpha) \mapsto (ZU, \alpha) \in \mathbb{R}^{n \times (r+1)}$, where $U$ is a $r$-dimensional rotation matrix. The transform $\mathcal{T}$ applied on $\Phi$ does not change the conditional distribution of $A$. We define $\mathbb{P}_{\mathcal{T}\Phi^*}$ as the distribution of $\mathcal{T}\Phi^*$, then

$$
d_{\mathrm{KL}}(\mathbb{P}_A \parallel \mathbb{P}_{\widetilde{A}}^A) = \mathbb{E}_{\mathbb{P}_{A \mid \Phi} \times \mathbb{P}_0^{\otimes n}} \Big[ \log \frac{\mathbb{P}_{A|\mathcal{T}\Phi^*}}{\mathbb{P}_{\widetilde{A}|\tilde{\Phi}}}(A \mid \Phi) + \log \frac{\mathbb{P}_{\mathcal{T}\Phi^*}}{\mathbb{P}_{\tilde{\Phi}}^A}(\Phi) \Big]
$$

$$
= \mathbb{E}_{\mathbb{P}_{A \mid \Phi} \times \mathbb{P}_0^n} \Big[ 2 \sum_{i<j} \log \frac{p(A_{ij} \mid z_i^\top z_j + \alpha_i + \alpha_j + \rho^*)}{p(\widetilde{A}_{ij} \mid z_i^\top z_j + \alpha_i + \alpha_j + \hat\rho)} \Big] + d_{\mathrm{KL}}(\mathbb{P}_{\mathcal{T}\Phi^*} \parallel \mathbb{P}_{\tilde{\Phi}}^A)
$$

$$
= n^2 \, \mathbb{E}_{\mathbb{P}_{A_{12} \mid \phi_1, \phi_2}} \Big[ \log \frac{p(A_{12} \mid z_1^\top z_2 + \alpha_1 + \alpha_2 + \rho^*)}{p(A_{12} \mid z_1^\top z_2 + \alpha_1 + \alpha_2 + \hat\rho)} \Big] + d_{\mathrm{KL}}(\mathbb{P}_{\mathcal{T}\Phi^*} \parallel \mathbb{P}_{\tilde{\Phi}}^A).
$$

Here the last equality holds because of the exchangeability of the distributions of $(\phi_1, \phi_2, \ldots, \phi_n)$. We decompose the K-L divergence in the right-hand side as follows

$$
d_{\mathrm{KL}}(\mathbb{P}_{\mathcal{T}\Phi^*} \parallel \mathbb{P}_{\tilde{\Phi}}^A) = n \cdot \mathbb{E}_{\mathbb{P}_0^{\mathcal{T}}} \Big[ \log \frac{\mathbb{P}_0^{\mathcal{T}}}{\mathbb{P}_{\tilde{\phi}}^A}(\phi) \Big]
$$

$$
= n \cdot \mathbb{E}_{\mathbb{P}_0^{\mathcal{T}}} \Big[ \log \frac{\mathbb{P}_0^{\mathcal{T}}}{\mathbb{P}_{\hat\phi}}(\phi) + \log \frac{\mathbb{P}_{\hat\phi}}{\mathbb{P}_{\tilde\phi}^A}(\phi) \Big]
$$

$$
= n \cdot \Big( \mathbb{E}_{\mathbb{P}_0^{\mathcal{T}}} \Big[ \log \frac{\mathbb{P}_0^{\mathcal{T}}}{\mathbb{P}_{\hat\phi}}(\phi) \Big] + \mathbb{E}_{\mathbb{P}_0^{\mathcal{T}}} \Big[ \log \frac{\mathbb{P}_{\hat\phi}}{\mathbb{P}_{\tilde\phi}^A}(\phi) \Big] - \mathbb{E}_{\mathbb{P}_{\hat\phi}} \Big[ \log \frac{\mathbb{P}_{\hat\phi}}{\mathbb{P}_{\tilde\phi}^A}(\phi) \Big]
$$

$$
+ \mathbb{E}_{\mathbb{P}_{\hat\phi}} \Big[ \log \frac{\mathbb{P}_{\hat\phi}}{\mathbb{P}_{\tilde\phi}^A}(\phi) \Big] \Big).
$$

Here $\mathbb{P}_{\hat\phi}$ is the marginal distribution of $\hat\phi_i$ for each $i$, which is the same for all $i$ because of the exchangeability. Minimizing over all transform $\mathcal{T}$ concludes the proof of the theorem. $\square$

## A.2 PROOF OF THEOREM 3.2

*Proof of Theorem 3.2.* We denote $\theta_{ij} = z_i^\top z_j + \alpha_i + \alpha_j$. The Lipschitz continuity of $l_{ij}(\pi) = \log p(A_{ij} \mid \pi)$ in $\pi$ implies that

$$\left| \log p(A_{ij} \mid \theta_{ij} + \rho^*) - \log p(A_{ij} \mid \theta_{ij} + \hat{\rho}) \right| \leq M|\rho^* - \hat{\rho}|.$$

Now combining with Lemma A.4 implies the desired result. □

## A.3 PROOF OF THEOREM 3.3

*Proof of Theorem 3.3.* Fix $\phi \in \mathbb{R}^{r+1}$ and set $\psi := \mathrm{proj}_{\mathcal{G}_{\gamma_n}}(\phi) \in \mathcal{G}_{\gamma_n}$. By the definition of $q_{\gamma_n}$, for any grid point $\psi \in \mathcal{G}_{\gamma_n}$ we can write

$$q_{\gamma_n}(\psi) = \int_{\mathbb{R}^{r+1}} \prod_{i=1}^{r+1} \mathbb{1}\{u_i \in [\psi_i - \gamma_n/2, \, \psi_i + \gamma_n/2]\}\, p_0(u)\, du = \int_{C(\psi)} p_0(u)\, du,$$

where $C(\psi) := \prod_{i=1}^{r+1} [\psi_i - \gamma_n/2, \, \psi_i + \gamma_n/2]$ is the hypercube that is centered at $\psi$ and has side length $\gamma_n$ with volume $\gamma_n^{r+1}$. By the definition of $p_{\gamma_n}$, we have

$$p_{\gamma_n}(\phi) = q_{\gamma_n}\big(\mathrm{proj}_{\mathcal{G}_{\gamma_n}}(\phi)\big)\, \gamma_n^{-(r+1)}$$
$$= q_{\gamma_n}(\psi)\, \gamma_n^{-(r+1)}$$
$$= \frac{1}{\gamma_n^{r+1}} \int_{C(\psi)} p_0(u)\, du.$$

Thus $p_{\gamma_n}(\phi)$ is exactly the average of $p_0$ over the cube $C(\psi)$.

Next we bound the distance between $\phi$ and any point $u \in C(\psi)$. Using the triangle inequality, we have for any $u \in C(\psi)$ that

$$\|u - \phi\| \leq \|u - \psi\| + \|\psi_i - \phi_i\| \leq \sqrt{r+1}\,\gamma_n.$$

Since $p_{\gamma_n}(\phi)$ is a local average of $p_0$ over the cube $C(\psi)$, we have by Lipschitz continuity of $p_0$ that

$$\big|p_0(\phi) - p_{\gamma_n}(\phi)\big| = \left| p_0(\phi) - \frac{1}{\gamma_n^{r+1}} \int_{C(\psi)} p_0(u)\, du \right|$$
$$= \frac{1}{\gamma_n^{r+1}} \left| \int_{C(\psi)} \big(p_0(\phi) - p_0(u)\big)\, du \right|$$
$$\leq \frac{1}{\gamma_n^{r+1}} \int_{C(\psi)} |p_0(\phi) - p_0(u)|\, du$$
$$\leq \frac{1}{\gamma_n^{r+1}} \int_{C(\psi)} L\gamma_n\sqrt{r+1}\, du$$
$$= L\gamma_n\sqrt{r+1} \cdot \frac{|(C(\psi))|}{\gamma_n^{r+1}}$$
$$= L\gamma_n\sqrt{r+1}.$$

Since $\phi$ was arbitrary, this completes the proof. □

## A.4 PROOF OF LEMMA 3.1

*Proof of Lemma 3.1.* For fixed $\phi \in \mathcal{G}$, using Hoeffding's inequality yields that

$$\mathbb{P}\Big(|q_{\gamma_n}(\phi) - \hat{q}_{\gamma_n}(\phi)| \geq t\Big) \leq 2\exp(-2nt^2).$$

Using the union bound over all $(R/\gamma_n)^{r+1}$ points in $\mathcal{G}_{\gamma_n}$ yields that

$$\mathbb{P}\Big(\max_{\phi \in \mathcal{G}_{\gamma_n}} |q_{\gamma_n}(\phi) - \hat{q}_{\gamma_n}(\phi)| \geq t\Big) \leq 2(R/\gamma_n)^{r+1}\exp(-2nt^2).$$

Therefore, setting $t = \sqrt{(r+1)\log(R/\gamma_n) + \log(2/\delta)}/\sqrt{2n}$ yields the desired result. □

## A.5 PROOF OF THEOREM 3.4

*Proof of Theorem 3.4.* Define the event $\mathcal{E}_n = \{n^{-1} \sum_{i \leq n} \|\hat{\phi}_i - \mathcal{T}\phi_i^*\|_2^2 \leq \gamma_n'^2\}$, where $\gamma_n' = \Omega((w_n n)^{-1/2+\epsilon/2})$ for some fixed $\epsilon > 0$. Then, we have that $\mathbb{P}(\mathcal{E}_n) \to 1$ as $n \to \infty$, as shown in Lemma A.4. On the other hand, we have that

$$\max_{\phi \in \mathcal{G}_{\gamma_n}} |\check{q}_{\gamma_n} - \hat{q}_{\gamma_n}| \leq \frac{1}{n} \sum_{i=1}^{n} \mathbb{1}\{\text{proj}_{\mathcal{G}_{\gamma_n}}(\hat{\phi}_i) \neq \text{proj}_{\mathcal{G}_{\gamma_n}}(\mathcal{T}\phi_i^*)\}$$

$$\leq \frac{1}{n} \sum_{i=1}^{n} \mathbb{1}\{\|\hat{\phi}_i - \mathcal{T}\phi_i^*\|_2 > 2\gamma_n\}$$

$$\leq \frac{1}{n} \sum_{i=1}^{n} \|\hat{\phi}_i - \mathcal{T}\phi_i^*\|_2^2 / (4\gamma_n^2).$$

Here, the last inequality holds by Markov's inequality. Therefore, we have that

$$\mathbb{P}\Big(\max_{\phi \in \mathcal{G}_{\gamma_n}} |\check{q}_{\gamma_n} - \hat{q}_{\gamma_n}| > \epsilon\Big) \leq \mathbb{P}\Big(\frac{1}{n} \sum_{i=1}^{n} \|\hat{\phi}_i - \mathcal{T}\phi_i^*\|_2^2 > 4\epsilon\gamma_n^2\Big)$$

$$\leq \mathbb{P}(\mathcal{E}_n^c) + \mathbb{P}\Big(\frac{1}{n} \sum_{i=1}^{n} \|\hat{\phi}_i - \mathcal{T}\phi_i^*\|_2^2 > 4\epsilon\gamma_n^2 \mid \mathcal{E}_n\Big)\mathbb{P}(\mathcal{E}_n).$$

For sufficiently large $n$ and fixed $\epsilon$, we have that $4\epsilon\gamma_n^2 > \gamma_n'^2$, therefore the second term vanishes for sufficiently large $n$. And the left-hand-side is upper bounded by $\mathbb{P}(\mathcal{E}_n^c)$, which goes to zero as $n \to \infty$. This concludes the proof of Theorem 3.4. □

## A.6 SUPPORTING LEMMAS AND PROOFS

**Lemma A.1** (Theorem 5 in Chung & Radcliffe (2011)). *Let $X_1, \ldots, X_m$ be independent random $n \times n$ Hermitian matrices. Assume $\|X_i - \mathbb{E}X_i\| \leq M$ for all $i$, and put*

$$\nu^2 = \Big\|\sum_{i=1}^{m} \text{Var}(X_i)\Big\|.$$

*Let $X = \sum_{i=1}^{m} X_i$. Then for any $a > 0$,*

$$\mathbb{P}(\|X - \mathbb{E}X\| > a) \leq 2n \exp\Big(-\frac{a^2}{2\nu^2 + \frac{2}{3}Ma}\Big).$$

**Lemma A.2.** *Under the model in Section 2.1 and Assumption 2.1, let $\partial l^* \in \mathbb{R}^{n \times n}$ be the matrix where each entry is $\partial l_{ij}^* = l'_{A_{ij}}(\pi_{ij}^*)$. Then we have that*

$$\|\partial l^*\| = O_p(w_n^{1/2} n^{1/2} (\log n)^{1/2}).$$

*Proof of Lemma A.2.* Let $E^{i,j}$ be the $n \times n$ matrix with 1 in the $(i,j)$ and $(j,i)$ positions and 0 elsewhere. Denote $p_{ij}^* = \mathbb{E}_A[l'_{A_{ij}}(\pi_{ij}^*)]$. To use Lemma A.1, write $\partial l^*$ as the sum of matrices $A^{i,j}$ defined as

$$A^{i,j} = (A_{ij} - p_{ij}^*)E^{i,j}, \qquad 1 \leq i < j \leq n,$$

so that $\partial l^* = \sum_{i=1}^{n} \sum_{j=i+1}^{n} A^{i,j}$. Note that $\|A^{i,j}\| \leq 1$, $\mathbb{E}[A^{i,j}] = 0_{n \times n}$, and

$$\mathbb{E}[(A^{i,j})^2] = (p_{ij}^* - (p_{ij}^*)^2)(E^{i,i} + E^{j,j}).$$

Let

$$\nu^2 = \Big\|\sum_{i=1}^{n} \sum_{j=i+1}^{n} \mathbb{E}[(A^{i,j})^2]\Big\| = \Big\|\sum_{i=1}^{n} \sum_{j=i+1}^{n} (p_{ij}^* - (p_{ij}^*)^2)(E^{i,i} + E^{j,j})\Big\|.$$

Then

$$\nu^2 = \left\| \sum_{i=1}^{n} \Big( \sum_{j=i+1}^{n} \big( p_{ij}^* - (p_{ij}^*)^2 \big) \Big) E^{i,i} + \sum_{j=2}^{n} \Big( \sum_{i=1}^{j-1} \big( p_{ij}^* - (p_{ij}^*)^2 \big) \Big) E^{j,j} \right\|$$

$$\leq 2 \max \left\{ \max_{i=1,\dots,n} \sum_{j=1}^{n} \big( p_{ij}^* - (p_{ij}^*)^2 \big) \right\} \leq 2n \max_{i,j} \big( p_{ij}^* - (p_{ij}^*)^2 \big) \leq 2n \max_{i,j} p_{ij}^*.$$

For $\epsilon' > 0$, we set $a = \sqrt{5n \max_{i,j} p_{ij}^* \, \log(2n/\epsilon')}$. By Assumption 2.1, for sufficiently large $n$, it holds that

$$n \max_{i,j} p_{ij}^* \geq 2\sqrt{\tfrac{5}{3} \, n \max_{i,j} p_{ij}^* \, \log(2n/\epsilon')}.$$

Applying Lemma A.1, we obtain

$$\mathbb{P}(\|\partial l^*\| > a) \leq 2n \exp \left\{ -\frac{5n \max_{i,j} p_{ij}^* \, \log(2n/\epsilon')}{4n \max_{i,j} p_{ij}^* + 2\sqrt{\tfrac{5}{3} \, n \max_{i,j} p_{ij}^* \, \log(2n/\epsilon')}} \right\}$$

$$\leq 2n \exp\{-\log(2n/\epsilon')\} = \epsilon'.$$

Noting that $\max_{i,j} p_{ij}^* \asymp w_n$ as $n \to \infty$, there exists $M' > 0$ such that, for any $\epsilon' > 0$,

$$\mathbb{P}\Big( \|\partial l^*\| \geq M'\sqrt{n w_n \log(n/\epsilon')} \Big) \leq \epsilon',$$

which concludes the proof. $\qquad\square$

We need the following lemma to characterize the estimation error of the latent embedding from the graph.

**Lemma A.3** (Lemma 5.4 in Tu et al. (2016))**.** *Suppose that $Z_1, Z_2 \in \mathbb{R}^{n \times r}$ are two matrices such that $\lambda_{\min}(Z_1^\top Z_1) > \sigma$. Then we have*

$$\inf_{U \in \mathcal{O}(r)} \|Z_1 - Z_2 U\|_F \leq \frac{1}{2(\sqrt{2}-1)\sigma^2} \|Z_1 Z_1^\top - Z_2 Z_2^\top\|_F.$$

**Lemma A.4.** *Suppose that each $l'_{ij}(\pi_{ij}^*)$ is i.i.d. bounded with mean zero. Then for any $\epsilon > 0$, there exists a constant $M > 0$ such that with probability at least $1 - \epsilon$, there exists a transform $\mathcal{T}$ such that*

$$\frac{1}{n} \sum_{i=1}^{n} \|\hat{\phi}_i - \mathcal{T}\phi_i^*\|_2^2 = O_p(w_n^{-1} n^{-1} \log n), \quad \text{and} \quad |\hat{\rho} - \rho^*| = O_p(w_n^{-1/2} n^{-1/2} \log n).$$

*Proof of Lemma A.4.* Let $\pi_{ij} = z_i^\top z_j + \alpha_i + \alpha_j + \rho$ and $\hat{\pi}$ be its estimated version. Applying Taylor's expansion to each $l_{ij}$ at $\pi_{ij}^*$ yields that

$$\sum_{i<j} l_{ij}(\hat{\pi}_{ij}) = \sum_{i<j} l_{ij}(\pi_{ij}^*) + \sum_{i<j} l'_{ij}(\pi_{ij}^*)(\hat{\pi}_{ij} - \pi_{ij}^*) + \frac{1}{2} \sum_{i<j} l''_{ij}(\xi_{ij})(\hat{\pi}_{ij} - \pi_{ij}^*)^2,$$

where each $\xi_{ij} \in [\min\{\hat{\pi}_{ij}, \pi_{ij}^*\}, \max\{\hat{\pi}_{ij}, \pi_{ij}^*\}]$. Using the optimality condition, it holds that $\sum_{i<j} l_{ij}(\hat{\pi}_{ij}) \geq \sum_{i<j} l_{ij}(\pi_{ij}^*)$. And therefore

$$\sum_{i<j} l'_{ij}(\pi_{ij}^*)(\hat{\pi}_{ij} - \pi_{ij}^*) \geq \sum_{i<j} -l''_{ij}(\xi_{ij})(\hat{\pi}_{ij} - \pi_{ij}^*)^2$$

$$\geq \frac{1}{4} w_n \cdot e^{-2R^2} \sum_{i<j} |\hat{\pi}_{ij} - \pi_{ij}^*|^2. \qquad (4)$$

To facilitate the matrix inequalities, we define $\partial l^*, \Pi^*, \hat{\Pi} \in \mathbb{R}^{n \times n}$ such that $\partial l^*_{ij} = l'_{A_{ij}}(\pi^*_{ij})$, $\Pi^*_{ij} = \pi^*_{ij} = z^{*\top}_i z^*_j + \alpha^*_i + \alpha^*_j + \rho^*_n$, and $\hat{\Pi}_{ij} = \hat{\pi}_{ij} = \hat{z}^\top_i \hat{z}_j + \hat{\alpha}_i + \hat{\alpha}^*_j + \hat{\rho}$. Then, we can upper bound the left-handed-side in Eq. (4) as

$$\sum_{i<j} l'_{ij}(\pi^*_{ij})(\hat{\pi}_{ij} - \pi^*_{ij}) \leq |\langle \partial l^*, \hat{\Pi} - \Pi^* \rangle|$$

$$\leq \sqrt{2r+3} \cdot \|\partial l^*\| \cdot \|\hat{\Pi} - \Pi^*\|_{\mathrm{F}}. \tag{5}$$

Last inequality holds since $\hat{\Pi} - \Pi^*$ has rank at most $2r+3$. Invoking Lemma A.2, we have that $\|\partial l^*\| = O_p(w_n^{1/2} n^{1/2} \log n)$. Combining Eqs. (4) and (5) yields that $\|\hat{\Pi} - \Pi^*\|_{\mathrm{F}} = O_p(w_n^{-1/2} n^{1/2} \log n)$.

Before obtaining the estimation error of $\hat{\rho}$, we need to involve an identifiability transform over $Z^*, \alpha^*$, since their sample average is not necessarily zero. Define $\beta = \frac{1}{n} Z^{*\top} \mathbf{1}_n$, $g = Z^* Z^{*\top} \mathbf{1}_n$, $a = \mathbf{1}^\top_n \alpha^*$ and $u = \mathbf{1}^\top_n Z^* Z^{*\top} \mathbf{1}_n$. Then we further define $Z^\dagger = Z^* - \mathbf{1}_n \beta^\top$, $\alpha^\dagger = \alpha^* + \frac{1}{2n} g - \frac{a}{n} \cdot \mathbf{1}_n - \frac{u}{n^2} \cdot \mathbf{1}_n$, and $\rho^\dagger = \rho^* + \frac{2a}{n} + \frac{u}{n^2}$. Then, it is evident that

$$Z^{\dagger \top} Z^\dagger + \alpha^\dagger \mathbf{1}^\top_n + \mathbf{1}_n \alpha^{\dagger \top} + \rho^\dagger \mathbf{1}_n \mathbf{1}^\top_n = Z^* Z^{*\top} + \alpha^* \mathbf{1}^\top_n + \mathbf{1}_n \alpha^{*\top} + \rho^* \mathbf{1}_n \mathbf{1}^\top_n.$$

Additionally, we have that $Z^{\dagger \top} \mathbf{1}_n = 0_n$ and $\mathbf{1}^\top_n \alpha^\dagger = 0$. It then follows that

$$\|Z^\dagger - Z^*\|_{\mathrm{F}} \leq \sqrt{n} \cdot \|\beta\|_2 = O_p(1), \tag{6}$$

$$\|\alpha^\dagger - \alpha^*\|_2 \leq \frac{1}{2n} \|g\|_2 + \frac{|a|}{n} \cdot \sqrt{n} + \frac{|u|}{n^2} \cdot \sqrt{n} = O_p(1), \tag{7}$$

$$|\rho^\dagger - \rho^*| \leq \frac{2|a|}{n} + \frac{|u|}{n^2} = O_p(n^{-1/2}). \tag{8}$$

On the other hand, we can expand $\|\hat{\Pi} - \Pi^*\|^2_{\mathrm{F}}$ as

$$\begin{aligned}
\|\hat{\Pi} - \Pi^*\|^2_{\mathrm{F}} &= \|\hat{Z}\hat{Z}^\top - Z^\dagger Z^{\dagger \top}\|^2_{\mathrm{F}} + \|(\hat{\alpha} - \alpha^\dagger)\mathbf{1}^\top_n + \mathbf{1}_n(\alpha^\dagger - \hat{\alpha})^\top\|^2_{\mathrm{F}} + n^2 |\hat{\rho} - \rho^\dagger|^2 \\
&\quad + 2\langle \hat{Z}\hat{Z}^\top - Z^\dagger Z^{\dagger \top}, (\hat{\alpha} - \alpha^\dagger)\mathbf{1}^\top_n + \mathbf{1}_n(\hat{\alpha} - \alpha^\dagger)^\top \rangle \\
&\quad + 2\langle \hat{Z}\hat{Z}^\top - Z^\dagger Z^{\dagger \top}, (\hat{\rho} - \rho^\dagger)\mathbf{1}_n \mathbf{1}^\top_n \rangle \\
&\quad + 2\langle (\hat{\alpha} - \alpha^\dagger)\mathbf{1}^\top_n + \mathbf{1}_n(\hat{\alpha} - \alpha^\dagger)^\top, (\hat{\rho} - \rho^\dagger)\mathbf{1}_n \mathbf{1}^\top_n \rangle \\
&= \|\hat{Z}\hat{Z}^\top - Z^\dagger Z^{\dagger \top}\|^2_{\mathrm{F}} + 2n\|\hat{\alpha} - \alpha^\dagger\|^2 + n^2 |\hat{\rho} - \rho^\dagger|^2.
\end{aligned}$$

Here the second line holds because $Z^{\dagger \top} \mathbf{1}_n = 0_n$ and $\mathbf{1}^\top_n \alpha^\dagger = 0$. Combining with the fact that $\|\hat{\Pi} - \Pi^*\|^2_{\mathrm{F}} = O_p(w_n^{-1} n \log n)$, we have that

$$\|\hat{Z}\hat{Z}^\top - Z^\dagger Z^{\dagger \top}\|^2_{\mathrm{F}} = O_p(w_n^{-1} n \log^2 n)$$

$$n^{-1} \|\hat{\alpha} - \alpha^\dagger\|^2_2 = O_p(w_n^{-1} n^{-1} \log^2 n)$$

$$|\hat{\rho} - \rho^\dagger|^2 = O_p(w_n^{-1} n^{-1} \log^2 n).$$

On the other hand, we have that $\lambda_{\min}(Z^{\dagger \top} Z^\dagger) = \Omega_p(n)$ because of Assumption 2.1. Using Lemma A.3, we have that $n^{-1} \|Z^\dagger - \hat{Z} U\|^2_{\mathrm{F}} = O_p(w_n^{-1} n^{-1} \log^2 n)$ for some rotation matrix $U$. In this sense, the fluctuation in Eqs. (6) to (8) is always of smaller magnitude. Thus, the desired result follows.

$\square$

# B  CONNECTIONS BETWEEN CLASSICAL NETWORK GENERATIVE MODELS AND THE LSM

We illustrate the connection between these models and the latent space model below. First of all, the general latent space network model assumes that each $A_{ij} \sim p(\cdot \mid \pi_{ij})$ where $\pi_{ij} = z^\top_i z_j + \alpha_i + \alpha_j$ and $p(\cdot \mid \pi)$ is a link function that can be chosen flexibly. Below, we explain its connection to several classical network models with details. We remark that some classical node-embedding models already belong to the latent space model. For example:

- The Chung-Lu graph model (Chung & Lu, 2002) assumes that each node $i$ is equipped with a degree parameter $w_i > 0$, and lets $W := \sum_{k=1}^{n} w_k$ be the total "weight". This model assumes independent edges with $\mathbb{P}(A_{ij} = 1) = w_i w_j / W$. We can then construct the latent embedding for the $i$-th node as $z_i = \frac{w_i}{\sqrt{W}} \in \mathbb{R}$, and collect them into the embedding matrix $Z = (z_1, \ldots, z_n)^\top \in \mathbb{R}^{n \times 1}$. With this parametrization, we have that $\mathbb{E}[A] = ZZ^\top$. Therefore, the Chung–Lu model is a latent space model with a linear link function $p(\cdot \mid \pi) = \text{Bernoulli}(\pi)$.

- The random dot product graph (RDPG) model (Athreya et al., 2018) assumes that each node $i$ has a latent position $z_i \in \mathbb{R}^r$ such that $z_i^\top z_j \in [0, 1]$ for all $i, j$. With the embedding matrix $Z = (z_1, \ldots, z_n)^\top \in \mathbb{R}^{n \times r}$, RDPG assumes that $A \sim \text{Bernoulli}(ZZ^\top)$, which is exactly a latent space model with the linear link function $p(\cdot \mid \pi) = \text{Bernoulli}(\pi)$.

Besides, many block-structured graph models also fall into the scope of latent space network models. For example:

- The (degree corrected) block model (DCBM/SBM) (Holland et al., 1983; Karrer & Newman, 2011), similar to the BTER model, assumes that each node is equipped with a cluster label $g_i \in [K]$ and a degree parameter $\theta_i$. Then it assumes that $\mathbb{E}[A]_{ij} = \theta_i \theta_j B_{g(i)g(j)}$ where $B \in [0, 1]^{K \times K}$ is symmetric and positive semi-definite, and $\theta_i \in [0, 1]$ is the degree parameter for node $i$. Let $B = UU^\top$ be the symmetric decomposition of $B$, where $U_k$ is the $k$-th row of $U$. Then, we can construct the latent embedding for the $i$-th node as $z_i = (\theta_i U_{g_i}^\top) \in \mathbb{R}^K$. Using the linear link function $p(\cdot \mid \pi) = \text{Bernoulli}(\pi)$, it can be formulated as the latent space model $A \sim p(\cdot \mid ZZ^\top)$

- Mixed-Member ship block models (Airoldi et al., 2008). Beyond classical block models, this model assumes that each node is associated with multiple blocks. Specifically, this model assumes that $\mathbb{E}[A]_{ij} = \pi_i B \pi_j$, where each $\pi_i$ belongs to the probability simplex $\Delta_K = \{\pi : \pi \in [0, 1]^K, \|\pi\|_1 = 1\}$. Whenever $B$ is positive definite with symmetric decomposition $B = UU^\top$ with $U \in \mathbb{R}^{K \times r}$, we can construct $Z = (z_1, \ldots, z_n)^\top$ such that its $i$-th row vector $z_i = U^\top \pi_i \in \mathbb{R}^K$. With the linear link function $p(\cdot \mid \pi) = \text{Bernoulli}(\pi)$, we have that $A \sim \text{Bernoulli}(ZZ^\top)$, which is an instance of latent space model.

In general, we see that these classical methods can generally be approximated by the latent space model with a suitable choice of the latent embedding and the link function. Therefore, we believe that the latent space model is sufficiently general to cover classical network models.

# C  Supplemental Materials for Experiments

## C.1  Deferred Algorithms

**Estimation in the latent space model.**  Suppose that we observe a network $A$, and we want to fit a latent space network model on $A$ with proper conditional model $p(\cdot \mid \cdot)$ and candidate latent dimension $r$. We use the following Algorithm 2 to solve Eq. (1).

---

**Algorithm 2** Projected Gradient Descent

---

**Require:** Network observation $A \in \mathbb{R}^{n \times n}$, model $p(\cdot|\cdot)$, stepsizes $\eta_Z, \eta_\alpha > 0$, number of iterations $N \in \mathbb{N}$;
1: **for** $i = 0$ **to** $N - 1$ **do**
2:     $\Pi \leftarrow ZZ^\top + \alpha \mathbf{1}^\top + \mathbf{1}\alpha^\top$;
3:     $Z \leftarrow Z + 2\eta_Z \, \partial_\pi p(A|\Pi)Z$;
4:     $\alpha \leftarrow \alpha + 2\eta_\alpha \, \partial_\pi p(A|\Pi)\mathbf{1}_n$;
5:     $Z \leftarrow \left(Z - n^{-1}\mathbf{1}_n\mathbf{1}_n^\top Z\right)R$, where $R \in \mathbb{R}^{r \times r}$ is the orthonormal matrix such that $n^{-1}(Z - n^{-1}\mathbf{1}_n\mathbf{1}_n^\top Z)^\top (Z - n^{-1}\mathbf{1}_n\mathbf{1}_n^\top Z)$.
6: **end for**
7: **return**  $\hat{Z} = Z, \hat{\alpha} = \alpha - \alpha^\top \mathbf{1}_n / n, \hat{\rho} = \alpha^\top \mathbf{1}_n$.

---

The convergence of Algorithm 2 in the well-specified setting can be found in Ma et al. (2020). In practice, we need to use a proper initialization for $Z$ and $\alpha$. And we use the output of universal singular value thresholding (USVT) (Chatterjee, 2015) as the initialization of $Z$ and $\alpha$. The detail of this initialization algorithm can be found in Ma et al. (2020).

## C.2 DATASETS DETAILS

**Simulated Datasets.** In the simulated datasets evaluation, we consider $(n,r) \in \{500, 1000, 1500\} \times \{2, 3, 4\}$. For each $(n, r)$ pair and each replicate $t = 1, \ldots, 200$, we generate an undirected sparse simple graph $A \in \{0, 1\}^{n \times n}$ as follows.

We first draw the degree parameters $\alpha_i \overset{\text{i.i.d.}}{\sim} \text{Unif}([-1/2, 1/2])$ for $i = 1, \ldots, n$ and set $\alpha = (\alpha_1, \ldots, \alpha_n)$. Let $\widetilde{z}_i \in \mathbb{R}^r$ be i.i.d. realizations of $\text{proj}_{[-2/\sqrt{r}, 2/\sqrt{r}]^r} \# \mathcal{N}_r(0, I_r/r)$ (i.e., a scaled Gaussian distribution truncated to $[-2/\sqrt{r}, 2/\sqrt{r}]^r$). We then independently draw two centers $v^{(1)}, v^{(2)} \in \mathbb{R}^r$ from $\text{Unif}([-1, 1]^r)$. For each node, we independently sample a label $L_i$ with $\mathbb{P}(L_i = 1) = \mathbb{P}(L_i = 2) = 1/2$ for $i = 1, \ldots, n$. Finally, we set $z'_i = \widetilde{z}_i + v^{(L_i)}$ and $z_i = z'_i \cdot (n^{-1} \| \sum_i z'_i z'^{\top}_i \|_{\text{F}})^{-1/2}$.

Given the latent positions and the degree parameters, we generate the network edges. We set the sparsity parameter $\rho^*_n = -0.4 \log n$. For each pair of nodes $1 \le i < j \le n$, we calculate $p_{ij} = \sigma(\alpha_i + \alpha_j + z^{\top}_i z_j + \rho^*_n)$. Then we independently sample $A_{ij} = A_{ij} \sim \text{Bernoulli}(p_{ij})$ for $i < j$ and set $A_{ii} = 0$ for all $i \le n$.

Table 4: Dataset statistics for Yelp, YouTube, DBLP and PolBlogs.

| Dataset | Original Dataset | | Subgraph | | Statistics | | |
|---|---|---|---|---|---|---|---|
| | Nodes | Edges | Nodes | Edges | Density | Clustering Coef. | Triangle Density |
| Yelp | 906,179 | 7,305,874 | 4,530 | 541,655 | 0.0527 | 0.1976 | 0.0010 |
| YouTube | 1,134,890 | 2,987,624 | 1,991 | 51,756 | 0.0261 | 0.1891 | 0.0002 |
| DBLP | 317,080 | 1,049,866 | 1,481 | 18,901 | 0.0172 | 0.9116 | 0.0008 |
| PolBlogs | 1,490 | 19,090 | 1,222 | 16,714 | 0.0224 | 0.2259 | 0.0003 |

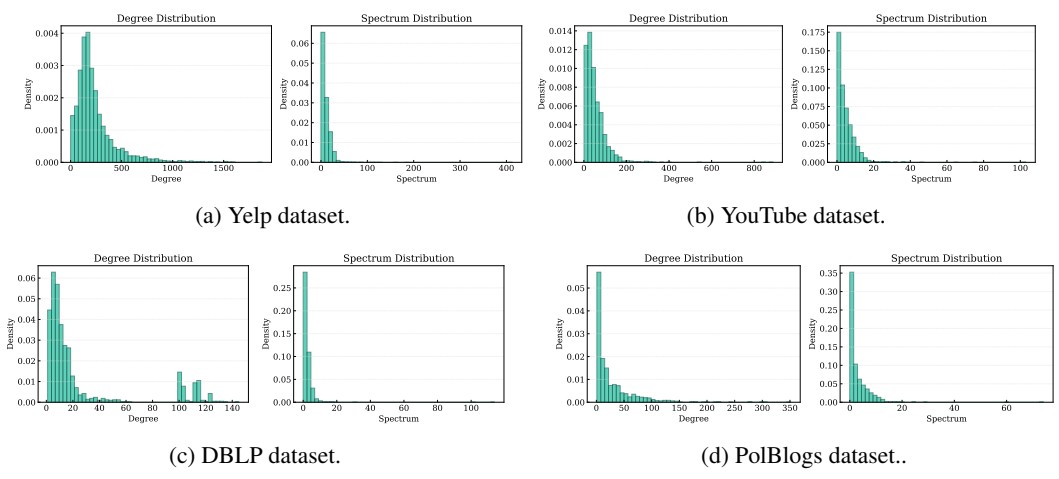

(a) Yelp dataset.     (b) YouTube dataset.

(c) DBLP dataset.     (d) PolBlogs dataset..

Figure 4: Degree and eigenvalue distributions for four real-world datasets.

**Real-world datasets.** We evaluate on four networks spanning thousands to millions of nodes. For Yelp, YouTube, and DBLP, whose full graphs are extremely large and highly sparse, we construct tractable training sets by extracting high-degree nodes and then taking the largest connected component (LCC).

In the Yelp and YouTube datasets, nodes represent users and an undirected edge between users represents a social tie (friendship/subscription). In the DBLP dataset, nodes represent authors, and an

edge connects two authors if they have coauthored at least one paper. In the PolBlogs dataset, nodes represent U.S. political blogs from the 2004 election blogosphere and two blogs are connected if one of them contains a link to the other.

Since the Yelp, YouTube, and DBLP datasets are very large and contain many low-degree nodes, we sample induced subgraphs for tractable evaluation. Our general procedure is to rank nodes by degree, take the induced subgraph on the top-$k$ nodes (for $k$ between 1,000 and 5,000), and then extract the largest connected component (LCC). For Yelp, we select the top $0.5\%$ of users by degree, yielding an LCC of 4,530 nodes and 541,655 edges. For YouTube and DBLP, we take the top 2,000 and 1,500 nodes, resulting in LCCs of 1,991 and 1,481 nodes, respectively. To avoid out-of-memory (OOM) issues for some baseline methods, we cap most subgraphs at $\leq 2,000$ nodes. The PolBlogs network is relatively smaller, so we use its full LCC of 1,222 nodes and 16,714 edges. For all networks, we symmetrize edges and remove self-loops. Key statistics for the original and the extracted graphs are in Table 4, with degree and eigenvalue(spectrum) distributions shown in Figure 4.

## C.3 EVALUATION DETAILS

**Metrics for similarity.** We assess the quality of the generated networks by comparing some metrics that capture both numerical and structural aspects of a network. For the *numerical* characteristics, we use the triangle density and the global clustering coefficient. For the *structural* characteristics, we consider the distribution of degree centralities and the eigenvalues of the adjacency matrix. For any network adjacency $A$, we consider the following numerical characteristics:

- The triangle density: $\mathsf{TD}(A) = \mathsf{NT}(A)/\binom{n}{3}$ where $\mathsf{NT}(A) = \frac{1}{6}\operatorname{tr}(A^3)$ is the number of triangles in the graph;

- The global clustering coefficient: $\mathsf{GC}(A) = 3\,\mathsf{NT}(A)\big/\sum_{i=1}^{n}\binom{d_i}{2}$ where $d_i = \sum_{j\neq i} A_{ij}$ is the degree of node $i$.

For each input network and generative model, we generate $S = 200$ independent networks $\widetilde{A}_1, \ldots, \widetilde{A}_S$ and compute the empirical distribution of each numerical characteristic. Specifically, for a numerical characteristic $f$ with $f \in \{\mathsf{TD}, \mathsf{GC}\}$ and a collection of generated networks $\{\widetilde{A}^{(s)}\}_{s=1}^{S}$, we compute

$$\mathrm{RMSE}_f = \Big(\frac{1}{S}\sum_{s=1}^{S}\big(f(A) - f(\widetilde{A}^{(s)})\big)^2\Big)^{1/2},$$

$$\mathrm{MAE}_f = \frac{1}{S}\sum_{s=1}^{S}\big|f(A) - f(\widetilde{A}^{(s)})\big|,$$

$$\mathrm{Bias}_f = \frac{1}{S}\sum_{s=1}^{S}\big(f(\widetilde{A}^{(s)}) - f(A)\big).$$

For the structural characteristics, we consider the following:

- The degree centrality: $\mathsf{DC}(A) = (d_1, \ldots, d_n)$, where $d_i = \sum_{j\neq i} A_{ij}$ is the degree of node $i$;

- The eigenvalues: $\mathsf{EV}(A) = (\lambda_1, \ldots, \lambda_n)$, where $\lambda_1 \geq \lambda_2 \geq \cdots \geq \lambda_n$ are the eigenvalues of $A$.

For two vectors, we consider the Wasserstein distance 1-distance, the Kolmogorov–Smirnov distance, energy distance and maximum mean discrepancy (MMD) as follows:

$$W^1(u, v) = \frac{1}{n} \sum_{i=1}^{n} |u_{(i)} - v_{(i)}|,$$

$$\mathrm{KS}(u, v) = \sup_{x \in \mathbb{R}} \left| \frac{1}{n} \sum_{i=1}^{n} \mathbf{1}\{u_i \le x\} - \frac{1}{n} \sum_{i=1}^{n} \mathbf{1}\{v_i \le x\} \right|,$$

$$\mathrm{ED}(u, v) = \frac{2}{n^2} \sum_{i,j=1}^{n} |u_i - v_j| - \frac{1}{n^2} \sum_{i,j=1}^{n} |u_i - u_j| - \frac{1}{n^2} \sum_{i,j=1}^{n} |v_i - v_j|,$$

$$\mathrm{MMD}(u, v) = \frac{1}{n^2} \sum_{i,j=1}^{n} k(u_i, u_j) + \frac{1}{n^2} \sum_{i,j=1}^{n} k(v_i, v_j) - \frac{2}{n^2} \sum_{i,j=1}^{n} k(u_i, v_j),$$

where $u_{(1)} \le u_{(2)} \le \cdots \le u_{(n)}$ are the order statistics of $u$, and $k(x, y) = \exp(-|x - y|^2/2)$ is the standard Gaussian RBF kernel. For a structural characteristic $f$ with $f \in \{\mathsf{DC}, \mathsf{EV}\}$ and a discrepancy metric $d$ with $d \in \{W^1, \mathrm{KS}, \mathrm{ED}, \mathrm{MMD}\}$, we compute the average distance between the original network and the generated networks as $\bar{d}_f = \frac{1}{S} \sum_{s=1}^{S} d(f(A), f(\widetilde{A}^{(s)}))$.

We additionally consider the graphlet frequency as the structural distance and calculate the $L^1/L^2$ distances between the original graph and the generated graph. The graphlet frequency is defined as follows. Let $\mathcal{S}_4 := \{S \subset [n] : |S| = 4\}$ be the collection of all 4-vertex subsets of $V$. For any $S = (i_1, i_2, i_3, i_4)$ with $i_1 < i_2 < i_3 < i_4$, we define the local degrees for $r \le 4$ as

$$d_r(S) := \sum_{\substack{s \in S \\ s \ne i_r}} A_{i_r s}, \qquad r = 1, 2, 3, 4.$$

Then we consider its order statistics $d_{(1)}(S) \le d_{(2)}(S) \le d_{(3)}(S) \le d_{(4)}(S)$ and the total edges $e(S) := \frac{1}{2} \sum_{r=1}^{4} d_r(S)$.

The six connected 4-node graphlets are denoted as

$$\mathcal{G}_4 := \{K_{1,3}, P_4, C_4, T, D, K_4\},$$

where $K_{1,3}$ is the 3-star, $P_4$ is the 4-path, $C_4$ is the 4-cycle, $T$ is the *triangled tail* (a triangle with a pendant vertex), $D$ is the *diamond* ($K_4$ with one edge removed), and $K_4$ is the complete graph on 4 vertices. Each graphlet $g \in \mathcal{G}_4$ is uniquely characterized (by isomorphism type) by its edge count $e_g$ and its ordered degree sequence $\delta_g = (\delta_{g,1}, \delta_{g,2}, \delta_{g,3}, \delta_{g,4})$, namely

$$
\begin{aligned}
g = K_{1,3} &\iff e_g = 3, \delta_g = (1, 1, 1, 3), \\
g = P_4 &\iff e_g = 3, \delta_g = (1, 1, 2, 2), \\
g = C_4 &\iff e_g = 4, \delta_g = (2, 2, 2, 2), \\
g = T &\iff e_g = 4, \delta_g = (1, 2, 2, 3), \\
g = D &\iff e_g = 5, \delta_g = (2, 2, 3, 3), \\
g = K_4 &\iff e_g = 6, \delta_g = (3, 3, 3, 3).
\end{aligned}
$$

For each $g \in \mathcal{G}_4$, define the indicator

$$\mathbf{1}_g(S) := \mathbf{1}\Big\{ e(S) = e_g, \ \big(d_{(1)}(S), d_{(2)}(S), d_{(3)}(S), d_{(4)}(S)\big) = \delta_g \Big\}, \qquad S \in \mathcal{S}_4.$$

We aggregate them to get the corresponding 4-node graphlet count: $C_g(A) := \sum_{S \in \mathcal{S}_4} \mathbf{1}_g(S)$, and normalize to get the 4-graphlet frequency $\mathrm{GF}_4(A)$ such that the $g$-th coordinate is $\mathrm{GF}_4(A)_g := \frac{C_g(A)}{\sum_{g' \in \mathcal{G}_4} C_{g'}(A)}$. We use $L^1$ and $L^2$ to measure the distance between two graphs, namely

$$\mathrm{GFD}_{L^1}(A, A') = \sum_{g \in \mathcal{G}_4} |\mathrm{GF}_4(A)_g - \mathrm{GF}_4(A')_g|;$$

$$\mathrm{GFD}_{L^2}(A, A') = \Big( \sum_{g \in \mathcal{G}_4} |\mathrm{GF}_4(A)_g - \mathrm{GF}_4(A')_g|^2 \Big)^{1/2}.$$

**Evaluation pipeline.** For a single input network $A$, we generate $S = 200$ independent networks and calculate the above metrics. In the real-world dataset setting, we directly report the averaged metrics for each input network and its associated standard deviation.

## C.4 Implementation Details

**Implementation of SyNGLER.** For the SyNG-D, we use `ForestDiffusion` (Jolicoeur-Martineau et al., 2024) to construct the score approximation. Table 5 lists all the hyperparameters for `ForestDiffusion` throughout our experiments.

Table 5: `ForestDiffusion` Hyperparameters For Data Generation.

| Category | Hyperparameter | Simulated Data | Real-world data | Description |
|---|---|---|---|---|
| ForestDiffusion | $n_t$ | 50 | 100 | Number of diffusion time steps. |
| | duplicate K | 100 | 100 | Sample duplication factor for training data. |
| | diffusion type | vp | vp | Use variance-preserving (VP) diffusion. |
| XGBoost | max depth | 7 | 7 | Maximum tree depth. |
| | number of estimators | 100 | 100 | Number of boosting trees. |
| | eta | 0.3 | 0.3 | Learning rate. |
| | tree method | hist | hist | Histogram-based tree construction. |
| | regression lambda | 0.0 | 0.0 | L2 regularization parameter. |
| | regression alpha | 0.0 | 0.0 | L1 regularization parameter. |
| | subsample | 1.0 | 1.0 | Row subsampling ratio per tree. |

**Experimental environment.** All experiments are conducted on NVIDIA GeForce RTX 4090 (24 GB) GPUs and 384 CPU cores.

The `ForestDiffusion` module is parallelized on the CPU and executes entirely on host cores. We deploy VGAE and SyNG-D models on CPUs. EDGE and GRAN are deployed on a single GPU, according to the default configuration in the original codebase.

**Implementation for baselines.** For the VGAE, EDGE and GRAN, we use the codebases hosted in gae, graph-generation-EDGE, GRAN, respectively. For each baseline method, we adopt the default models in the corresponding codebase without further clarification. In the real data experiments, we include the results for each method with several different configurations. For our SyNG-D and SyNG-R, we vary the number of the latent dimension $r$ from 2 to 6. For VGAE, we vary the number of the embeddings consecutively from 2 to 6, and include the default setting 16. For GRAN, we choose the dimension of the hidden layer from 128, 256 and 512. Other methods remain their default configurations.

## C.5 Evaluation Results on Non-sparse Simulated Network

**Structural characteristics.** The following Table 6 and Table 7 summarize the discrepancies between the structural statistics of the generated networks and those of the input networks.

Table 6: Averaged distance between the *degree centralities* of the original network and the generated output. All values are reported as $\times 10^{-1}$.

| Metric | Method | $n = 500$ | | | $n = 1000$ | | | $n = 1500$ | | |
|---|---|---|---|---|---|---|---|---|---|---|
| | | r=2 | r=3 | r=4 | r=2 | r=3 | r=4 | r=2 | r=3 | r=4 |
| W1-dist | SyNG-D | **0.08 ± 0.02** | **0.08 ± 0.02** | **0.08 ± 0.02** | **0.05 ± 0.01** | **0.05 ± 0.02** | **0.06 ± 0.02** | **0.04 ± 0.01** | **0.04 ± 0.01** | **0.04 ± 0.01** |
| | SyNG-R | **0.08 ± 0.02** | **0.08 ± 0.02** | **0.08 ± 0.02** | **0.05 ± 0.01** | 0.06 ± 0.02 | **0.06 ± 0.02** | **0.04 ± 0.01** | **0.04 ± 0.01** | **0.04 ± 0.01** |
| | VGAE | 0.49 ± 0.02 | 0.49 ± 0.03 | 0.50 ± 0.03 | 0.49 ± 0.02 | 0.50 ± 0.03 | 0.50 ± 0.03 | 0.50 ± 0.02 | 0.51 ± 0.02 | 0.51 ± 0.02 |
| KS-dist | SyNG-D | **0.69 ± 0.10** | **0.68 ± 0.10** | **0.70 ± 0.13** | **0.47 ± 0.07** | **0.47 ± 0.09** | **0.48 ± 0.09** | **0.38 ± 0.06** | **0.38 ± 0.07** | **0.38 ± 0.07** |
| | SyNG-R | 0.72 ± 0.09 | 0.72 ± 0.09 | 0.73 ± 0.12 | 0.49 ± 0.06 | 0.49 ± 0.08 | 0.51 ± 0.08 | 0.39 ± 0.05 | 0.40 ± 0.06 | 0.40 ± 0.06 |
| | VGAE | 4.56 ± 0.23 | 4.53 ± 0.24 | 4.57 ± 0.28 | 4.51 ± 0.19 | 4.51 ± 0.20 | 4.52 ± 0.21 | 4.52 ± 0.16 | 4.53 ± 0.16 | 4.55 ± 0.17 |
| Energy-dist | SyNG-D | **0.27 ± 0.06** | **0.27 ± 0.06** | **0.28 ± 0.07** | **0.19 ± 0.04** | **0.19 ± 0.05** | **0.19 ± 0.05** | **0.15 ± 0.03** | **0.15 ± 0.04** | **0.15 ± 0.04** |
| | SyNG-R | **0.27 ± 0.05** | 0.28 ± 0.05 | **0.28 ± 0.07** | **0.19 ± 0.04** | **0.19 ± 0.05** | **0.19 ± 0.05** | **0.15 ± 0.03** | **0.15 ± 0.04** | **0.15 ± 0.04** |
| | VGAE | 1.72 ± 0.06 | 1.72 ± 0.07 | 1.73 ± 0.07 | 1.72 ± 0.05 | 1.73 ± 0.06 | 1.73 ± 0.06 | 1.73 ± 0.05 | 1.75 ± 0.05 | 1.75 ± 0.05 |

**Numerical characteristics.** The following Table 8 and Table 9 summarize the distances between the numerical characteristics of the generated networks and the input networks.

Table 7: Averaged distance between the *eigenvalues* of the original network and the generated output. All values are reported as $\times 10^{-1}$.

| eigenvalues Distance | Method | n = 500 | | | n = 1000 | | | n = 1500 | | |
|---|---|---|---|---|---|---|---|---|---|---|
| | | r=2 | r=3 | r=4 | r=2 | r=3 | r=4 | r=2 | r=3 | r=4 |
| W1-dist | SyNG-D | **0.37** ± 0.04 | **0.43** ± 0.09 | **0.66** ± 0.37 | **0.24** ± 0.03 | **0.26** ± 0.04 | **0.36** ± 0.13 | **0.19** ± 0.02 | **0.20** ± 0.03 | **0.25** ± 0.05 |
| | SyNG-R | 0.40 ± 0.04 | 0.59 ± 0.09 | 0.98 ± 0.18 | 0.25 ± 0.03 | 0.35 ± 0.05 | 0.56 ± 0.15 | 0.20 ± 0.02 | 0.25 ± 0.03 | 0.38 ± 0.06 |
| | VGAE | 9.54 ± 0.51 | 9.70 ± 0.67 | 9.55 ± 0.78 | 9.55 ± 0.46 | 9.80 ± 0.69 | 9.91 ± 0.76 | 9.52 ± 0.50 | 9.80 ± 0.70 | 9.83 ± 0.82 |
| KS-dist | SyNG-D | **0.47** ± 0.06 | **0.48** ± 0.11 | **0.62** ± 0.29 | **0.32** ± 0.03 | **0.32** ± 0.06 | **0.38** ± 0.15 | **0.27** ± 0.03 | **0.26** ± 0.04 | **0.28** ± 0.06 |
| | SyNG-R | 0.52 ± 0.06 | 0.65 ± 0.12 | 0.96 ± 0.22 | 0.35 ± 0.03 | 0.42 ± 0.08 | 0.59 ± 0.18 | 0.28 ± 0.02 | 0.32 ± 0.05 | 0.42 ± 0.09 |
| | VGAE | 9.65 ± 0.01 | 9.61 ± 0.02 | 9.61 ± 0.04 | 9.81 ± 0.01 | 9.82 ± 0.01 | 9.81 ± 0.02 | 9.87 ± 0.01 | 9.88 ± 0.01 | 9.87 ± 0.01 |
| Energy-dist | SyNG-D | **0.28** ± 0.04 | **0.29** ± 0.07 | **0.41** ± 0.21 | **0.18** ± 0.02 | **0.18** ± 0.04 | **0.23** ± 0.11 | **0.15** ± 0.02 | **0.14** ± 0.03 | **0.16** ± 0.04 |
| | SyNG-R | 0.30 ± 0.04 | 0.39 ± 0.08 | 0.64 ± 0.16 | 0.19 ± 0.02 | 0.23 ± 0.05 | 0.36 ± 0.12 | **0.15** ± 0.01 | 0.17 ± 0.03 | 0.25 ± 0.06 |
| | VGAE | 10.15 ± 0.23 | 10.40 ± 0.31 | 10.48 ± 0.36 | 10.55 ± 0.20 | 10.78 ± 0.30 | 10.92 ± 0.32 | 10.70 ± 0.22 | 10.89 ± 0.28 | 10.99 ± 0.33 |

Table 8: Similarity between the the *triangle densities* of the original network and generated network. All values are reported as $\times 10^{-2}$.

| n | Method | RMSE | | | MAE ($\times 10^{-2}$) | | | Bias ($\times 10^{-2}$) | | |
|---|---|---|---|---|---|---|---|---|---|---|
| | | r=2 | r=3 | r=4 | r=2 | r=3 | r=4 | r=2 | r=3 | r=4 |
| 500 | SyNG-D | 0.45 | **0.44** | 0.50 | 0.37 ± 0.26 | **0.47** ± 0.34 | 0.50 ± 0.49 | 0.18 ± 0.42 | **0.10** ± 0.43 | 0.22 ± 0.66 |
| | SyNG-R | **0.43** | **0.44** | **0.49** | **0.35** ± 0.26 | 0.51 ± 0.37 | **0.39** ± 0.30 | **0.17** ± 0.40 | 0.13 ± 0.42 | **0.18** ± 0.46 |
| | VGAE | 0.68 | 0.73 | 0.71 | 0.58 ± 0.36 | 0.61 ± 0.39 | 0.58 ± 0.41 | -0.54 ± 0.42 | -0.57 ± 0.45 | -0.52 ± 0.49 |
| 1000 | SyNG-D | 0.31 | **0.32** | **0.33** | 0.32 ± 0.17 | **0.31** ± 0.25 | 0.35 ± 0.29 | **0.09** ± 0.29 | **0.05** ± 0.32 | **0.06** ± 0.45 |
| | SyNG-R | **0.30** | 0.43 | 0.34 | **0.25** ± 0.17 | 0.34 ± 0.26 | **0.27** ± 0.20 | **0.09** ± 0.28 | 0.19 ± 0.18 | 0.10 ± 0.32 |
| | VGAE | 0.66 | 0.67 | 0.68 | 0.58 ± 0.32 | 0.59 ± 0.33 | 0.59 ± 0.33 | -0.56 ± 0.35 | -0.56 ± 0.38 | -0.56 ± 0.38 |
| 1500 | SyNG-D | 0.31 | **0.31** | **0.32** | 0.25 ± 0.14 | **0.24** ± 0.19 | **0.25** ± 0.21 | 0.08 ± 0.30 | **0.03** ± 0.25 | **-0.03** ± 0.32 |
| | SyNG-R | **0.24** | 0.32 | 0.34 | **0.20** ± 0.13 | 0.25 ± 0.20 | 0.27 ± 0.20 | **0.04** ± 0.23 | 0.13 ± 0.30 | 0.13 ± 0.31 |
| | VGAE | 0.71 | 0.65 | 0.65 | 0.65 ± 0.29 | 0.59 ± 0.28 | 0.58 ± 0.31 | -0.65 ± 0.29 | -0.57 ± 0.32 | -0.57 ± 0.33 |

Table 9: Similarity between the the *global clustering coefficients* of the original network and generated network.

| n | Method | RMSE | | | MAE ($\times 10^{-2}$) | | | Bias ($\times 10^{-2}$) | | |
|---|---|---|---|---|---|---|---|---|---|---|
| | | r=2 | r=3 | r=4 | r=2 | r=3 | r=4 | r=2 | r=3 | r=4 |
| 500 | SyNG-D | 0.62 | **0.58** | **0.70** | **0.51** ± 0.34 | **0.47** ± 0.34 | **0.50** ± 0.49 | **0.34** ± 0.51 | **0.21** ± 0.54 | **0.22** ± 0.66 |
| | SyNG-R | **0.61** | 0.63 | **0.70** | **0.51** ± 0.34 | 0.51 ± 0.37 | 0.57 ± 0.41 | 0.37 ± 0.49 | 0.36 ± 0.52 | 0.43 ± 0.56 |
| | VGAE | 2.59 | 2.60 | 2.58 | 2.49 ± 0.72 | 2.45 ± 0.86 | 2.40 ± 0.93 | -2.49 ± 0.72 | -2.45 ± 0.86 | -2.40 ± 0.94 |
| 1000 | SyNG-D | **0.40** | **0.39** | **0.45** | **0.32** ± 0.23 | **0.31** ± 0.25 | **0.35** ± 0.29 | **0.17** ± 0.36 | **0.07** ± 0.39 | **0.06** ± 0.45 |
| | SyNG-R | **0.40** | 0.43 | 0.52 | **0.32** ± 0.24 | 0.34 ± 0.26 | 0.41 ± 0.32 | 0.20 ± 0.35 | 0.19 ± 0.38 | 0.26 ± 0.45 |
| | VGAE | 2.59 | 2.57 | 2.61 | 2.50 ± 0.70 | 2.43 ± 0.85 | 2.47 ± 0.85 | -2.50 ± 0.70 | -2.43 ± 0.85 | -2.47 ± 0.85 |
| 1500 | SyNG-D | 0.31 | **0.31** | **0.32** | 0.25 ± 0.17 | **0.24** ± 0.19 | **0.25** ± 0.21 | **0.08** ± 0.30 | **0.03** ± 0.31 | **-0.03** ± 0.32 |
| | SyNG-R | **0.30** | 0.32 | 0.34 | **0.24** ± 0.17 | 0.25 ± 0.20 | 0.27 ± 0.20 | 0.10 ± 0.28 | 0.13 ± 0.30 | 0.13 ± 0.31 |
| | VGAE | 2.71 | 2.46 | 2.53 | 2.63 ± 0.64 | 2.35 ± 0.74 | 2.40 ± 0.82 | -2.63 ± 0.64 | -2.35 ± 0.74 | -2.40 ± 0.82 |

## C.6 EVALUATION RESULTS ON SPARSE SIMULATED NETWORK

**Structural characteristics.** Below, we present the quality of the synthetic networks on the sparse simulated networks. The following Table 10 and Table 11 summarize the discrepancies between the structural statistics of the generated networks and those of the input networks.

Table 10: Averaged distance between the *degree centralities* of the original network and the generated output, sparse network. All values reported as $\times 10^{-1}$.

| Metric | Method | n = 500 | | | n = 1000 | | | n = 1500 | | |
|---|---|---|---|---|---|---|---|---|---|---|
| | | r=2 | r=3 | r=4 | r=2 | r=3 | r=4 | r=2 | r=3 | r=4 |
| W1-dist | SyNG-D | **0.07** ± 0.02 | **0.11** ± 0.11 | 0.51 ± 0.55 | **0.04** ± 0.01 | **0.04** ± 0.01 | **0.05** ± 0.09 | **0.03** ± 0.01 | **0.03** ± 0.01 | **0.03** ± 0.01 |
| | SyNG-R | 0.08 ± 0.02 | 0.12 ± 0.03 | 0.24 ± 0.12 | 0.04 ± 0.01 | 0.05 ± 0.01 | 0.07 ± 0.02 | 0.03 ± 0.01 | 0.04 ± 0.01 | 0.05 ± 0.01 |
| | VGAE | 0.21 ± 0.02 | 0.20 ± 0.02 | **0.19** ± 0.02 | 0.18 ± 0.02 | 0.17 ± 0.02 | 0.16 ± 0.02 | 0.16 ± 0.02 | 0.15 ± 0.02 | 0.15 ± 0.02 |
| KS-dist | SyNG-D | **1.11** ± 0.24 | **1.44** ± 0.88 | 3.75 ± 2.78 | **0.71** ± 0.16 | **0.75** ± 0.20 | **0.92** ± 0.60 | **0.55** ± 0.13 | **0.58** ± 0.15 | **0.64** ± 0.17 |
| | SyNG-R | 1.25 ± 0.25 | 1.65 ± 0.39 | **2.93** ± 1.13 | 0.79 ± 0.16 | 0.99 ± 0.20 | 1.29 ± 0.27 | 0.60 ± 0.13 | 0.75 ± 0.15 | 0.96 ± 0.17 |
| | VGAE | 4.45 ± 0.83 | 4.45 ± 0.83 | 4.50 ± 0.79 | 4.53 ± 0.83 | 4.57 ± 0.85 | 4.62 ± 0.85 | 4.62 ± 0.89 | 4.81 ± 0.79 | 4.82 ± 0.71 |
| Energy-dist | SyNG-D | **0.33** ± 0.08 | **0.46** ± 0.37 | 1.61 ± 1.47 | **0.19** ± 0.05 | **0.20** ± 0.06 | **0.26** ± 0.26 | **0.14** ± 0.04 | **0.14** ± 0.04 | **0.16** ± 0.05 |
| | SyNG-R | 0.37 ± 0.08 | 0.52 ± 0.14 | **1.01** ± 0.45 | 0.21 ± 0.05 | 0.27 ± 0.06 | 0.37 ± 0.09 | 0.15 ± 0.04 | 0.19 ± 0.04 | 0.25 ± 0.05 |
| | VGAE | 1.07 ± 0.11 | 1.06 ± 0.11 | 1.05 ± 0.11 | 0.98 ± 0.11 | 0.97 ± 0.11 | 0.96 ± 0.11 | 0.94 ± 0.11 | 0.93 ± 0.10 | 0.93 ± 0.08 |

Table 11: Averaged distance between the *eigenvalues* of the original network and the generated output, sparse network. All values reported as $\times 10^{-1}$.

| eigenvalues Distance | Method | $n = 500$ | | | $n = 1000$ | | | $n = 1500$ | | |
|---|---|---|---|---|---|---|---|---|---|---|
| | | r=2 | r=3 | r=4 | r=2 | r=3 | r=4 | r=2 | r=3 | r=4 |
| W1-dist | SyNG-D | **0.66** ± 0.12 | **1.38** ± 1.23 | 6.34 ± 6.01 | **0.36** ± 0.06 | **0.56** ± 0.12 | **1.05** ± 1.02 | **0.26** ± 0.04 | **0.37** ± 0.08 | **0.63** ± 0.16 |
| | SyNG-R | 0.74 ± 0.11 | 1.46 ± 0.26 | 2.99 ± 0.72 | 0.40 ± 0.06 | 0.76 ± 0.11 | 1.38 ± 0.22 | 0.28 ± 0.04 | 0.51 ± 0.07 | 0.93 ± 0.14 |
| | VGAE | 1.72 ± 0.49 | 1.60 ± 0.53 | **1.42** ± 0.47 | 1.86 ± 0.55 | 1.73 ± 0.61 | 1.66 ± 0.58 | 1.86 ± 0.57 | 1.79 ± 0.61 | 1.63 ± 0.59 |
| KS-dist | SyNG-D | **0.91** ± 0.23 | **1.53** ± 0.70 | 3.56 ± 1.85 | **0.57** ± 0.12 | **0.91** ± 0.28 | **1.53** ± 0.63 | **0.46** ± 0.10 | **0.70** ± 0.22 | **1.21** ± 0.41 |
| | SyNG-R | 0.98 ± 0.24 | 1.65 ± 0.46 | 3.03 ± 0.85 | 0.62 ± 0.12 | 1.06 ± 0.29 | 1.81 ± 0.52 | 0.47 ± 0.10 | 0.80 ± 0.23 | 1.44 ± 0.42 |
| | VGAE | 1.57 ± 0.27 | 1.55 ± 0.53 | **1.54** ± 0.58 | 2.15 ± 0.49 | 2.40 ± 0.83 | 2.61 ± 0.94 | 2.58 ± 0.64 | 3.12 ± 0.86 | 3.25 ± 0.94 |
| Energy-dist | SyNG-D | **0.59** ± 0.13 | **1.15** ± 0.71 | 3.69 ± 2.62 | **0.33** ± 0.06 | **0.56** ± 0.16 | **1.03** ± 0.58 | **0.24** ± 0.05 | **0.39** ± 0.11 | **0.71** ± 0.22 |
| | SyNG-R | 0.64 ± 0.13 | 1.25 ± 0.28 | 2.49 ± 0.60 | 0.36 ± 0.06 | 0.71 ± 0.15 | 1.31 ± 0.29 | 0.26 ± 0.04 | 0.49 ± 0.11 | 0.94 ± 0.20 |
| | VGAE | 1.02 ± 0.29 | 1.03 ± 0.44 | **0.98** ± 0.43 | 1.38 ± 0.46 | 1.45 ± 0.57 | 1.53 ± 0.61 | 1.56 ± 0.49 | 1.74 ± 0.56 | 1.73 ± 0.59 |

**Numerical characteristics.** The following Table 12 and Table 13 summarize the distances between the numerical characteristics of the generated networks and the input networks.

Table 12: Similarity between the the *triangle densities* of the original network and generated network, sparse network.

| $n$ | Method | RMSE ($\times 10^{-4}$) | | | MAE ($\times 10^{-4}$) | | | Bias ($\times 10^{-4}$) | | |
|---|---|---|---|---|---|---|---|---|---|---|
| | | r=2 | r=3 | r=4 | r=2 | r=3 | r=4 | r=2 | r=3 | r=4 |
| 500 | SyNG-D | **4.60** | 18.34 | 157.01 | **4.37** ± 1.44 | 8.16 ± 16.47 | 90.99 ± 128.29 | **4.37** ± 1.44 | 8.16 ± 16.47 | 90.99 ± 128.29 |
| | SyNG-R | 5.37 | **7.31** | 17.22 | 5.11 ± 1.66 | **6.94** ± 2.32 | 14.73 ± 8.94 | 5.11 ± 1.66 | **6.94** ± 2.32 | 14.73 ± 8.94 |
| | VGAE | 9.70 | 8.46 | **7.63** | 9.16 ± 3.20 | 7.97 ± 2.85 | **7.19** ± 2.56 | −9.16 ± 3.20 | −7.97 ± 2.85 | **−7.19** ± 2.56 |
| 1000 | SyNG-D | **1.60** | **1.29** | 15.14 | **1.51** ± 0.53 | **1.21** ± 0.46 | 3.06 ± 14.87 | **1.51** ± 0.53 | **1.20** ± 0.46 | 3.06 ± 14.87 |
| | SyNG-R | 1.88 | 2.22 | **2.91** | 1.77 ± 0.62 | 2.08 ± 0.77 | **2.75** ± 0.94 | 1.77 ± 0.62 | 2.08 ± 0.77 | **2.75** ± 0.94 |
| | VGAE | 5.95 | 4.92 | 4.52 | 5.55 ± 2.14 | 4.53 ± 1.93 | 4.19 ± 1.70 | −5.55 ± 2.14 | −4.53 ± 1.93 | −4.19 ± 1.70 |
| 1500 | SyNG-D | **0.87** | **0.62** | **0.63** | **0.80** ± 0.34 | **0.57** ± 0.22 | **0.58** ± 0.25 | **0.80** ± 0.34 | **0.57** ± 0.23 | **0.57** ± 0.27 |
| | SyNG-R | 0.99 | 1.07 | 1.35 | 0.92 ± 0.36 | 1.00 ± 0.37 | 1.26 ± 0.48 | 0.92 ± 0.36 | 1.00 ± 0.37 | 1.26 ± 0.48 |
| | VGAE | 4.59 | 3.34 | 3.05 | 4.26 ± 1.73 | 3.05 ± 1.36 | 2.78 ± 1.25 | −4.26 ± 1.73 | −3.05 ± 1.36 | −2.78 ± 1.25 |

Table 13: Similarity between the the *global clustering coefficients* of the original network and generated network, sparse network.

| $n$ | Method | RMSE ($\times 10^{-2}$) | | | MAE ($\times 10^{-2}$) | | | Bias ($\times 10^{-2}$) | | |
|---|---|---|---|---|---|---|---|---|---|---|
| | | r=2 | r=3 | r=4 | r=2 | r=3 | r=4 | r=2 | r=3 | r=4 |
| 500 | SyNG-D | **1.57** | 4.62 | 19.69 | **1.52** ± 0.40 | 2.42 ± 3.95 | 13.33 ± 14.53 | **1.52** ± 0.40 | 2.42 ± 3.95 | 13.33 ± 14.53 |
| | SyNG-R | 1.81 | **2.37** | 4.22 | 1.77 ± 0.38 | **2.33** ± 0.47 | 3.99 ± 1.39 | 1.77 ± 0.38 | **2.33** ± 0.47 | 3.99 ± 1.39 |
| | VGAE | 5.31 | 4.51 | **4.09** | 5.03 ± 1.71 | 4.28 ± 1.45 | **3.89** ± 1.25 | −5.02 ± 1.75 | −4.26 ± 1.48 | **−3.87** ± 1.32 |
| 1000 | SyNG-D | **0.84** | **0.69** | 4.05 | **0.82** ± 0.20 | **0.65** ± 0.24 | 1.16 ± 3.89 | **0.82** ± 0.20 | **0.65** ± 0.24 | 1.15 ± 3.89 |
| | SyNG-R | 0.98 | 1.18 | **1.52** | 0.96 ± 0.20 | 1.16 ± 0.21 | **1.50** ± 0.27 | 0.96 ± 0.20 | 1.16 ± 0.21 | 1.50 ± 0.27 |
| | VGAE | 5.37 | 4.40 | 4.10 | 5.05 ± 1.81 | 4.10 ± 1.60 | 3.81 ± 1.52 | −5.05 ± 1.81 | −4.08 ± 1.65 | −3.78 ± 1.58 |
| 1500 | SyNG-D | **0.59** | **0.46** | **0.46** | **0.57** ± 0.16 | **0.43** ± 0.16 | **0.42** ± 0.19 | **0.57** ± 0.16 | **0.43** ± 0.17 | **0.41** ± 0.21 |
| | SyNG-R | 0.67 | 0.80 | 0.98 | 0.65 ± 0.14 | 0.79 ± 0.14 | 0.96 ± 0.17 | 0.65 ± 0.14 | 0.79 ± 0.14 | 0.96 ± 0.17 |
| | VGAE | 5.43 | 4.06 | 3.74 | 5.09 ± 1.91 | 3.79 ± 1.45 | 3.50 ± 1.32 | −5.09 ± 1.91 | −3.79 ± 1.47 | −3.50 ± 1.32 |

## C.7 EVALUATION RESULTS ON REAL-WORLD DATASETS

In this subsection, we list all the experiment results on the real-world datasets. For each dataset, we present three tables detailing the generation quality. The first table evaluates the similarity of degree centrality distributions, the second assesses the similarity of eigenvalue distributions, and the third reports on numerical characteristics such as global clustering coefficient and triangle density.

**YouTube dataset.** Tables 14 to 16 summarize the generation quality on the YouTube dataset.

Table 14: Generation quality in degree centralities distribution similarity on YouTube dataset. All reported distance values are scaled by $10^{-2}$.

| Method | Config | W1 dist. | KS dist. | Energy dist. | MMD |
|---|---|---|---|---|---|
| SyNG-D | 2 | **0.15 ± 0.07** | **3.76 ± 1.43** | **0.01 ± 0.01** | **2.13 ± 1.80** |
| | 3 | 0.23 ± 0.09 | 5.61 ± 1.67 | 0.01 ± 0.01 | 4.46 ± 2.02 |
| | 4 | 0.32 ± 0.10 | 7.42 ± 1.91 | 0.03 ± 0.02 | 6.71 ± 2.15 |
| | 5 | 0.49 ± 0.09 | 11.79 ± 1.78 | 0.07 ± 0.02 | 11.49 ± 1.90 |
| | 6 | 0.58 ± 0.09 | 13.92 ± 1.87 | 0.10 ± 0.03 | 13.80 ± 1.98 |
| SyNG-D (MLP) | 2 | 0.19 ± 0.05 | 6.05 ± 1.42 | **0.01 ± 0.00** | 4.24 ± 1.50 |
| | 3 | **0.18 ± 0.05** | 5.91 ± 1.29 | **0.01 ± 0.00** | 4.06 ± 1.43 |
| | 4 | **0.18 ± 0.05** | **5.88 ± 1.51** | 0.01 ± 0.01 | **3.89 ± 1.68** |
| | 5 | 0.22 ± 0.05 | 8.00 ± 1.57 | **0.01 ± 0.01** | 6.34 ± 1.48 |
| | 6 | 0.29 ± 0.08 | 9.82 ± 1.64 | 0.03 ± 0.01 | 8.13 ± 1.78 |
| SyNG-R | 2 | 0.13 ± 0.06 | **2.86 ± 1.13** | **0.00 ± 0.00** | **1.10 ± 1.43** |
| | 3 | 0.13 ± 0.06 | 2.91 ± 1.16 | 0.00 ± 0.00 | 1.21 ± 1.47 |
| | 4 | 0.13 ± 0.06 | 2.91 ± 1.11 | 0.00 ± 0.00 | 1.18 ± 1.50 |
| | 5 | 0.13 ± 0.06 | 2.98 ± 1.16 | 0.00 ± 0.00 | 1.15 ± 1.47 |
| | 6 | 0.13 ± 0.06 | 3.00 ± 1.21 | 0.00 ± 0.00 | 1.16 ± 1.52 |
| VGAE | 2 | 0.89 ± 0.01 | 23.74 ± 0.44 | 0.20 ± 0.00 | 33.85 ± 0.46 |
| | 3 | 0.89 ± 0.01 | 23.49 ± 0.40 | 0.20 ± 0.00 | 33.39 ± 0.55 |
| | 4 | **0.80 ± 0.01** | **20.10 ± 0.43** | **0.15 ± 0.00** | **27.83 ± 0.41** |
| | 5 | 0.91 ± 0.01 | 24.40 ± 0.39 | 0.21 ± 0.00 | 34.96 ± 0.36 |
| | 6 | 0.82 ± 0.01 | 20.33 ± 0.43 | 0.16 ± 0.00 | 28.78 ± 0.36 |
| | 16 | 0.85 ± 0.01 | 21.76 ± 0.43 | 0.18 ± 0.00 | 31.03 ± 0.40 |
| GRAN | 128 | 0.78 ± 0.38 | 15.38 ± 5.47 | 0.12 ± 0.17 | 16.02 ± 4.24 |
| | 256 | 5.71 ± 0.41 | 57.59 ± 3.00 | 3.87 ± 0.48 | 62.02 ± 3.20 |
| | 512 | **0.76 ± 0.20** | **9.24 ± 2.00** | **0.08 ± 0.04** | **7.86 ± 1.72** |
| EDGE | – | **0.41 ± 0.05** | **7.82 ± 0.86** | **0.04 ± 0.01** | **7.84 ± 0.95** |
| GraphMaker | – | **1.35 ± 0.00** | **37.68 ± 0.47** | **0.55 ± 0.01** | **64.99 ± 0.47** |
| E-R | – | **1.41 ± 0.01** | **47.08 ± 0.44** | **0.67 ± 0.01** | **69.22 ± 0.40** |
| BTER | – | **0.04 ± 0.01** | **1.64 ± 0.32** | **0.00 ± 0.00** | **0.00 ± 0.00** |
| mKPGM | – | **1.06 ± 0.01** | **31.71 ± 0.39** | **0.33 ± 0.01** | **44.86 ± 0.45** |

**DBLP dataset.** The following Tables 17 to 19 present the full experimental results for the DBLP dataset.

**Yelp dataset.** Here we provide the detailed evaluation for the Yelp dataset. The results for degree centrality, eigenvalue distribution, and other numerical characteristics are shown in the Tables 20 to 22 respectively.

**PolBlogs dataset.** Finally, we present the comprehensive results for the PolBlogs dataset. The subsequent Tables 23 to 25 detail the performance of each method in capturing the structural and numerical properties of the original network.

Table 15: Generation quality in eigenvalue distribution similarity on YouTube dataset. W1 dist. is reported in its original scale, while KS dist., Energy dist., and MMD are scaled by $10^{-1}$.

| Method | Config | W1 dist. | KS dist. | Energy dist. | MMD |
|---|---|---|---|---|---|
| SyNG-D | 2 | 0.31 ± 0.06 | 0.30 ± 0.07 | 0.11 ± 0.05 | 0.26 ± 0.09 |
| | 3 | 0.23 ± 0.04 | 0.20 ± 0.06 | 0.05 ± 0.03 | 0.08 ± 0.10 |
| | 4 | **0.19 ± 0.04** | **0.14 ± 0.04** | **0.03 ± 0.01** | **0.01 ± 0.03** |
| | 5 | 0.30 ± 0.08 | 0.20 ± 0.06 | 0.07 ± 0.04 | 0.02 ± 0.05 |
| | 6 | 0.40 ± 0.08 | 0.28 ± 0.07 | 0.14 ± 0.07 | 0.11 ± 0.11 |
| SyNG-D (MLP) | 2 | 0.30 ± 0.06 | 0.28 ± 0.06 | 0.10 ± 0.05 | 0.24 ± 0.09 |
| | 3 | 0.28 ± 0.06 | 0.27 ± 0.07 | 0.09 ± 0.05 | 0.20 ± 0.11 |
| | 4 | 0.19 ± 0.05 | 0.18 ± 0.06 | 0.04 ± 0.03 | 0.04 ± 0.07 |
| | 5 | **0.17 ± 0.04** | **0.14 ± 0.04** | **0.03 ± 0.02** | 0.01 ± 0.04 |
| | 6 | 0.19 ± 0.06 | **0.14 ± 0.04** | **0.03 ± 0.02** | **0.00 ± 0.01** |
| SyNG-R | 2 | 0.43 ± 0.07 | 0.44 ± 0.07 | 0.24 ± 0.09 | 0.45 ± 0.09 |
| | 3 | 0.38 ± 0.06 | 0.39 ± 0.07 | 0.19 ± 0.07 | 0.39 ± 0.10 |
| | 4 | 0.33 ± 0.06 | 0.36 ± 0.06 | 0.15 ± 0.06 | 0.34 ± 0.09 |
| | 5 | 0.31 ± 0.06 | 0.36 ± 0.06 | 0.14 ± 0.06 | 0.32 ± 0.09 |
| | 6 | **0.29 ± 0.05** | **0.35 ± 0.06** | **0.13 ± 0.05** | **0.31 ± 0.09** |
| VGAE | 2 | 1.27 ± 0.01 | 1.51 ± 0.01 | 2.35 ± 0.04 | 1.85 ± 0.01 |
| | 3 | 1.26 ± 0.01 | 1.51 ± 0.01 | 2.33 ± 0.04 | 1.84 ± 0.01 |
| | 4 | **1.15 ± 0.01** | **1.41 ± 0.01** | **1.94 ± 0.03** | **1.72 ± 0.01** |
| | 5 | 1.29 ± 0.01 | 1.53 ± 0.01 | 2.41 ± 0.04 | 1.87 ± 0.01 |
| | 6 | 1.17 ± 0.01 | 1.42 ± 0.01 | 2.02 ± 0.03 | 1.74 ± 0.01 |
| | 16 | 1.21 ± 0.01 | 1.46 ± 0.01 | 2.14 ± 0.04 | 1.79 ± 0.01 |
| GRAN | 128 | 0.97 ± 0.46 | **0.70 ± 0.29** | 1.14 ± 1.25 | **0.71 ± 0.32** |
| | 256 | 4.00 ± 0.23 | 2.40 ± 0.12 | 14.20 ± 1.53 | 2.58 ± 0.14 |
| | 512 | **0.94 ± 0.17** | 0.76 ± 0.10 | **0.99 ± 0.33** | 0.79 ± 0.08 |
| EDGE | – | **0.34 ± 0.04** | **0.39 ± 0.08** | **0.14 ± 0.05** | **0.44 ± 0.07** |
| GraphMaker | – | **1.62 ± 0.01** | **1.81 ± 0.01** | **3.68 ± 0.04** | **2.31 ± 0.01** |
| E-R | – | **1.95 ± 0.01** | **2.10 ± 0.01** | **5.54 ± 0.06** | **2.60 ± 0.01** |
| BTER | – | **0.76 ± 0.01** | **0.60 ± 0.01** | **0.57 ± 0.02** | **0.64 ± 0.01** |
| mKPGM | – | **1.72 ± 0.01** | **1.84 ± 0.01** | **4.19 ± 0.06** | **2.24 ± 0.01** |

Table 16: Generation quality in numerical characteristics on YouTube dataset.

| Method | Config | Clus ($\times 10^{-2}$) | | | Tri ($\times 10^{-4}$) | | |
|---|---|---|---|---|---|---|---|
| | | RMSE | MAE | Bias | RMSE | MAE | Bias |
| SyNG-D | 2 | 2.28 | 2.15 | -2.15 | 0.97 | 0.92 | 0.92 |
| | 3 | 1.69 | 1.46 | -1.41 | 0.89 | 0.84 | 0.84 |
| | 4 | **1.34** | **1.08** | **-0.95** | 0.83 | 0.78 | 0.78 |
| | 5 | 1.50 | 1.22 | -1.09 | 0.58 | 0.52 | 0.52 |
| | 6 | 1.42 | 1.19 | -1.01 | **0.41** | **0.35** | **0.35** |
| SyNG-D (MLP) | 2 | **1.12** | **0.88** | **-0.46** | 1.35 | 1.30 | 1.30 |
| | 3 | 1.87 | 1.63 | 1.54 | 1.47 | 1.43 | 1.43 |
| | 4 | 1.58 | 1.34 | 1.15 | 1.62 | 1.58 | 1.58 |
| | 5 | 2.98 | 2.83 | 2.80 | 1.40 | 1.36 | 1.36 |
| | 6 | 3.13 | 2.93 | 2.88 | **1.18** | **1.14** | **1.14** |
| SyNG-R | 2 | 2.23 | 2.08 | -2.07 | **1.15** | **1.11** | **1.11** |
| | 3 | **1.07** | **0.87** | **-0.40** | 1.38 | 1.34 | 1.34 |
| | 4 | 1.20 | 0.96 | 0.66 | 1.53 | 1.49 | 1.49 |
| | 5 | 1.57 | 1.30 | 1.19 | 1.61 | 1.57 | 1.57 |
| | 6 | 1.84 | 1.57 | 1.52 | 1.65 | 1.61 | 1.61 |
| VGAE | 2 | 11.80 | 11.80 | -11.80 | 0.81 | 0.81 | -0.81 |
| | 3 | 11.66 | 11.66 | -11.66 | 0.80 | 0.80 | -0.80 |
| | 4 | **8.00** | **8.00** | **-8.00** | **0.56** | **0.56** | **-0.56** |
| | 5 | 12.07 | 12.07 | -12.07 | 0.83 | 0.83 | -0.83 |
| | 6 | 9.24 | 9.24 | -9.24 | 0.65 | 0.65 | -0.65 |
| | 16 | 9.87 | 9.87 | -9.87 | 0.69 | 0.69 | -0.69 |
| GRAN | 128 | 11.34 | 11.32 | -11.32 | **0.85** | **0.56** | **0.54** |
| | 256 | **5.30** | **5.29** | **-5.29** | 15.05 | 14.95 | 14.95 |
| | 512 | 10.11 | 10.10 | -10.10 | 1.13 | 1.08 | 1.08 |
| EDGE | – | **14.45** | **13.81** | **-13.80** | **0.78** | **0.75** | **-0.50** |
| GraphMaker | – | **16.71** | **16.71** | **-16.71** | **1.12** | **1.12** | **-1.12** |
| E-R | – | **16.30** | **16.30** | **-16.30** | **1.05** | **1.05** | **-1.05** |
| BTER | – | **9.73** | **9.73** | **-9.73** | **0.02** | **0.02** | **-0.01** |
| mKPGM | – | **16.00** | **16.00** | **-16.00** | **1.03** | **1.03** | **-1.03** |

Table 17: Generation quality in degree centralities distribution similarity on DBLP dataset. W1 dist. and Energy dist. are scaled by $10^{-2}$; KS dist. and MMD are scaled by $10^{-1}$.

| Method | Config | W1 dist. | KS dist. | Energy dist. | MMD |
|---|---|---|---|---|---|
| SyNG-D | 2 | $0.19 \pm 0.08$ | $\mathbf{0.75 \pm 0.17}$ | $\mathbf{0.02 \pm 0.01}$ | $\mathbf{0.93 \pm 0.23}$ |
| | 3 | $\mathbf{0.18 \pm 0.06}$ | $0.91 \pm 0.16$ | $\mathbf{0.02 \pm 0.01}$ | $1.03 \pm 0.19$ |
| | 4 | $0.22 \pm 0.07$ | $1.22 \pm 0.14$ | $0.03 \pm 0.01$ | $1.29 \pm 0.17$ |
| | 5 | $0.33 \pm 0.09$ | $1.74 \pm 0.18$ | $0.05 \pm 0.02$ | $1.78 \pm 0.20$ |
| | 6 | $0.44 \pm 0.10$ | $2.19 \pm 0.18$ | $0.09 \pm 0.02$ | $2.22 \pm 0.20$ |
| SyNG-D (MLP) | 2 | $\mathbf{0.31 \pm 0.07}$ | $1.81 \pm 0.17$ | $\mathbf{0.04 \pm 0.01}$ | $1.50 \pm 0.16$ |
| | 3 | $0.42 \pm 0.06$ | $1.33 \pm 0.13$ | $0.07 \pm 0.02$ | $1.67 \pm 0.18$ |
| | 4 | $0.47 \pm 0.15$ | $\mathbf{1.19 \pm 0.19}$ | $0.06 \pm 0.03$ | $\mathbf{1.40 \pm 0.25}$ |
| | 5 | $0.47 \pm 0.10$ | $1.36 \pm 0.14$ | $0.07 \pm 0.02$ | $1.72 \pm 0.16$ |
| | 6 | $0.35 \pm 0.06$ | $1.53 \pm 0.18$ | $0.06 \pm 0.01$ | $1.73 \pm 0.19$ |
| SyNG-R | 2 | $\mathbf{0.15 \pm 0.07}$ | $\mathbf{0.73 \pm 0.18}$ | $\mathbf{0.01 \pm 0.01}$ | $\mathbf{0.75 \pm 0.28}$ |
| | 3 | $\mathbf{0.15 \pm 0.07}$ | $\mathbf{0.73 \pm 0.18}$ | $\mathbf{0.01 \pm 0.01}$ | $\mathbf{0.75 \pm 0.28}$ |
| | 4 | $\mathbf{0.15 \pm 0.07}$ | $\mathbf{0.73 \pm 0.18}$ | $\mathbf{0.01 \pm 0.01}$ | $0.77 \pm 0.28$ |
| | 5 | $0.16 \pm 0.07$ | $0.74 \pm 0.18$ | $\mathbf{0.01 \pm 0.01}$ | $0.78 \pm 0.27$ |
| | 6 | $0.16 \pm 0.07$ | $0.75 \pm 0.17$ | $\mathbf{0.01 \pm 0.01}$ | $0.80 \pm 0.27$ |
| VGAE | 2 | $0.49 \pm 0.01$ | $\mathbf{2.33 \pm 0.08}$ | $0.10 \pm 0.00$ | $2.84 \pm 0.10$ |
| | 3 | $\mathbf{0.29 \pm 0.00}$ | $2.65 \pm 0.07$ | $\mathbf{0.09 \pm 0.00}$ | $\mathbf{2.80 \pm 0.06}$ |
| | 4 | $0.35 \pm 0.01$ | $3.20 \pm 0.08$ | $0.12 \pm 0.00$ | $3.28 \pm 0.06$ |
| | 5 | $0.35 \pm 0.01$ | $3.16 \pm 0.08$ | $0.12 \pm 0.00$ | $3.23 \pm 0.07$ |
| | 6 | $0.32 \pm 0.00$ | $2.96 \pm 0.07$ | $0.11 \pm 0.00$ | $3.02 \pm 0.05$ |
| | 16 | $0.32 \pm 0.00$ | $2.96 \pm 0.08$ | $0.10 \pm 0.00$ | $2.96 \pm 0.06$ |
| GRAN | 128 | $1.48 \pm 0.25$ | $3.79 \pm 0.88$ | $0.41 \pm 0.14$ | $5.13 \pm 0.96$ |
| | 256 | $1.25 \pm 0.03$ | $4.61 \pm 0.12$ | $0.54 \pm 0.02$ | $6.12 \pm 0.14$ |
| | 512 | $\mathbf{1.06 \pm 0.01}$ | $\mathbf{2.21 \pm 0.08}$ | $\mathbf{0.31 \pm 0.01}$ | $\mathbf{2.97 \pm 0.09}$ |
| EDGE | – | $\mathbf{0.23 \pm 0.12}$ | $\mathbf{0.79 \pm 0.11}$ | $\mathbf{0.02 \pm 0.02}$ | $\mathbf{0.99 \pm 0.23}$ |
| GraphMaker | – | $\mathbf{1.37 \pm 0.01}$ | $\mathbf{5.56 \pm 0.07}$ | $\mathbf{0.68 \pm 0.01}$ | $\mathbf{7.51 \pm 0.07}$ |
| E-R | – | $\mathbf{1.57 \pm 0.01}$ | $\mathbf{6.63 \pm 0.07}$ | $\mathbf{1.02 \pm 0.02}$ | $\mathbf{8.72 \pm 0.06}$ |
| BTER | – | $\mathbf{0.08 \pm 0.01}$ | $\mathbf{0.56 \pm 0.05}$ | $\mathbf{0.00 \pm 0.00}$ | $\mathbf{0.46 \pm 0.05}$ |
| mKPGM | – | $\mathbf{1.13 \pm 0.01}$ | $\mathbf{3.25 \pm 0.08}$ | $\mathbf{0.37 \pm 0.00}$ | $\mathbf{4.57 \pm 0.08}$ |

Table 18: Generation quality in eigenvalue distribution similarity on DBLP dataset. W1, KS, and MMD are scaled by $10^{-1}$; Energy dist. is scaled by $10^{-2}$.

| Method | Config | W1 dist. | KS dist. | Energy dist. | MMD |
|---|---|---|---|---|---|
| SyNG-D | 2 | 3.02 ± 0.32 | **0.81 ± 0.06** | 2.35 ± 0.43 | 0.88 ± 0.06 |
| | 3 | 2.13 ± 0.28 | 0.91 ± 0.07 | **1.57 ± 0.21** | **0.77 ± 0.05** |
| | 4 | **1.73 ± 0.17** | 1.06 ± 0.07 | 1.65 ± 0.20 | **0.77 ± 0.05** |
| | 5 | 2.56 ± 0.43 | 1.32 ± 0.08 | 2.99 ± 0.57 | 0.96 ± 0.08 |
| | 6 | 3.80 ± 0.46 | 1.63 ± 0.08 | 5.62 ± 0.89 | 1.28 ± 0.08 |
| SyNG-D (MLP) | 2 | **2.85 ± 0.34** | 1.10 ± 0.07 | **2.37 ± 0.31** | **0.93 ± 0.05** |
| | 3 | 6.09 ± 0.47 | 1.10 ± 0.08 | 8.94 ± 1.27 | 1.34 ± 0.07 |
| | 4 | 4.46 ± 0.41 | **0.83 ± 0.05** | 5.05 ± 0.87 | 1.12 ± 0.07 |
| | 5 | 5.75 ± 0.46 | 0.97 ± 0.07 | 7.42 ± 1.06 | 1.32 ± 0.06 |
| | 6 | 5.92 ± 0.45 | 1.03 ± 0.08 | 8.06 ± 1.13 | 1.31 ± 0.06 |
| SyNG-R | 2 | 2.89 ± 0.33 | 0.63 ± 0.05 | 2.02 ± 0.50 | 0.80 ± 0.07 |
| | 3 | 2.25 ± 0.33 | 0.58 ± 0.05 | 1.35 ± 0.39 | 0.65 ± 0.07 |
| | 4 | 1.70 ± 0.29 | 0.50 ± 0.06 | 0.83 ± 0.25 | 0.49 ± 0.07 |
| | 5 | 1.31 ± 0.30 | 0.46 ± 0.05 | 0.52 ± 0.18 | 0.37 ± 0.07 |
| | 6 | **1.19 ± 0.32** | **0.41 ± 0.05** | **0.40 ± 0.15** | **0.27 ± 0.07** |
| VGAE | 2 | 12.38 ± 0.16 | 2.35 ± 0.03 | 37.67 ± 1.01 | 2.62 ± 0.03 |
| | 3 | 4.04 ± 0.06 | 0.98 ± 0.03 | 3.47 ± 0.17 | 1.20 ± 0.04 |
| | 4 | 3.83 ± 0.10 | 0.67 ± 0.03 | 2.81 ± 0.18 | 0.68 ± 0.03 |
| | 5 | 3.76 ± 0.12 | 0.65 ± 0.03 | 2.69 ± 0.21 | 0.65 ± 0.03 |
| | 6 | **3.51 ± 0.08** | **0.61 ± 0.02** | **2.30 ± 0.13** | **0.64 ± 0.03** |
| | 16 | 3.57 ± 0.09 | 0.63 ± 0.03 | 2.48 ± 0.16 | 0.65 ± 0.03 |
| GRAN | 128 | **6.40 ± 1.61** | 1.61 ± 0.20 | 14.92 ± 6.20 | 1.87 ± 0.22 |
| | 256 | 17.55 ± 0.48 | 3.06 ± 0.05 | 75.24 ± 3.29 | 3.54 ± 0.06 |
| | 512 | 6.95 ± 0.24 | **1.49 ± 0.05** | **11.81 ± 1.00** | **1.76 ± 0.07** |
| EDGE | – | **5.50 ± 1.34** | **1.22 ± 0.25** | **9.72 ± 4.23** | **1.39 ± 0.30** |
| GraphMaker | – | **18.39 ± 0.14** | **3.32 ± 0.02** | **85.40 ± 1.14** | **3.93 ± 0.02** |
| E-R | – | **22.55 ± 0.14** | **3.77 ± 0.02** | **121.43 ± 1.26** | **4.41 ± 0.02** |
| BTER | – | **9.44 ± 0.11** | **1.14 ± 0.02** | **14.67 ± 0.35** | **1.48 ± 0.03** |
| mKPGM | – | **13.13 ± 0.13** | **2.52 ± 0.02** | **43.78 ± 0.90** | **3.00 ± 0.03** |

Table 19: Generation quality in numerical characteristics on DBLP dataset.

| Method | Config | Clus ($\times 10^{-2}$) | | | Tri ($\times 10^{-4}$) | | |
| --- | --- | --- | --- | --- | --- | --- | --- |
| | | RMSE | MAE | Bias | RMSE | MAE | Bias |
| SyNG-D | 2 | 6.22 | 5.87 | -5.87 | **1.58** | **1.26** | **0.46** |
| | 3 | 5.34 | **4.94** | **-4.94** | 1.61 | 1.37 | -0.87 |
| | 4 | 5.51 | 5.23 | -5.23 | 1.81 | 1.57 | -1.39 |
| | 5 | **5.27** | 4.95 | -4.95 | 2.34 | 2.13 | -2.07 |
| | 6 | 5.71 | 5.37 | -5.37 | 3.01 | 2.86 | -2.86 |
| SyNG-D (MLP) | 2 | 14.00 | 13.61 | -13.61 | 2.91 | 2.67 | -2.62 |
| | 3 | 24.21 | 23.91 | -23.91 | 3.74 | 3.64 | -3.64 |
| | 4 | **10.12** | **9.92** | **-9.92** | 3.63 | 3.20 | 3.13 |
| | 5 | 14.38 | 14.14 | -14.14 | 2.29 | 1.81 | 1.47 |
| | 6 | 17.15 | 16.94 | -16.94 | **1.80** | **1.54** | **-1.37** |
| SyNG-R | 2 | 3.78 | 3.28 | -3.25 | **1.43** | **1.12** | **0.11** |
| | 3 | 3.22 | 2.69 | -2.64 | 1.44 | **1.12** | 0.19 |
| | 4 | 2.63 | 2.10 | -1.96 | 1.45 | **1.12** | 0.25 |
| | 5 | 2.27 | 1.76 | -1.51 | 1.46 | 1.13 | 0.28 |
| | 6 | **2.14** | **1.66** | **-1.32** | 1.46 | 1.13 | 0.31 |
| VGAE | 2 | 63.28 | 63.28 | -63.28 | 6.65 | 6.65 | -6.65 |
| | 3 | **1.36** | **1.34** | **-1.34** | 0.58 | 0.58 | -0.58 |
| | 4 | 2.94 | 2.93 | 2.93 | 0.28 | 0.28 | 0.28 |
| | 5 | 3.01 | 3.00 | 3.00 | 0.26 | 0.26 | 0.26 |
| | 6 | 3.32 | 3.31 | 3.31 | 0.11 | 0.10 | 0.10 |
| | 16 | 3.10 | 3.10 | 3.10 | **0.05** | **0.05** | **0.05** |
| GRAN | 128 | **87.63** | **87.63** | **-87.63** | **6.78** | **6.73** | **-6.73** |
| | 256 | 88.93 | 88.93 | -88.93 | 7.93 | 7.93 | -7.93 |
| | 512 | 89.21 | 89.21 | -89.21 | 7.99 | 7.99 | -7.99 |
| EDGE | – | **13.27** | **10.86** | **-10.86** | **2.20** | **1.77** | **-1.77** |
| GraphMaker | – | **89.78** | **89.78** | **-89.78** | **7.98** | **7.98** | **-7.98** |
| E-R | – | **89.43** | **89.43** | **-89.43** | **7.96** | **7.96** | **-7.96** |
| BTER | – | **67.75** | **67.75** | **-67.75** | **6.09** | **6.09** | **-6.09** |
| mKPGM | – | **89.97** | **89.97** | **-89.97** | **8.00** | **8.00** | **-8.00** |

Table 20: Generation quality in degree centralities distribution similarity on Yelp dataset. All metrics scaled by $10^{-2}$.

| Method | Config | W1 dist. | KS dist. | Energy dist. | MMD |
|---|---|---|---|---|---|
| SyNG-D | 2 | **0.15 ± 0.05** | **2.49 ± 0.74** | **0.54 ± 0.19** | **1.28 ± 0.87** |
| | 3 | 0.21 ± 0.10 | 3.55 ± 1.26 | 0.77 ± 0.34 | 2.36 ± 1.43 |
| | 4 | 0.24 ± 0.10 | 4.54 ± 1.12 | 0.98 ± 0.33 | 3.46 ± 1.09 |
| | 5 | 0.36 ± 0.11 | 6.02 ± 1.16 | 1.42 ± 0.37 | 5.03 ± 1.15 |
| | 6 | 0.44 ± 0.13 | 7.72 ± 1.42 | 1.81 ± 0.44 | 6.72 ± 1.36 |
| SyNG-D (MLP) | 2 | 0.32 ± 0.12 | **4.68 ± 1.36** | 1.18 ± 0.43 | **3.26 ± 1.42** |
| | 3 | 0.39 ± 0.12 | 5.40 ± 1.30 | 1.38 ± 0.40 | 4.32 ± 1.18 |
| | 4 | 0.69 ± 0.15 | 8.56 ± 1.33 | 2.49 ± 0.46 | 7.82 ± 1.23 |
| | 5 | **0.23 ± 0.08** | 5.22 ± 1.03 | **1.01 ± 0.26** | 4.17 ± 0.79 |
| | 6 | 0.30 ± 0.09 | 4.80 ± 1.06 | 1.19 ± 0.33 | 4.51 ± 0.96 |
| SyNG-R | 2 | **0.13 ± 0.06** | 2.02 ± 0.80 | **0.47 ± 0.22** | 0.65 ± 0.94 |
| | 3 | **0.13 ± 0.06** | **1.98 ± 0.78** | **0.47 ± 0.22** | **0.60 ± 0.90** |
| | 4 | **0.13 ± 0.06** | 1.99 ± 0.82 | **0.47 ± 0.23** | 0.62 ± 0.92 |
| | 5 | 0.14 ± 0.06 | 2.05 ± 0.80 | 0.48 ± 0.22 | 0.67 ± 0.94 |
| | 6 | **0.13 ± 0.06** | **1.98 ± 0.78** | **0.47 ± 0.22** | 0.62 ± 0.91 |
| VGAE | 2 | **1.65 ± 0.00** | **22.12 ± 0.23** | **5.79 ± 0.02** | **32.23 ± 0.16** |
| | 3 | 1.70 ± 0.00 | 23.14 ± 0.22 | 5.98 ± 0.02 | 33.32 ± 0.19 |
| | 4 | 1.71 ± 0.00 | 23.50 ± 0.21 | 6.03 ± 0.02 | 33.67 ± 0.17 |
| | 5 | 1.69 ± 0.00 | 22.81 ± 0.22 | 5.97 ± 0.02 | 33.38 ± 0.16 |
| | 6 | 1.79 ± 0.00 | 24.47 ± 0.20 | 6.34 ± 0.02 | 35.49 ± 0.14 |
| | 16 | 1.73 ± 0.00 | 23.37 ± 0.21 | 6.09 ± 0.02 | 33.74 ± 0.18 |
| GraphMaker | – | **2.59 ± 0.00** | **41.44 ± 0.20** | **10.33 ± 0.01** | **69.26 ± 0.15** |
| E-R | – | **2.81 ± 0.00** | **57.17 ± 0.18** | **12.09 ± 0.03** | **77.36 ± 0.13** |
| BTER | – | **0.04 ± 0.00** | **0.97 ± 0.17** | **0.14 ± 0.01** | **0.00 ± 0.00** |
| mKPGM | – | **3.10 ± 0.00** | **49.19 ± 0.18** | **13.07 ± 0.02** | **49.97 ± 0.14** |

Table 21: Generation quality in eigenvalue distribution similarity on Yelp dataset. W1 unchanged; KS, Energy, and MMD scaled by $10^{-1}$.

| Method | Config | W1 dist. | KS dist. | Energy dist. | MMD |
|---|---|---|---|---|---|
| SyNG-D | 2 | $1.29 \pm 0.09$ | $0.53 \pm 0.04$ | $2.93 \pm 0.24$ | $0.61 \pm 0.05$ |
| | 3 | $1.09 \pm 0.10$ | $0.44 \pm 0.05$ | $2.43 \pm 0.27$ | $0.51 \pm 0.05$ |
| | 4 | $0.92 \pm 0.08$ | $0.38 \pm 0.04$ | $2.03 \pm 0.23$ | $0.43 \pm 0.05$ |
| | 5 | $0.77 \pm 0.08$ | $0.30 \pm 0.04$ | $1.62 \pm 0.22$ | $0.34 \pm 0.05$ |
| | 6 | $\mathbf{0.64 \pm 0.09}$ | $\mathbf{0.23 \pm 0.05}$ | $\mathbf{1.28 \pm 0.25}$ | $\mathbf{0.25 \pm 0.06}$ |
| SyNG-D (MLP) | 2 | $2.07 \pm 0.10$ | $0.84 \pm 0.04$ | $4.88 \pm 0.25$ | $0.96 \pm 0.05$ |
| | 3 | $1.88 \pm 0.10$ | $0.75 \pm 0.04$ | $4.43 \pm 0.25$ | $0.87 \pm 0.04$ |
| | 4 | $2.00 \pm 0.11$ | $0.82 \pm 0.04$ | $4.79 \pm 0.26$ | $0.94 \pm 0.05$ |
| | 5 | $\mathbf{1.21 \pm 0.10}$ | $\mathbf{0.51 \pm 0.04}$ | $\mathbf{2.83 \pm 0.25}$ | $\mathbf{0.57 \pm 0.05}$ |
| | 6 | $1.55 \pm 0.09$ | $0.66 \pm 0.04$ | $3.73 \pm 0.23$ | $0.75 \pm 0.04$ |
| SyNG-R | 2 | $1.32 \pm 0.09$ | $0.55 \pm 0.04$ | $3.04 \pm 0.25$ | $0.64 \pm 0.05$ |
| | 3 | $1.22 \pm 0.09$ | $0.51 \pm 0.04$ | $2.82 \pm 0.25$ | $0.59 \pm 0.05$ |
| | 4 | $1.10 \pm 0.09$ | $0.47 \pm 0.04$ | $2.55 \pm 0.24$ | $0.54 \pm 0.05$ |
| | 5 | $1.00 \pm 0.09$ | $0.44 \pm 0.04$ | $2.33 \pm 0.24$ | $0.49 \pm 0.05$ |
| | 6 | $\mathbf{0.94 \pm 0.09}$ | $\mathbf{0.42 \pm 0.04}$ | $\mathbf{2.21 \pm 0.23}$ | $\mathbf{0.47 \pm 0.05}$ |
| VGAE | 2 | $2.44 \pm 0.01$ | $\mathbf{1.40 \pm 0.00}$ | $6.51 \pm 0.02$ | $\mathbf{1.73 \pm 0.00}$ |
| | 3 | $2.46 \pm 0.01$ | $1.41 \pm 0.00$ | $6.56 \pm 0.02$ | $1.76 \pm 0.00$ |
| | 4 | $2.47 \pm 0.01$ | $1.41 \pm 0.00$ | $6.58 \pm 0.02$ | $1.76 \pm 0.00$ |
| | 5 | $2.48 \pm 0.01$ | $1.42 \pm 0.00$ | $6.60 \pm 0.02$ | $1.75 \pm 0.00$ |
| | 6 | $2.51 \pm 0.01$ | $1.44 \pm 0.00$ | $6.71 \pm 0.02$ | $1.79 \pm 0.00$ |
| | 16 | $\mathbf{2.42 \pm 0.01}$ | $\mathbf{1.40 \pm 0.00}$ | $\mathbf{6.49 \pm 0.02}$ | $1.76 \pm 0.00$ |
| GraphMaker | – | $\mathbf{2.94 \pm 0.01}$ | $\mathbf{1.60 \pm 0.00}$ | $\mathbf{7.63 \pm 0.02}$ | $\mathbf{2.07 \pm 0.00}$ |
| E-R | – | $\mathbf{3.83 \pm 0.01}$ | $\mathbf{2.00 \pm 0.00}$ | $\mathbf{10.22 \pm 0.02}$ | $\mathbf{2.45 \pm 0.00}$ |
| BTER | – | $\mathbf{1.75 \pm 0.01}$ | $\mathbf{0.70 \pm 0.00}$ | $\mathbf{3.96 \pm 0.02}$ | $\mathbf{0.83 \pm 0.00}$ |
| mKPGM | – | $\mathbf{2.06 \pm 0.01}$ | $\mathbf{0.91 \pm 0.00}$ | $\mathbf{4.77 \pm 0.02}$ | $\mathbf{0.93 \pm 0.00}$ |

Table 22: Generation quality in numerical characteristics on Yelp dataset.

| Method | Config | Clus ($\times 10^{-2}$) | | | Tri ($\times 10^{-4}$) | | |
|---|---|---|---|---|---|---|---|
| | | RMSE | MAE | Bias | RMSE | MAE | Bias |
| SyNG-D | 2 | 2.56 | 2.52 | -2.52 | **1.24** | **1.08** | **-1.05** |
| | 3 | 2.65 | 2.61 | -2.61 | 1.71 | 1.57 | -1.56 |
| | 4 | 2.01 | 1.96 | -1.96 | 1.59 | 1.47 | -1.46 |
| | 5 | 1.83 | 1.77 | -1.77 | 1.82 | 1.72 | -1.71 |
| | 6 | **1.77** | **1.70** | **-1.70** | 2.00 | 1.89 | -1.88 |
| SyNG-D (MLP) | 2 | 2.35 | 2.31 | -2.31 | **0.74** | **0.61** | **-0.22** |
| | 3 | 1.05 | 0.96 | -0.96 | 1.40 | 1.18 | 1.15 |
| | 4 | 0.75 | 0.63 | 0.58 | 3.33 | 3.18 | 3.18 |
| | 5 | 0.81 | 0.71 | -0.68 | 0.98 | 0.84 | -0.72 |
| | 6 | **0.65** | **0.53** | **0.47** | 1.44 | 1.26 | 1.23 |
| SyNG-R | 2 | 2.79 | 2.76 | -2.76 | 1.49 | 1.35 | -1.34 |
| | 3 | 2.40 | 2.36 | -2.36 | 1.33 | 1.17 | -1.15 |
| | 4 | 1.56 | 1.50 | -1.50 | 1.01 | 0.85 | -0.73 |
| | 5 | 0.99 | 0.90 | -0.89 | 0.83 | 0.69 | -0.43 |
| | 6 | **0.76** | **0.65** | **-0.61** | **0.78** | **0.64** | **-0.30** |
| VGAE | 2 | 10.36 | 10.36 | -10.36 | **7.36** | **7.36** | **-7.36** |
| | 3 | 10.55 | 10.55 | -10.55 | 7.45 | 7.45 | -7.45 |
| | 4 | 10.56 | 10.56 | -10.56 | 7.46 | 7.46 | -7.46 |
| | 5 | 10.77 | 10.77 | -10.77 | 7.49 | 7.49 | -7.49 |
| | 6 | 11.26 | 11.26 | -11.26 | 7.67 | 7.67 | -7.67 |
| | 16 | **10.25** | **10.25** | **-10.25** | 7.41 | 7.41 | -7.41 |
| GraphMaker | – | **15.52** | **15.52** | **-15.52** | **8.82** | **8.82** | **-8.82** |
| E-R | – | **14.48** | **14.48** | **-14.48** | **8.12** | **8.12** | **-8.12** |
| BTER | – | **4.45** | **4.45** | **-4.45** | **2.27** | **2.27** | **-2.27** |
| mKPGM | – | **15.98** | **15.98** | **-15.98** | **9.35** | **9.35** | **-9.35** |

Table 23: Generation quality in degree centralities distribution similarity on PolBlogs dataset. All metrics scaled by $10^{-2}$.

| Method | Config | W1 dist. | KS dist. | Energy dist. | MMD |
|---|---|---|---|---|---|
| SyNG-D | 2 | **0.18 ± 0.08** | **4.53 ± 0.97** | **0.01 ± 0.01** | **0.97 ± 1.44** |
| | 3 | 0.22 ± 0.11 | 5.02 ± 1.34 | **0.01 ± 0.01** | 1.95 ± 2.15 |
| | 4 | 0.28 ± 0.13 | 5.74 ± 1.64 | 0.02 ± 0.02 | 3.18 ± 2.50 |
| | 5 | 0.44 ± 0.14 | 8.29 ± 2.04 | 0.04 ± 0.02 | 6.49 ± 2.44 |
| | 6 | 0.52 ± 0.15 | 9.98 ± 2.11 | 0.06 ± 0.03 | 8.31 ± 2.50 |
| SyNG-D (MLP) | 2 | 0.20 ± 0.07 | 9.59 ± 1.21 | **0.01 ± 0.01** | 1.47 ± 1.71 |
| | 3 | 0.24 ± 0.10 | 9.81 ± 1.09 | **0.01 ± 0.01** | 3.05 ± 2.25 |
| | 4 | 0.33 ± 0.15 | 10.06 ± 1.49 | 0.02 ± 0.02 | 3.20 ± 2.26 |
| | 5 | 0.23 ± 0.07 | **8.99 ± 0.93** | **0.01 ± 0.01** | 2.79 ± 1.77 |
| | 6 | **0.19 ± 0.08** | 9.16 ± 1.35 | **0.01 ± 0.01** | **1.41 ± 1.73** |
| SyNG-R | 2 | **0.18 ± 0.10** | **4.47 ± 0.92** | **0.01 ± 0.01** | **0.92 ± 1.45** |
| | 3 | **0.18 ± 0.10** | 4.49 ± 1.11 | **0.01 ± 0.01** | 0.93 ± 1.50 |
| | 4 | **0.18 ± 0.10** | 4.57 ± 1.23 | **0.01 ± 0.01** | 0.94 ± 1.50 |
| | 5 | **0.18 ± 0.10** | 4.68 ± 1.30 | **0.01 ± 0.01** | 1.06 ± 1.62 |
| | 6 | **0.18 ± 0.10** | 4.88 ± 1.49 | **0.01 ± 0.01** | 1.12 ± 1.67 |
| VGAE | 2 | 0.93 ± 0.01 | 35.38 ± 0.49 | 0.24 ± 0.01 | 35.42 ± 0.61 |
| | 3 | 0.97 ± 0.01 | 36.24 ± 0.53 | 0.26 ± 0.01 | 37.16 ± 0.51 |
| | 4 | 0.94 ± 0.01 | 35.65 ± 0.52 | 0.24 ± 0.01 | 36.02 ± 0.60 |
| | 5 | **0.92 ± 0.01** | **35.31 ± 0.51** | **0.23 ± 0.01** | **35.33 ± 0.64** |
| | 6 | 0.97 ± 0.01 | 36.35 ± 0.53 | 0.26 ± 0.01 | 37.42 ± 0.57 |
| | 16 | 0.98 ± 0.01 | 36.56 ± 0.49 | 0.26 ± 0.01 | 37.82 ± 0.56 |
| GRAN | 128 | **0.56 ± 0.19** | **15.88 ± 2.70** | **0.08 ± 0.05** | **13.86 ± 2.01** |
| | 256 | 2.25 ± 0.31 | 35.58 ± 3.15 | 0.64 ± 0.15 | 33.78 ± 3.61 |
| | 512 | 9.30 ± 0.67 | 63.37 ± 2.38 | 6.36 ± 0.72 | 67.41 ± 2.69 |
| EDGE | – | **0.06 ± 0.00** | **4.91 ± 0.44** | **0.00 ± 0.00** | **0.00 ± 0.00** |
| GraphMaker | – | 1.72 ± 0.01 | 49.48 ± 0.60 | 0.82 ± 0.01 | 74.15 ± 0.75 |
| E-R | – | 1.83 ± 0.01 | 56.70 ± 0.56 | 1.04 ± 0.01 | 79.24 ± 0.59 |
| BTER | – | **0.06 ± 0.01** | 5.69 ± 0.62 | **0.00 ± 0.00** | **0.00 ± 0.00** |
| mKPGM | – | 1.57 ± 0.01 | 33.09 ± 0.37 | 0.60 ± 0.01 | 50.67 ± 0.90 |

Table 24: Generation quality in eigenvalue distribution similarity on PolBlogs dataset. W1 is unscaled; KS, Energy, and MMD are scaled by $10^{-2}$.

| Method | Config | W1 dist. | KS dist. | Energy dist. | MMD |
|---|---|---|---|---|---|
| SyNG-D | 2 | $0.21 \pm 0.06$ | $4.82 \pm 0.94$ | $0.92 \pm 0.51$ | $2.58 \pm 1.37$ |
| | 3 | $\mathbf{0.20 \pm 0.04}$ | $5.02 \pm 1.03$ | $0.90 \pm 0.38$ | $3.15 \pm 1.16$ |
| | 4 | $\mathbf{0.20 \pm 0.05}$ | $4.41 \pm 0.96$ | $\mathbf{0.74 \pm 0.32}$ | $2.52 \pm 1.22$ |
| | 5 | $0.25 \pm 0.07$ | $\mathbf{3.22 \pm 0.77}$ | $0.80 \pm 0.46$ | $1.51 \pm 1.18$ |
| | 6 | $0.30 \pm 0.09$ | $\mathbf{3.32 \pm 0.89}$ | $1.19 \pm 0.72$ | $\mathbf{1.43 \pm 1.36}$ |
| SyNG-D (MLP) | 2 | $0.46 \pm 0.10$ | $7.00 \pm 0.89$ | $3.36 \pm 1.34$ | $5.54 \pm 1.05$ |
| | 3 | $\mathbf{0.37 \pm 0.09}$ | $7.01 \pm 0.76$ | $\mathbf{2.51 \pm 1.04}$ | $\mathbf{5.27 \pm 0.90}$ |
| | 4 | $0.65 \pm 0.11$ | $8.90 \pm 0.90$ | $6.88 \pm 1.98$ | $8.17 \pm 1.11$ |
| | 5 | $0.38 \pm 0.10$ | $7.00 \pm 0.89$ | $2.81 \pm 1.18$ | $5.57 \pm 1.08$ |
| | 6 | $0.39 \pm 0.10$ | $\mathbf{6.74 \pm 0.92}$ | $2.80 \pm 1.18$ | $5.35 \pm 1.12$ |
| SyNG-R | 2 | $0.24 \pm 0.08$ | $\mathbf{5.04 \pm 1.10}$ | $\mathbf{1.24 \pm 0.82}$ | $\mathbf{3.01 \pm 1.59}$ |
| | 3 | $0.23 \pm 0.07$ | $5.45 \pm 1.02$ | $1.33 \pm 0.76$ | $3.64 \pm 1.34$ |
| | 4 | $0.23 \pm 0.07$ | $5.58 \pm 1.06$ | $1.35 \pm 0.79$ | $3.86 \pm 1.32$ |
| | 5 | $\mathbf{0.22 \pm 0.07}$ | $5.68 \pm 0.99$ | $1.34 \pm 0.77$ | $3.98 \pm 1.20$ |
| | 6 | $\mathbf{0.22 \pm 0.06}$ | $5.69 \pm 0.98$ | $1.36 \pm 0.73$ | $4.15 \pm 1.17$ |
| VGAE | 2 | $\mathbf{1.23 \pm 0.01}$ | $27.33 \pm 0.18$ | $42.20 \pm 0.72$ | $31.11 \pm 0.20$ |
| | 3 | $1.27 \pm 0.01$ | $27.67 \pm 0.18$ | $44.05 \pm 0.82$ | $31.57 \pm 0.21$ |
| | 4 | $1.24 \pm 0.01$ | $27.41 \pm 0.18$ | $42.65 \pm 0.74$ | $31.23 \pm 0.20$ |
| | 5 | $\mathbf{1.23 \pm 0.01}$ | $\mathbf{27.27 \pm 0.18}$ | $\mathbf{41.88 \pm 0.75}$ | $\mathbf{31.10 \pm 0.22}$ |
| | 6 | $1.27 \pm 0.01$ | $27.72 \pm 0.18$ | $44.16 \pm 0.74$ | $31.62 \pm 0.19$ |
| | 16 | $1.28 \pm 0.01$ | $27.81 \pm 0.18$ | $44.70 \pm 0.77$ | $31.80 \pm 0.20$ |
| GRAN | 128 | $\mathbf{0.48 \pm 0.18}$ | $10.91 \pm 1.15$ | $\mathbf{4.63 \pm 2.88}$ | $\mathbf{9.04 \pm 0.98}$ |
| | 256 | $1.23 \pm 0.16$ | $\mathbf{10.89 \pm 1.18}$ | $17.03 \pm 4.31$ | $9.39 \pm 1.26$ |
| | 512 | $3.52 \pm 0.19$ | $24.55 \pm 1.24$ | $122.93 \pm 12.78$ | $26.84 \pm 1.41$ |
| EDGE | – | $\mathbf{0.28 \pm 0.04}$ | $\mathbf{8.21 \pm 1.17}$ | $\mathbf{1.90 \pm 0.59}$ | $\mathbf{4.69 \pm 1.26}$ |
| GraphMaker | – | $\mathbf{1.78 \pm 0.01}$ | $\mathbf{32.20 \pm 0.15}$ | $\mathbf{72.49 \pm 0.85}$ | $\mathbf{38.28 \pm 0.13}$ |
| E-R | – | $\mathbf{2.03 \pm 0.01}$ | $\mathbf{34.42 \pm 0.12}$ | $\mathbf{93.20 \pm 0.92}$ | $\mathbf{40.10 \pm 0.11}$ |
| BTER | – | $\mathbf{0.41 \pm 0.02}$ | $\mathbf{4.53 \pm 0.40}$ | $\mathbf{1.87 \pm 0.17}$ | $\mathbf{3.04 \pm 0.34}$ |
| mKPGM | – | $\mathbf{1.31 \pm 0.01}$ | $\mathbf{23.36 \pm 0.22}$ | $\mathbf{31.95 \pm 0.46}$ | $\mathbf{31.15 \pm 0.22}$ |

Table 25: Generation quality in numerical characteristics on PolBlogs dataset.

| Method | Config | Clus ($\times 10^{-2}$) | | | Tri ($\times 10^{-4}$) | | |
|---|---|---|---|---|---|---|---|
| | | RMSE | MAE | Bias | RMSE | MAE | Bias |
| SyNG-D | 2 | 1.90 | 1.51 | 1.15 | 0.71 | **0.55** | **0.13** |
| | 3 | **1.56** | **1.23** | 0.54 | **0.68** | 0.56 | -0.33 |
| | 4 | 1.85 | 1.45 | 0.90 | 0.78 | 0.66 | -0.43 |
| | 5 | 1.66 | 1.33 | **0.49** | 1.09 | 1.00 | -0.98 |
| | 6 | 1.91 | 1.49 | 0.93 | 1.19 | 1.09 | -1.07 |
| SyNG-D (MLP) | 2 | 3.58 | 3.26 | 3.23 | 0.91 | 0.69 | 0.56 |
| | 3 | **2.23** | **1.88** | **1.73** | **0.59** | **0.48** | **-0.16** |
| | 4 | 5.69 | 5.45 | 5.45 | 1.80 | 1.61 | 1.60 |
| | 5 | 5.18 | 4.77 | 4.74 | 1.15 | 0.90 | 0.77 |
| | 6 | 3.15 | 2.80 | 2.72 | 0.84 | 0.65 | 0.50 |
| SyNG-B | 2 | **2.45** | **2.00** | **1.83** | **0.83** | **0.62** | **0.39** |
| | 3 | 2.68 | 2.23 | 2.12 | 0.85 | 0.64 | 0.44 |
| | 4 | 2.88 | 2.44 | 2.37 | 0.87 | 0.66 | 0.49 |
| | 5 | 2.91 | 2.50 | 2.44 | 0.89 | 0.68 | 0.51 |
| | 6 | 3.13 | 2.74 | 2.69 | 0.92 | 0.71 | 0.55 |
| VGAE | 2 | 3.72 | 3.71 | -3.71 | 2.10 | 2.10 | **-2.07** |
| | 3 | 4.49 | 4.49 | -4.49 | 2.20 | 2.20 | -2.20 |
| | 4 | 4.03 | 4.02 | -4.02 | 2.14 | 2.14 | -2.14 |
| | 5 | **3.26** | **3.25** | **-3.25** | **2.07** | **2.07** | **-2.07** |
| | 6 | 4.87 | 4.87 | -4.87 | 2.22 | 2.22 | -2.22 |
| | 16 | 4.54 | 4.54 | -4.54 | 2.22 | 2.22 | -2.22 |
| GRAN | 128 | 11.42 | 11.38 | -11.38 | **1.57** | **1.49** | **-1.41** |
| | 256 | 10.82 | 10.78 | -10.78 | 4.42 | 4.22 | 4.22 |
| | 512 | **0.76** | **0.60** | **0.03** | 64.94 | 64.45 | 64.45 |
| EDGE | – | **5.69** | **5.43** | **-5.43** | **0.79** | **0.75** | **-0.75** |
| GraphMaker | – | **20.75** | **20.75** | **-20.75** | **3.27** | **3.27** | **-3.27** |
| E-R | – | **20.36** | **20.36** | **-20.36** | **3.22** | **3.22** | **-3.22** |
| BTER | – | **5.27** | **5.27** | **-5.27** | **0.88** | **0.88** | **-0.88** |
| mKPGM | – | **21.43** | **21.43** | **-21.43** | **3.32** | **3.32** | **-3.32** |

Table 26: 4-node graphlet frequency distance (GFD) results on the YouTube, DBLP, PolBlogs, and Yelp datasets. We report both L1 and L2 distances as measures of similarity, and highlight the best result for each method in bold. All entries are scaled by $10^{-1}$.

| Method | Config | $\text{GFD}_{L1} \downarrow$ | | | | $\text{GFD}_{L2} \downarrow$ | | | |
|---|---|---|---|---|---|---|---|---|---|
| | | YouTube | DBLP | PolBlogs | Yelp | YouTube | DBLP | PolBlogs | Yelp |
| SyNG-D | 2 | 1.42 ± 0.91 | 3.80 ± 0.96 | 0.76 ± 0.39 | 0.38 ± 0.09 | 0.70 ± 0.48 | 1.97 ± 0.51 | 0.33 ± 0.19 | 0.16 ± 0.05 |
| | 3 | **1.39 ± 0.97** | **3.03 ± 0.98** | **0.72 ± 0.42** | 0.42 ± 0.11 | **0.68 ± 0.51** | **1.58 ± 0.52** | **0.32 ± 0.21** | 0.17 ± 0.05 |
| | 4 | 1.45 ± 0.98 | 3.35 ± 0.86 | 0.86 ± 0.46 | **0.30 ± 0.11** | 0.71 ± 0.50 | 1.75 ± 0.45 | 0.38 ± 0.23 | **0.13 ± 0.05** |
| | 5 | 1.45 ± 0.97 | 3.17 ± 0.90 | 0.89 ± 0.52 | 0.32 ± 0.13 | 0.71 ± 0.50 | 1.67 ± 0.47 | 0.40 ± 0.26 | 0.14 ± 0.06 |
| | 6 | 1.69 ± 1.05 | 3.46 ± 0.94 | 0.97 ± 0.51 | 0.31 ± 0.14 | 0.83 ± 0.54 | 1.83 ± 0.49 | 0.43 ± 0.25 | 0.14 ± 0.06 |
| SyNG-D(MLP) | 2 | 2.08 ± 0.98 | 6.92 ± 1.11 | **1.19 ± 0.52** | 0.35 ± 0.11 | 1.04 ± 0.50 | 3.60 ± 0.59 | **0.51 ± 0.26** | 0.14 ± 0.05 |
| | 3 | 2.55 ± 1.03 | 9.51 ± 0.86 | 1.23 ± 0.52 | 0.37 ± 0.19 | 1.24 ± 0.50 | 5.04 ± 0.48 | 0.55 ± 0.25 | 0.17 ± 0.10 |
| | 4 | **1.71 ± 0.86** | **6.17 ± 0.74** | 1.47 ± 0.38 | 0.34 ± 0.13 | **0.82 ± 0.43** | **3.19 ± 0.39** | 0.61 ± 0.18 | 0.13 ± 0.05 |
| | 5 | 3.02 ± 0.96 | 7.22 ± 0.86 | 1.41 ± 0.55 | **0.23 ± 0.14** | 1.45 ± 0.46 | 3.73 ± 0.45 | 0.56 ± 0.26 | **0.11 ± 0.07** |
| | 6 | 2.68 ± 1.14 | 7.86 ± 0.73 | 1.37 ± 0.74 | 0.27 ± 0.11 | 1.28 ± 0.55 | 4.08 ± 0.39 | 0.60 ± 0.38 | 0.11 ± 0.05 |
| SyNG-R | 2 | 1.38 ± 1.01 | 1.91 ± 0.91 | **0.81 ± 0.39** | 0.39 ± 0.11 | 0.67 ± 0.53 | 0.99 ± 0.51 | **0.35 ± 0.19** | 0.15 ± 0.04 |
| | 3 | **1.33 ± 1.05** | 1.60 ± 0.89 | 0.83 ± 0.38 | 0.33 ± 0.11 | 0.65 ± 0.53 | 0.83 ± 0.49 | 0.36 ± 0.18 | 0.13 ± 0.04 |
| | 4 | 1.34 ± 1.07 | 1.31 ± 0.80 | 0.86 ± 0.38 | 0.23 ± 0.12 | **0.65 ± 0.53** | 0.67 ± 0.44 | 0.36 ± 0.18 | 0.09 ± 0.05 |
| | 5 | 1.36 ± 1.07 | **1.13 ± 0.72** | 0.87 ± 0.39 | 0.18 ± 0.11 | 0.65 ± 0.52 | **0.58 ± 0.39** | 0.37 ± 0.18 | 0.08 ± 0.05 |
| | 6 | 1.37 ± 1.08 | 1.14 ± 0.81 | 0.88 ± 0.37 | **0.18 ± 0.11** | 0.66 ± 0.53 | 0.58 ± 0.44 | 0.37 ± 0.18 | **0.08 ± 0.05** |
| VGAE | 2 | 6.53 ± 0.04 | 13.69 ± 0.01 | **2.95 ± 0.09** | **3.66 ± 0.01** | 2.97 ± 0.01 | 7.39 ± 0.01 | **1.45 ± 0.04** | **1.53 ± 0.00** |
| | 3 | 6.50 ± 0.04 | 2.54 ± 0.13 | 3.21 ± 0.08 | 3.85 ± 0.01 | 2.96 ± 0.01 | 1.14 ± 0.09 | 1.54 ± 0.04 | 1.61 ± 0.00 |
| | 4 | **5.91 ± 0.03** | 1.57 ± 0.14 | 3.08 ± 0.08 | 3.86 ± 0.01 | **2.79 ± 0.01** | **0.68 ± 0.06** | 1.50 ± 0.04 | 1.61 ± 0.00 |
| | 5 | 6.61 ± 0.03 | 1.57 ± 0.14 | 3.00 ± 0.08 | 3.80 ± 0.01 | 3.00 ± 0.01 | 0.69 ± 0.07 | 1.48 ± 0.04 | 1.58 ± 0.00 |
| | 6 | 6.20 ± 0.03 | 1.69 ± 0.15 | 3.19 ± 0.09 | 4.09 ± 0.01 | 2.88 ± 0.01 | 0.85 ± 0.07 | 1.51 ± 0.04 | 1.70 ± 0.00 |
| | 16 | 6.30 ± 0.03 | **1.51 ± 0.11** | 3.37 ± 0.08 | 3.93 ± 0.01 | 2.92 ± 0.01 | 0.80 ± 0.07 | 1.62 ± 0.04 | 1.64 ± 0.00 |
| GRAN | 128 | 1.90 ± 0.30 | 18.35 ± 0.14 | **1.95 ± 0.33** | - | 0.77 ± 0.14 | 9.92 ± 0.04 | **0.79 ± 0.19** | - |
| | 256 | 2.42 ± 0.18 | 17.26 ± 0.15 | 3.88 ± 0.76 | - | 1.07 ± 0.10 | 8.49 ± 0.53 | 1.73 ± 0.37 | - |
| | 512 | **1.67 ± 0.38** | 17.17 ± 0.08 | 3.04 ± 0.30 | - | **0.67 ± 0.17** | 8.22 ± 0.31 | 1.40 ± 0.16 | - |
| EDGE | - | **5.35 ± 0.49** | 5.27 ± 2.28 | 1.14 ± 0.29 | - | 2.12 ± 0.19 | 2.57 ± 1.35 | 0.45 ± 0.11 | - |
| GraphMaker | - | **7.97 ± 0.00** | 17.45 ± 0.01 | 7.88 ± 0.01 | 5.17 ± 0.00 | 3.48 ± 0.00 | 8.32 ± 0.00 | 3.23 ± 0.00 | 2.14 ± 0.00 |
| ER | - | **7.90 ± 0.00** | 17.39 ± 0.01 | 7.80 ± 0.01 | 5.02 ± 0.00 | 3.47 ± 0.00 | 8.32 ± 0.00 | 3.22 ± 0.00 | 2.08 ± 0.00 |
| BTER | - | **2.24 ± 0.10** | 13.84 ± 0.01 | 2.09 ± 0.09 | 0.73 ± 0.01 | 0.95 ± 0.05 | 7.54 ± 0.00 | 0.98 ± 0.04 | 0.30 ± 0.01 |
| mKPGM | - | **7.57 ± 0.01** | 17.47 ± 0.01 | 7.83 ± 0.03 | 4.26 ± 0.01 | 3.32 ± 0.01 | 8.28 ± 0.00 | 3.09 ± 0.01 | 1.74 ± 0.01 |

C.8    SUPPLEMENTARY FOR THE EFFICIENCY COMPARISON

In this section, we begin with a note on e-FLOPs and then present a comparison of training and sampling time across methods, specifying the device environment used for each.

**A note on e-FLOPs.**    The definition of e-FLOPs considers two types of operations: the float-point operations on neural nets, and node visit operations on trees. These two operations are not directly comparable, as node visits involve comparison and branching which are different operations than the float-point operations on neural nets. In Section 4.3, we use e-FLOPs for a comparison which is less dependent on the implemented device environments. As a supplementary, we study the wall-block training and sampling time of each methods with their implemented device environments specified.

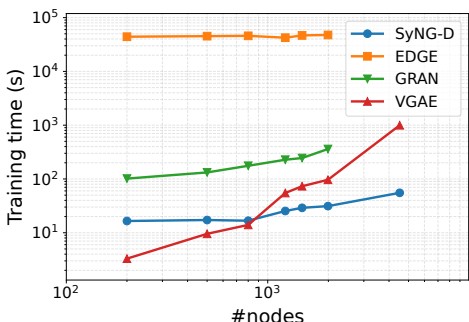

Figure 5: Wall-clock training time of different methods for datasets of different sizes.

**Evaluation metrics and configuration.**    We compare training and sampling efficiency between SyNG-D and the baseline methods through the time they spend during training and sampling. SyNG-D and VGAE are trained on CPUs, while GRAN and EDGE are trained on a single NVIDIA GeForce RTX 4090 with memory of 24GB. For each dataset, we train each model using the default training schedule, sample 128 networks from each model, and record the wall-clock training time and average sampling time.

**Results and discussion.**    Figure 5 presents the training time comparisons between our method and the baselines. In particular, our model can be trained in tens of seconds even for graphs with up to 5,000 nodes, whereas deep learning based methods typically require on the order of hundreds to thousands of seconds. As network size increases, the training time for our method grows at a moderate rate, while the training time for VGAE increases much faster. This indicates that the computational cost of SyNG-D remains relatively stable as network size grows, demonstrating its advantages on large-scale networks. The training time of EDGE and GRAN remains high across all dataset sizes.

For sampling speed, on networks with fewer than 1,000 nodes, both SyNG-D and VGAE complete a single draw in under $0.1\,\text{s}$ on average, whereas EDGE and GRAN require a few seconds. SyNG-D requires only a small number of diffusion steps to generate high-quality latent embeddings. For larger networks with 5,000 nodes, SyNG-D samples a synthetic network in only a few seconds, significantly faster than GRAN and EDGE, which require tens of seconds. The sampling time of SyNG-D is also more stable as network size increases, compared with the other methods.

These results highlight the scalability of SyNG-D and its advantage for large graphs.

C.9    ANALYSIS OF COMPLEXITY

We provide an analysis of the time complexity of the training process. As suggested by Ma et al. (2020), each iteration of the projected gradient descent involves only matrix multiplication between the adjacency matrix and the current latent embedding estimate. Therefore, the per-iteration time complexity of the projected gradient descent in estimating the latent node embeddings is $O(n^2 r)$, where $n$ is the number of nodes and $r$ is the latent dimension. Meanwhile, the cost of generating a

new group of latent embeddings is $O(nr)$, where the omitted constant depends on the model and implementation. Since the latent embedding is typically low-dimensional, the overall time complexity of the training and inference of our approach is $O(n^2 r)$.

In the context of a large-scale sparse network dataset, the proposed SyNGLER framework remains scalable with a simple adaptation. Concretely, under the Gaussian link function $p(a_{ij} \mid \pi_{ij}) \propto (2\pi\sigma^2)^{-1/2} \exp\{-(a_{ij} - \pi_{ij})^2/(2\sigma^2)\}$, the embedding estimation problem Eq. (1) can be efficiently solved with top-$r$ singular value decomposition. According to Saad (2003), the time complexity of eigen-decomposition for a sparse adjacency matrix $A$ is $O(|E(A)|r + nr)$, where $E(A)$ denotes the edge set associated with the adjacency matrix $A$. When the total number of edges is $O(n^c)$ with $c < 2$, the resulting time complexity grows more slowly than that of using a logistic link. In scenarios involving very large-scale and sparse networks, this approach is highly computationally efficient. We empirically study this approach on a network with one million nodes in Appendix D.

For other deep generative models, Zhu et al. (2022) suggests that the per-iteration time complexity to train a deep generative model over a graph of size $n$ is typically $O(n^2 M)$, where $M$ is the number of parameters in the model. Note that deep generative models typically have a large number of parameters $M$, which can be much larger than $r$. Therefore, we conclude that our method has a lower time complexity than deep generative models, which is consistent with our empirical results.

# D    EVALUATION OF SYNGLER ON LARGE SCALE NETWORK DATASET

In this section, we evaluate the scalability and effectiveness of SyNGLER on a large-scale synthetic network. To illustrate the capability of our framework under extreme graph sizes and sparsity levels, we construct a massive benchmark using the Stochastic Block Model (SBM) and assess whether SyNGLER is able to faithfully reproduce its structural patterns.

**Large-Scale Network Simulation via SBM.**    We first generate a network with $n = 10^6$ nodes from a three-block Stochastic Block Model. The network is designed to be extremely sparse, with an average degree of approximately 5. This setup provides a controlled environment to examine whether SyNGLER can recover community structure and degree behavior at scale.

**Latent Space Estimation.**    We apply our latent space model to embed the one-million-node network into a continuous low-dimensional space. To ensure computational feasibility at the one-million-node scale, we apply a linear latent space model to obtain the embeddings.

**Generative Resampling via SyNG-D (MLP).**    Using the estimated latent positions, we train our SyNG-D (MLP) model to resample latent embeddings. The model learns the distribution of the latent embeddings and enables resampling of synthetic latent vectors that preserve the structural structure and cluster patterns present in the original data.

Direct visualization or adjacency-level comparison is infeasible for networks of this scale. Instead, we validate the generative resampling by examining whether key structures are preserved. In particular, Figure 6 shows that SyNG-D(MLP) successfully recovers the global community interaction patterns and preserves important spectral characteristics of the graph.

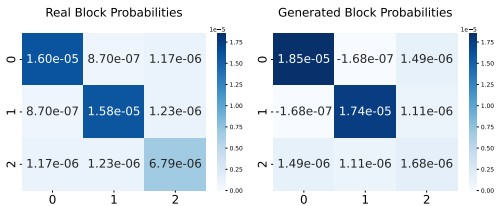 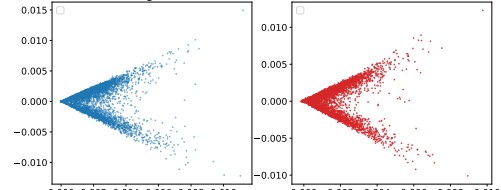

(a) Estimated $3 \times 3$ block interaction matrix from the generated network compared with the SBM ground truth.

(b) Scatter plot of the first two dimensions of the eigenvectors of the original and generated networks.

Figure 6: Evaluation of SyNG-D(MLP) on the one-million-node SBM network.

# E  DETAILS OF SYNGLER-ATTR AND ITS EVALUATION RESULTS

In this section, we provide additional details on the SyNGLER-Attr procedure and report its empirical performance on several attributed network datasets. We first present the full algorithmic workflow of SyNGLER-Attr, including latent factor estimation, attribute decomposition, and the joint mechanism for network and node attributes generation. The complete procedure is summarized in Algorithm 3. In practice, we use the sigmoid function as the link function when modeling binary networks, so that $\text{Bernoulli}(g(\cdot))$ reduces to a standard logistic formulation for edge probabilities.

---

**Algorithm 3** Synthetic Network Generation via Latent Emedding Reconstruction for Attributed Network

---

1: **Input:** Adjacency matrix $A \in \{0,1\}^{n \times n}$, Attribute matrix $Y \in \mathbb{R}^{n \times p}$.
2: Fit the latent space likelihood model $(A_{ij}|z_{1i}, z_{1j}, \alpha_i, \alpha_j) \sim \text{Bernoulli}(g(z_{1i}^\top z_{1j} + \alpha_i + \alpha_j))$ to obtain the MLE $(\widehat{Z}_1, \widehat{\alpha})$.
3: Regress $Y$ on $Z_1$ under $Y = \mathbf{1}_n \widehat{\mu}^\top + \widehat{Z}_1 \widehat{\Lambda}_1^\top + R$, where $R$ denotes the residual matrix.
4: Conduct an eigenvalue ratio test on residual $R$ to determine $d_2$, then fit $R = Z_2 \Lambda_2^\top + E$, $E_{i\cdot} \sim N(0, \Psi)$ to obtain $(\widehat{Z}_2, \widehat{\Lambda}_2)$.
5: Form the full latent embedding $\widehat{Z} = (\widehat{Z}_1, \widehat{Z}_2)$ and train a generative model Sampler $=$ $\text{GenModel}(\{(\widehat{z}_i, \widehat{\alpha}_i)\}_{i=1}^n)$.
6: Generate new latent samples $(\tilde{z}_i, \tilde{\alpha}_i) \sim \text{Sampler}$, $i = 1, \ldots, n$, where $\tilde{z}_i = (\tilde{z}_{1i}^\top, \tilde{z}_{2i}^\top)^\top$.
7: With sampled latent variables $\tilde{z}_i, \tilde{\alpha}_i$, generate network edges via $\tilde{A}_{ij} = \tilde{A}_{ji} \sim p(\cdot \mid \tilde{z}_{1i}^\top \tilde{z}_{1j} + \tilde{\alpha}_i + \tilde{\alpha}_j)$, and generate attributes via $\tilde{Y} = \mathbf{1}_n \widehat{\mu}^\top + \tilde{Z}_1 \widehat{\Lambda}_1^\top + \tilde{Z}_2 \widehat{\Lambda}_2^\top$.
8: **Output:** Generated network $\tilde{A}$ and generated attributes $\tilde{Y}$.

---

Empirically, we evaluate the SyNGLER-Attr procedure on the `Cora` dataset, a widely used attributed network benchmark.

Table 27: Generation performance of SyNG-Attr and GraphMaker on the Cora dataset.

| Method | $\text{RMSE}_{Tri.}\downarrow$ | $\text{RMSE}_{Clus.}\downarrow$ | $\text{MMD}_{Eig.}\downarrow$ | $\text{MMD}_{DegC.}\downarrow$ | ML-R |
|---|---|---|---|---|---|
| SyNG-Attr$_{\text{MLP}}$ | 36.07 | 5.38 | **0.68 $\pm$ 0.00** | 2.61 $\pm$ 0.15 | 0.98 $\pm$ 0.01 |
| SyNG-Attr$_{\text{Forest}}$ | 0.54 | **2.09** | 0.88 $\pm$ 0.00 | **0.32 $\pm$ 0.05** | 0.99 $\pm$ 0.02 |
| SyNG-Attr$_{\text{R}}$ | **0.48** | 5.21 | 0.89 $\pm$ 0.01 | 0.50 $\pm$ 0.04 | **1.00 $\pm$ 0.00** |
| GraphMaker | 0.58 | 7.72 | 1.03 $\pm$ 0.01 | 2.36 $\pm$ 0.01 | **1.00 $\pm$ 0.00** |

In comparison with GraphMaker (Li et al., 2023), we observe that SyNGLER-Attr produces synthetic graphs with close structural statistics while maintaining comparable attribute-level accuracy. Table 27 reports the structural and ML-utility metrics, showing that SyNGLER-Attr achieves competitive or superior performance on triangle density, clustering coefficient, and spectral and centrality-based measures. For the ML-utility metric, we adopt a link prediction task, with the full evaluation pro-

Table 28: KS and MMD distances between row sums of generated and original attributes for SyNG-Attr and GraphMaker on the Cora dataset.

| Method | $d_{\text{KS}}\downarrow$ | $d_{\mathcal{W}_1}\downarrow$ | MMD $\downarrow$ |
|---|---|---|---|
| SyNG-Attr$_{\text{MLP}}$ | 0.1398 | 2.8085 | **0.1676** |
| SyNG-Attr$_{\text{Forest}}$ | **0.1225** | **1.9584** | 0.1681 |
| SyNG-Attr$_{\text{R}}$ | 0.1457 | 2.3231 | 0.1701 |
| GraphMaker | 0.1400 | 5.1615 | 0.1678 |

tocol detailed in Appendix F. To further evaluate attribute quality, we compute the Kolmogorov–Smirnov (KS) distance and Maximum Mean Discrepancy (MMD) between the generated and original attributes. As reported in Table 28, SyNGLER-Attr achieves smaller discrepancies than GraphMaker, confirming its ability to preserve attribute distributions.

# F EVALUATIONS FOR GENERATED GRAPHS ON DOWNSTREAM TASKS

To assess whether a generated graph can serve as a reliable surrogate for downstream machine learning tasks, we employ a discriminative-model–based evaluation protocol adapted from Li et al. (2023). For a given dataset, we train a GCN-based graph auto-encoder (GAE) on the training split of the original graph, yielding a model with parameters $\widehat{W}$. We then train the same architecture on a generated graph $\hat{G}$ to obtain a second model with parameters $\breve{W}$. Both models are subsequently evaluated on the *same* held-out test split of the original graph, resulting in two AUC scores $\text{ACC}(G \mid G)$ and $\text{ACC}(G \mid \hat{G})$, corresponding to the model trained on the original graph and the one trained on the generated graph, respectively. We use the ratio

$$\frac{\text{ACC}(G \mid \hat{G})}{\text{ACC}(G \mid G)}$$

as the utility metric. A ratio close to one indicates that the generated graph offers comparable signal for training the GAE model, and therefore retains the structural information relevant for link prediction. All hyperparameters are tuned consistently across both training procedures to ensure a fair comparison. The results across all four datasets are summarized in Table 29.

Table 29: ML utility evaluation of SyNG-D, SyNG-R, EDGE, GRAN, and GraphMaker across four datasets. Entries marked with "–" indicate OOM issues.

| Method | Config | DBLP | PolBlogs | YouTube | Yelp |
|---|---|---|---|---|---|
| SyNG-D | 2 | **1.00 ± 0.00** | 0.98 ± 0.01 | 0.94 ± 0.02 | 0.98 ± 0.00 |
| | 3 | 1.00 ± 0.00 | 0.98 ± 0.01 | 0.98 ± 0.01 | 0.99 ± 0.00 |
| | 4 | 1.00 ± 0.00 | 0.99 ± 0.01 | 0.98 ± 0.01 | **0.99 ± 0.00** |
| | 5 | 1.00 ± 0.00 | 0.99 ± 0.01 | 0.99 ± 0.01 | 0.99 ± 0.00 |
| | 6 | 1.00 ± 0.00 | **0.99 ± 0.00** | **0.99 ± 0.01** | 0.99 ± 0.00 |
| SyNG-D(MLP) | 2 | 1.00 ± 0.00 | 0.98 ± 0.01 | 0.90 ± 0.02 | 0.98 ± 0.00 |
| | 3 | 1.00 ± 0.00 | **0.99 ± 0.01** | 0.98 ± 0.01 | 0.99 ± 0.00 |
| | 4 | **1.00 ± 0.00** | 0.96 ± 0.01 | 0.96 ± 0.01 | 0.99 ± 0.00 |
| | 5 | 1.00 ± 0.00 | 0.97 ± 0.01 | **0.99 ± 0.00** | **0.99 ± 0.00** |
| | 6 | 1.00 ± 0.00 | 0.98 ± 0.01 | 0.98 ± 0.01 | 0.99 ± 0.00 |
| SyNG-R | 2 | 1.00 ± 0.00 | 0.98 ± 0.01 | 0.96 ± 0.02 | 0.98 ± 0.00 |
| | 3 | 1.00 ± 0.00 | 0.98 ± 0.01 | 0.98 ± 0.01 | 0.99 ± 0.00 |
| | 4 | 1.00 ± 0.00 | 0.98 ± 0.01 | 0.98 ± 0.01 | 0.99 ± 0.00 |
| | 5 | **1.00 ± 0.00** | 0.98 ± 0.01 | 0.99 ± 0.01 | 1.00 ± 0.00 |
| | 6 | 1.00 ± 0.00 | **0.99 ± 0.01** | **0.99 ± 0.01** | **1.00 ± 0.00** |
| VGAE | 2 | 1.00 ± 0.00 | 1.01 ± 0.00 | 1.00 ± 0.00 | 1.00 ± 0.00 |
| | 3 | 1.00 ± 0.00 | 1.01 ± 0.00 | **1.00 ± 0.00** | 1.00 ± 0.00 |
| | 4 | **1.00 ± 0.00** | 1.01 ± 0.00 | 1.00 ± 0.00 | 1.00 ± 0.00 |
| | 5 | 1.00 ± 0.00 | 1.01 ± 0.00 | 1.00 ± 0.00 | 1.00 ± 0.00 |
| | 6 | 1.00 ± 0.00 | 1.01 ± 0.00 | 0.99 ± 0.00 | **1.00 ± 0.00** |
| | 16 | 1.00 ± 0.00 | 1.01 ± 0.00 | 0.99 ± 0.00 | 1.00 ± 0.00 |
| GRAN | 128 | **0.98 ± 0.08** | 0.92 ± 0.08 | **1.00 ± 0.00** | - |
| | 256 | 0.75 ± 0.00 | **1.04 ± 0.00** | 0.99 ± 0.00 | - |
| | 512 | 0.83 ± 0.06 | 1.04 ± 0.00 | 0.98 ± 0.02 | - |
| EDGE | – | **1.00 ± 0.00** | 0.98 ± 0.01 | **1.00 ± 0.00** | - |
| GraphMaker | – | **0.95 ± 0.02** | **1.00 ± 0.00** | **0.99 ± 0.00** | **0.83 ± 0.01** |
| ER | – | **0.90 ± 0.03** | **1.00 ± 0.00** | **0.99 ± 0.00** | **0.80 ± 0.01** |
| BTER | – | **0.85 ± 0.05** | **0.95 ± 0.01** | **0.91 ± 0.01** | **0.94 ± 0.01** |
| mKPGM | – | **0.96 ± 0.02** | **1.01 ± 0.00** | **0.92 ± 0.01** | **0.89 ± 0.02** |

We observe that both SyNG-D and SyNG-R consistently produce AUC ratios extremely close to one, indicating that the generated graphs preserve the predictive signal necessary for training link prediction models. In particular, SyNG-D achieves stable performance across all latent dimensions, and SyNG-R demonstrates similarly strong results with small variance. Compared with existing baselines such as EDGE, GRAN, and GraphMaker, SyNGLER exhibits both higher accuracy and greater robustness across datasets, further validating its effectiveness as a general-purpose synthetic graph generator for downstream ML tasks.

## G  VISUALIZATIONS

**Visualization of generated networks.**   We visualize the YouTube dataset with different layout algorithms to provide an intuitive comparison of graph generation quality across methods. In the main text, Figure 2 shows the visualization produced by the spring_layout method in networkx(Hagberg et al., 2008), which utilize the Fruchterman-Reingold force-directed algorithm to highlight the structural patterns of network. Since different visualization algorithms may reveal different aspects of a network's geometry, we include additional visualizations in this section under multiple layout schemes to offer a more comprehensive comparison of the generated graphs.

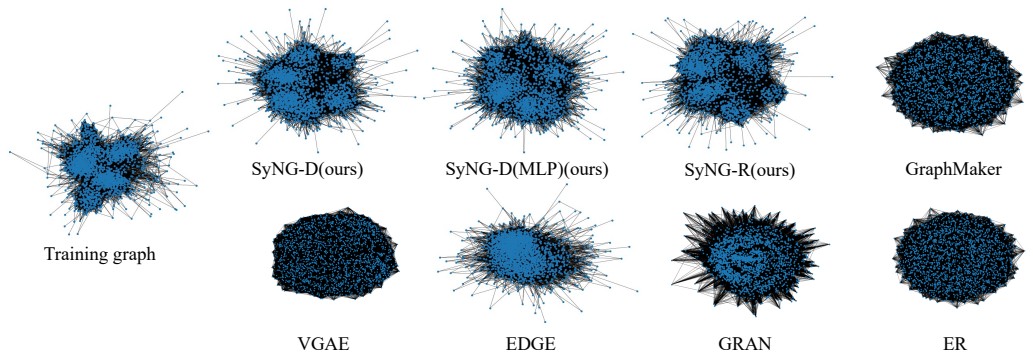

Figure 7: Visualization via the Spring layout.

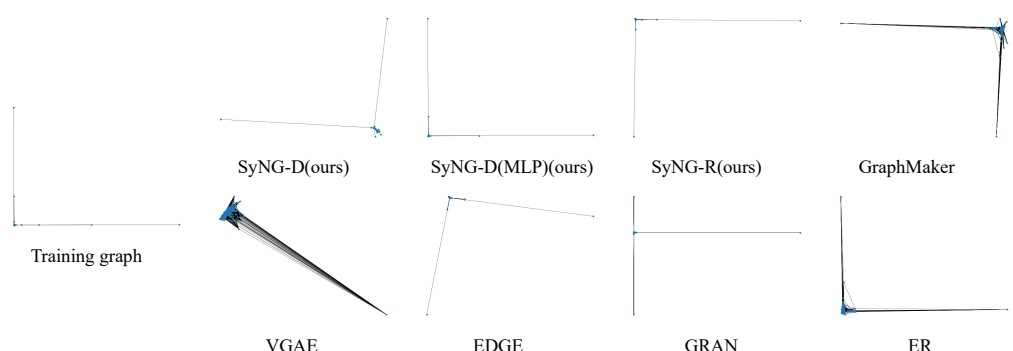

Figure 8: Visualization via the Spectral layout.

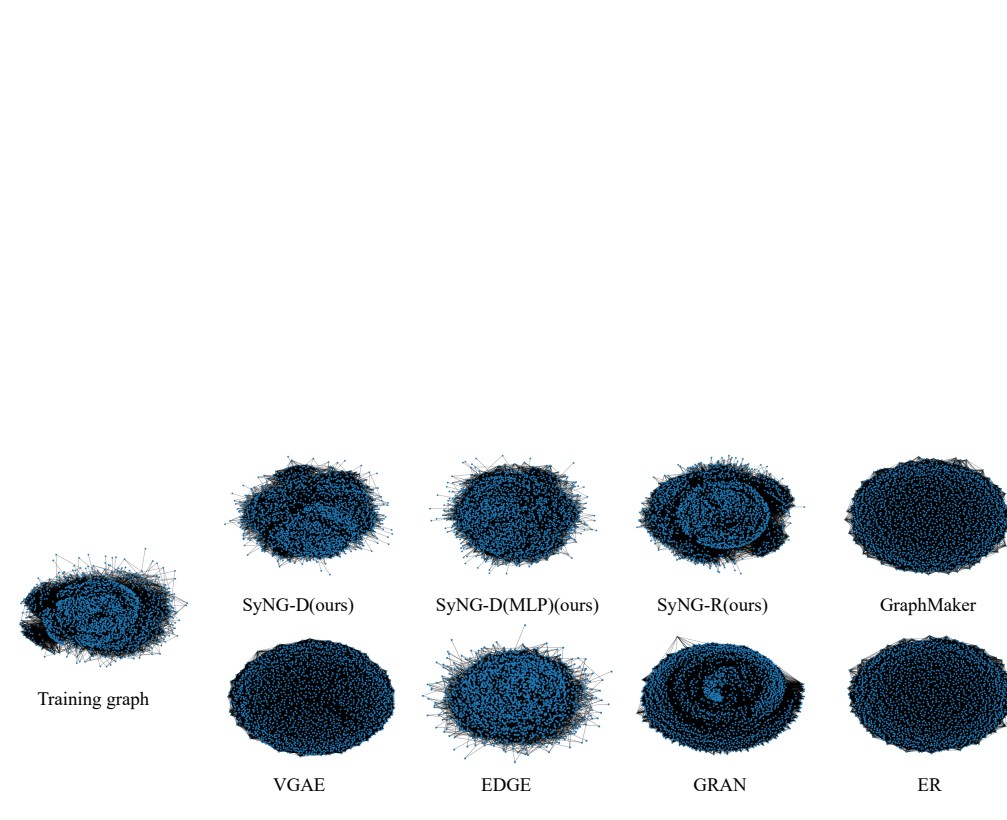

Figure 9: Visualization via the Kamada kawai layout.

