# OpenReview forum: "Efficient Synthetic Network Generation via Latent Embedding Reconstruction"
_ICLR.cc/2026/Conference — Submitted to ICLR 2026_

### Official Review · Reviewer_qzcR · 2025-10-18

**Soundness:** 2
**Presentation:** 1
**Contribution:** 3
**Rating:** 2
**Confidence:** 4

**Summary:**

The paper presents a new generative network model, SyNGLER, that can replicate the network's structure. SyNGLER embeds the nodes into a low-dimensional space of dimensions r. Each edge A_{ij} is sampled according to the parameter \pi_{ij}, which can have a different distribution depending on the edge type (binary, continuous, etc.). \pi_{ij} is defined by the sum of \z_i^Tz_j+\alpha_i+\alpha_j+\rho_n, where \z_i is the embedding of the node, \alpha_i is the degree parameter, and \rho_n is a global sparsity parameter.

**Strengths:**

The introduction and related work are directly focused on the main contribution.

The main contribution is interesting. It presents a general framework based on low-dimensional spatial embeddings that possesses several important properties.

Algorithm 1 appears to be general enough to work with any embedding used in the process. It is also a very simple and understandable pseudocode.

The synthetic experiment shows that the model is able to replicate networks using the same method.

**Weaknesses:**

The assumptions must be improved. I greatly appreciated the simple explanation for Assumption 2.1. However, given that all these assumptions appear to come from the paper by Li et al. (2025), which is on ArXiv and has not been peer-reviewed, this paper should provide a demonstration.

The replicability of the main paper is very low. While the algorithm is simple to follow, the embedding learning process is not described in the main paper. For example, subsection 2.3 starts with a learned embedding that is not described in the main paper. Similarly, the training process for \alpha and \rho and the SAMPLER process are not described. I understand that some cases are considered, such as the replication of the embedding, and others, but more details are needed. I also understand that the main idea is the proposition of a general framework, but some ideas must be given to allow the reproducibility of the paper.

Theorem 3.1 is not demonstrated. Eq. (3) states a separation of the d_{KL} into three terms, which are later explained, but I could not see the proof of the theorem. If the demonstration is in another paper (Chen et al.), it must explicitly state that this was demonstrated in that paper to avoid confusion about a potential contribution.

Most of the theoretical contributions are based on a reference to other papers that have not been peer-reviewed (Chen et al., Wu et al., Li et al.), and similar to Theorem 3.1, I could not figure out their demonstration.

One of the main claims of the paper is its low training time complexity. Unfortunately, there is no theoretical analysis of the process, only an empirical analysis of some embedding process.

The results seem to be misleading. The real networks have millions of nodes, but these are reduced to thousands by selecting the largest connected component. So, the final application is applied only to a small network (thousands of nodes), rather than millions of nodes, which is inconsistent with the main claim of the paper.

Several baselines use a similar approach (assuming a probability for each edge) and should be considered baselines (e.g., Chung-Lu, mKPGM, BTER, etc.). This will improve the paper's results.

Figure 2 is misleading; a network can be shown in different forms using different visualization algorithms. This should not be used for comparisons.

As shown in Figure 3, the proposed model is applied to very small networks. Another issue is the comparison against deep network models. Considering the similar process to the classical method (sampling an edge with a specific probability), it should be compared against these classical algorithms. Also, the training process seems to explode for a large number of nodes; therefore, the paper must increase the size of the network to consider millions of nodes.

Table 2 is not discussed in the main paper. The metrics are not discussed in the main paper either.

In general, the main contribution has potential, but an important rewriting of the paper must be done. The main paper is not self-contained, lacking important details, which reduces the replicability and analysis of the results.

Minor comments
The details of the Real-World datasets are in Appendix B.2

**Questions:**

Please the weaknesses.

---

> ### Author Response · Authors · 2025-11-24
> **Author Response to Reviewer qzcR**
>
> Thank you for taking the time and effort to review our paper. We appreciate your recognition of its potential and your many constructive comments, including those on the presentation. In this revision, we have addressed all of your comments to the best of our ability, leading to a substantially improved manuscript. Below, we respond to your comments one by one.
>
> **Your comment 1:** "The assumptions must be improved. I greatly appreciated the simple explanation for Assumption 2.1. However, given that all these assumptions appear to come from the paper by Li et al. (2025), which is on ArXiv and has not been peer-reviewed, this paper should provide a demonstration."
>
> **Our response:**
> Thank you for appreciating our explanation of the assumptions and for your suggestions on improving how we demonstrate them. In this revision, we have rewritten the proof, and now there are only two technical assumptions (instead of the four conditions previously listed in Assumption 2.1): Assumption 2.1, introduced with the model, and Assumption 3.1, stated before the main theorems. Each is followed by a discussion of its implications. We quote these two assumptions and the corresponding discussions below.
>
> Assumption 2.1 regarding the embedding distribution:
>
> > **Assumption 2.1 (embedding distribution).** The pairs $\{(z_i, \alpha_i)\}_{i=1}^n$ are i.i.d. from a distribution $\mathbb{P}_0$ on $\mathbb{R}^{r+1}$. We assume that the latent positions are centered, that is, $\mathbb{E}[z]= \mathbf{0} \in \mathbb{R}^r$.  Moreover, there exists $R>0$ such that $\|z\|_2\le R$ for all $(z,\alpha) \in \mathrm{supp}(\mathbb{P} _ 0)$.
> >
> >
> > Assumption 2.1 guarantees the boundedness of the embedding space and the identifiability of the node degree distribution, since joint shifts in $\alpha$ and $z$ can leave the edge distribution unchanged. Specifically, in (1), for all $i\in[n]$, let $\check z_i=z_i+a$ and $\check\alpha_i=\alpha_i-a^\top z_i-a^\top a/2$ for some $a\in\mathbb{R}^r$. Then $\check z_i^\top\check z_j+\check\alpha_i+\check\alpha_j=z_i^\top z_j+\alpha_i+\alpha_j$ for all $i,j\in[n]$, and the likelihood remains unchanged. Assumption 2.1 thus fixes this shift as $a=-\mathbb{E}_{\mathbb{P}_0}[z_i]$ to ensure node-degree identifiability. Similar conditions have been commonly considered in the latent space literature for node degree identifiability, for example, in Ma et al. (2020) and Zhang et al. (2022).
>
> Assumption 3.1 regarding network sparsity
>
> >**Assumption 3.1 (Network sparsity)** The global edge sparsity follows that $w_n=\exp(\rho_n^*)=\Omega(\log{n}/n)$.
> >
> > Assumption~3.1 states that the edge density is bounded below by  $\Omega(\log n / n)$, and accordingly the expected node degrees are at least of order $\log n$ as $n$ grows. Such a sparsity regime is consistent with the network analysis literature;
> > see, for example, Athreya et al. (2018) and Ma et al. (2020)
>
> The current assumptions not only appear in Li et al. (2025) but also in many other peer-reviewed references.
> We have provided additional references and discussion regarding these assumptions as quoted above.
>
> ---
>
> **Your comment 2:** "The replicability of the main paper is very low. While the algorithm is simple to follow, the embedding learning process is not described in the main paper. For example, subsection 2.3 starts with a learned embedding that is not described in the main paper. Similarly, the training process for $\alpha$ and $\rho$ and the SAMPLER process are not described. I understand that some cases are considered, such as the replication of the embedding, and others, but more details are needed. I also understand that the main idea is the proposition of a general framework, but some ideas must be given to allow the reproducibility of the paper."
>
> **Our response:** Thanks for your suggestion on improving replicability of the main paper by providing more details.
>
> To ensure replicability, we have released all the code in an anonymous repository at https://github.com/SyNGLER/SyNGLER/. This repository includes all the algorithmic implementations on fitting the latent space model and generating fresh embeddings.

---

> ### Author Response · Authors · 2025-11-24
>
> We have also provided more details regarding the implementation of the algorithm in the updated manuscript. The learned embedding is defined in the network embedding subsection before it is used in subsection 2.3 and we quote:
>
> > Once the conditional model is determined, degree parameters and latent embeddings can be estimated by maximizing the following likelihood function over the parameters $(Z,\alpha)$, where $Z = (z\_1^\top, z\_2^\top, \ldots, z\_n^\top)^\top$ is the latent position matrix with rows $z\_1,z\_2,\dots,z\_n$ and $\alpha = (\alpha\_1,\ldots,\alpha\_n)$ is the vector of degree parameters: $$ (\hat Z, \hat{\alpha}) = \mathrm{argmax}\_{\{(Z, \alpha) \in \mathbb{R}^{n \times (r+1)}
> : Z^{\top} \mathbf{1}\_n = \mathbf{0}\_r\}} \sum\_{1 \le i < j \le n} \log p\big(A\_{ij} \mid z\_i^\top z\_j + \alpha\_i + \alpha\_j\big). (2)$$
> >
> > The constraint in Eq. (2) ensures identifiability of the degree parameters and is designed based on Assumption 2.1. A projected gradient descent algorithm can be employed to efficiently solve Eq. (2) with provable convergence guarantees (Ma et al., 2020). We leave the details of the optimization algorithm to Appendix C.1.
>
> The training process has been specified in the main text as specifying the denoising score-matching objective:
>
> > Given the forward process, the parameter $\theta$ in $s\_{\theta}(x,t)$ is optimized by minimizing the denoising score-matching objective (Song et al., 2020) constructed using the learned embeddings:
> >
> > $$
> \hat{\theta}= \mathrm{argmin}\_{\theta} \mathbb{E}\_{t\sim \mathcal{U}[0,1],\, z\sim \mathcal{N}(0,I\_{r+1})} \Bigg[\frac{1}{n} \sum\_{i=1}^n\Big\|s\_{\theta}\big(e^{-t}\hat\phi\_i + \sqrt{1-e^{-2t}}\,z,\,t\big)+\frac{z}{\sqrt{1-e^{-2t}}}\Big\|^2\Bigg].
> $$
>
> We have also added more details in the revision to make the algorithmic framework easier to reproduce, including the projected gradient descent algorithm and the fitting of the sparsity parameter $\hat\rho$ (we have reorganized the presentation so that $\rho$ appears only when we specifically investigate sparse binary networks as the expected node degree $\mathbb{E}[\alpha_i]$) and the degree parameters $\alpha$ in Appendix B.1, as well as additional details on the score-based generative model in Appendix C.4; these are referenced in the empirical study section of the main text.
>
>
> ---
>
> **Your comment 3:** "Theorem 3.1 is not demonstrated. Eq. (3) states a separation of the $d_{KL}$ into three terms, which are later explained, but I could not see the proof of the theorem. If the demonstration is in another paper (Chen et al.), it must explicitly state that this was demonstrated in that paper to avoid confusion about a potential contribution."
>
> **Our response:** Thanks for your critique on presentation of the theoretical proofs. The proof for Theorem 3.1 was included in Appendix A in the initial submission, but appeared only after two technical lemmas. We realize it might not be easy for readers to find the proof; we have updated the main text to include a pointer to the proof in the appendix. Moreover, the proof section in the appendix (Appendix A) has been reorganized so that the proofs of the theorems appear in the order in which they are stated in the main text, with the technical lemmas placed at the end of the section.
>
> ---
>
> **your comment 4:** "Most of the theoretical contributions are based on a reference to other papers that have not been peer-reviewed (Chen et al., Wu et al., Li et al.), and similar to Theorem 3.1, I could not figure out their demonstration."
>
> **Our response:**
> Thank you for the critique on the self-containedness of the theoretical results in our paper. In this revision, we have rewritten parts of the proofs so that all theoretical results are proved within the paper, without relying on results from other work. Specifically, Appendix A.1–A.5 now provide detailed proofs of Theorems 3.1–3.3, Lemma 3.1, and Theorem 3.4, respectively. In addition, all supporting lemmas and their proofs are included in Appendix A.6. The updated version of the proofs is fully self-contained and covers all details needed to establish the corresponding theoretical results. Due to the technical difficulty of rendering equations in this comment chunk, we kindly refer you to Appendix A of our updated manuscript for more details.

---

> ### Author Response · Authors · 2025-11-24
>
> **Your comment 5:** "One of the main claims of the paper is its low training time complexity. Unfortunately, there is no theoretical analysis of the process, only an empirical analysis of some embedding process."
>
> **Our response:** Thanks for your suggestion on the theoretical analysis of the training time. We agree that it is important to understand the efficiency of our method from a theoretical perspective for demonstrating this point. Below we first explain the efficiency and then show discussion regarding the theoretical complexity.
>
> First, the efficiency of our method, compared with many deep architectures, lies in its ability to avoid training deep models on a high-dimensional parameter space: we first embed the high-dimensional network into a low-dimensional continuous space and then use a score-based generative model to reconstruct this low-dimensional embedding space. We have emphasized this in the Introduction and quote it below:
>
> > Via the latent space approach, SyNGLER avoids training deep models directly on the high-dimensional network space by learning low-dimensional node embeddings with flexibly chosen likelihood models and requiring only lightweight generative model training in the latent space, thereby reducing computational cost.
>
> Second, the flexibility in choosing the likelihood models allows us to achieve an efficiency–accuracy tradeoff. For example, when dealing with very large networks (millions of nodes), one can use sparse top-$r$ singular value decomposition (SVD) to efficiently obtain the embeddings, with time complexity $O(nr + |E|r)$, where $|E|$ is the number of edges. For smaller-scale networks, we can use the logistic model adopted in most parts of this paper, whose time complexity is $O(n^2 r)$ per iteration; this typically yields better accuracy with lower embedding dimensions than sparse top-$r$ SVD, but is more time-consuming. Nevertheless, these embedding approaches have much smaller time complexity than training deep models directly on the networks, which is typically $O(n^2 M)$, where $M$ is the number of parameters. We will provide more details on these points in our responses to your comments 6, 7, and 9 below. Meanwhile, we are drafting a corresponding discussion with more details in the manuscript and will update it as soon as it is finished.
>
> ---
> **Your comment 6:** "The results seem to be misleading. The real networks have millions of nodes, but these are reduced to thousands by selecting the largest connected component. So, the final application is applied only to a small network (thousands of nodes), rather than millions of nodes, which is inconsistent with the main claim of the paper."
>
> **Our response:** We appreciate your suggestion to include larger networks in our experiments to better demonstrate the scalability of our method.
>
> As a remark, deep graph generative models typically handle only small networks with hundreds or thousands of nodes due to their high time complexity. Many tasks involve networks at this scale (e.g., [7, 8, 9]), for which our method is also applicable. Training deep graph generative models on larger networks (millions of nodes) is usually infeasible, as the training cost would be prohibitive.
>
> By contrast, our method can still be applied to sparse large networks with millions of nodes. As noted in the response to your comment 5, using sparse SVD for embedding is one option within our framework (which we will formally describe in response to your point 9), and it can handle networks of this size. To demonstrate this, we have conducted a simulation on a network of this scale. The procedure is as follows.
>
>
> 1. **Large-Scale Network Simulation via SBM.** We first generate a network with $n = 10^6$ nodes from a three-block Stochastic Block Model. The network is designed to be extremely sparse, with an average degree of approximately 5.
> 2. **Latent Space Estimation.** We embed the one-million-node network into a continuous low-dimensional space via applying a linear latent space model and using sparse SVD.
> 3. **Generative Resampling via SyNG-D (MLP).** Using the estimated latent positions, we train our SyNG-D (MLP) model to sample new latent embeddings and generate the network accordingly.
>
> As our results show, the model learns the distribution of the latent embeddings and enables resampling of synthetic latent vectors that preserve the structural patterns and cluster structure present in the original data. Direct visualization or adjacency-level comparison is not very informative for networks of this scale and sparsity. Instead, we empirically verify that the generated network remains close to the original one by comparing key structural summaries, including degree distributions, the leading two eigenvectors of the adjacency matrices, and the recovered block structures. Due to image-uploading constraints, all corresponding visualizations have been included in **Appendix D** of the revision.
>
> ---

---

> ### Author Response · Authors · 2025-11-25
>
> **Your comment 7:** "Several baselines use a similar approach (assuming a probability for each edge) and should be considered baselines (e.g., Chung-Lu, mKPGM, BTER, etc.). This will improve the paper's results."
>
> **Response:** Thanks for this constructive comment on these baselines. First, we would like to highlight that our framework encompasses general latent space network models, which include a wide class of classical network models such as the Chung–Lu model; in this revision, we have provided more detailed discussion of this point. Second, we have conducted additional empirical studies on these baselines, including the mKPGM and BTER models. Below we discuss these two points in sequence.
>
> For the generality of our framework and the network latent space model, we have included a discussion in the main text, which we quote below:
>
> > Remark 1. The latent space network model in Eq. (1) covers a wide class of classical network reconstruction models given appropriate choices of link functions and parametrizations of the node embeddings, including the Erdős–Rényi graph (Erdos & Rényi, 1960), the Chung–Lu graph (Chung & Lu, 2002), and the stochastic block model and its mixed-membership variants (Holland et al., 1983; Karrer & Newman, 2011; Airoldi et al., 2008). Further discussion is provided in Appendix B.
>
> We also quote the corresponding discussion in Appendix B as follows:
>
> > We illustrate the connection between these models and the latent space model below.
> > First of all, the general latent space network model assumes that each $A_{ij}\sim p(\cdot \mid \pi_{ij})$ where $\pi_{ij} = z_i^\top z_j + \alpha_i + \alpha_j$ and $p(\cdot \mid \pi)$ is a link function that can be chosen flexibly.
> > Below, we explain its connection to several classical network models with details.
> > We remark that some classical node-embedding models already belong to the latent space model.
> > For example:
> >
> > - **Chung-Lu graph model**: Assumes each node $i$ has a degree parameter $w_i > 0$, with $W := \sum_{k=1}^n w_k$ as the total weight. Edge probabilities are $\mathbb{P}(A_{ij} = 1) = w\_i w\_j / W$. This can be reformulated as a latent space model by setting $z_i = w_i / \sqrt{W} \in \mathbb{R}$ and constructing $Z = (z_1, \ldots, z_n)^\top \in \mathbb{R}^{n \times 1}$. With the linear link function $p(\cdot \mid \pi) = \mathrm{Bernoulli}(\pi)$, we have $\mathbb{E}[A] = ZZ^\top$.
> >
> > - **Random Dot Product Graph (RDPG) model**: Assumes each node $i$ has a latent position $z_i \in \mathbb{R}^r$ such that $z_i^\top z_j \in [0,1]$. With embedding matrix $Z = (z_1, \ldots, z_n)^\top \in \mathbb{R}^{n \times r}$, the model assumes $A \sim \mathrm{Bernoulli}(ZZ^\top)$, which is exactly a latent space model with the linear link function $p(\cdot \mid \pi) = \mathrm{Bernoulli}(\pi)$.
> >
> > - **Degree-Corrected Block Model (DCBM) / Stochastic Block Model (SBM)**: Assumes each node has a cluster label $g\_i \in [K]$ and degree parameter $\theta\_i$. The model specifies $\mathbb{E}[A]\_{ij} = \theta\_i \theta\_j B\_{g(i)g(j)}$, where $B \in [0,1]^{K \times K}$ is symmetric and positive semi-definite. Using symmetric decomposition $B = UU^\top$, we construct latent embeddings $z\_i = (\theta\_i U\_{g_i}^\top) \in \mathbb{R}^K$. With the linear link function $p(\cdot \mid \pi) = \mathrm{Bernoulli}(\pi)$, this becomes a latent space model $A \sim p(\cdot \mid ZZ^\top)$.
> >
> > - **Mixed-Membership Block Models**: Extends classical block models by allowing each node to be associated with multiple blocks. Specifies $\mathbb{E}[A]_{ij} = \pi_i B \pi_j^\top$, where each $\pi_i$ belongs to the probability simplex $\Delta_K = \{\pi: \pi \in [0,1]^K, \|\pi\|_1 = 1\}$. When $B$ has symmetric decomposition $B = UU^\top$ with $U \in \mathbb{R}^{K \times r}$, we can construct $Z = (z_1, \ldots, z_n)^\top$ with rows $z_i = U^\top \pi_i \in \mathbb{R}^K$. With the linear link function $p(\cdot \mid \pi) = \mathrm{Bernoulli}(\pi)$, we have $A \sim \mathrm{Bernoulli}(ZZ^\top)$, which is an instance of the latent space model.
> >
> >
> > These models fall within the latent space modeling framework considered in this paper under suitable choices of parametrization and link function, and can be incorporated into SyNGLER according to practitioners’ preferences, depending on the application context.
>
> Regarding mKPGM and BTER, we have included these models as additional baselines in our experiments on real-world networks. We list the comparison results in the tables below.

---

> ### Author Response · Authors · 2025-11-25
>
> **Table 1.** Yelp dataset, degree centrality. All entries scaled by $10^{-2}$.
>
> |Method|Config|W1 dist.|KS dist.|Energy dist.|MMD|
> |---|---|---|---|---|---|
> |SyNG-D|2|**0.15 ± 0.05**|**2.49 ± 0.74**|**0.54 ± 0.19**|**1.28 ± 0.87**|
> | |3|0.21 ± 0.10|3.55 ± 1.26|0.77 ± 0.34|2.36 ± 1.43|
> | |4|0.24 ± 0.10|4.54 ± 1.12|0.98 ± 0.33|3.46 ± 1.09|
> | |5|0.36 ± 0.11|6.02 ± 1.16|1.42 ± 0.37|5.03 ± 1.15|
> | |6|0.44 ± 0.13|7.72 ± 1.42|1.81 ± 0.44|6.72 ± 1.36|
> |SyNG-D(MLP)|2|0.32 ± 0.12|**4.68 ± 1.36**|1.18 ± 0.43|**3.26 ± 1.42**|
> | |3|0.39 ± 0.12|5.40 ± 1.30|1.38 ± 0.40|4.32 ± 1.18|
> | |4|0.69 ± 0.15|8.56 ± 1.33|2.49 ± 0.46|7.82 ± 1.23|
> | |5|**0.23 ± 0.08**|5.22 ± 1.03|**1.01 ± 0.26**|4.17 ± 0.79|
> | |6|0.30 ± 0.09|4.80 ± 1.06|1.19 ± 0.33|4.51 ± 0.96|
> |SyNG-R|2|**0.13 ± 0.06**|2.02 ± 0.80|**0.47 ± 0.22**|0.65 ± 0.94|
> | |3|**0.13 ± 0.06**|**1.98 ± 0.78**|**0.47 ± 0.22**|**0.60 ± 0.90**|
> | |4|**0.13 ± 0.06**|1.99 ± 0.82|**0.47 ± 0.23**|0.62 ± 0.92|
> | |5|0.14 ± 0.06|2.05 ± 0.80|0.48 ± 0.22|0.67 ± 0.94|
> | |6|**0.13 ± 0.06**|**1.98 ± 0.78**|**0.47 ± 0.22**|0.62 ± 0.91|
> |VGAE|2|**1.65 ± 0.00**|**22.12 ± 0.23**|**5.79 ± 0.02**|**32.23 ± 0.16**|
> | |3|1.70 ± 0.00|23.14 ± 0.22|5.98 ± 0.02|33.32 ± 0.19|
> | |4|1.71 ± 0.00|23.50 ± 0.21|6.03 ± 0.02|33.67 ± 0.17|
> | |5|1.69 ± 0.00|22.81 ± 0.22|5.97 ± 0.02|33.38 ± 0.16|
> | |6|1.79 ± 0.00|24.47 ± 0.20|6.34 ± 0.02|35.49 ± 0.14|
> | |16|1.73 ± 0.00|23.37 ± 0.21|6.09 ± 0.02|33.74 ± 0.18|
> |GraphMaker|-|**2.59 ± 0.00**|**41.44 ± 0.20**|**10.33 ± 0.01**|**69.26 ± 0.15**|
> |ER|-|**2.81 ± 0.00**|**57.17 ± 0.18**|**12.09 ± 0.03**|**77.36 ± 0.13**|
> |BTER|-|**0.04 ± 0.00**|**0.97 ± 0.17**|**0.14 ± 0.01**|**0.00 ± 0.00**|
> |mKPGM|-|**3.10 ± 0.00**|**49.19 ± 0.18**|**13.07 ± 0.02**|**49.97 ± 0.14**|
>
> **Table 2.** yelp, eigenvalues. W1 unscaled, KS, MMD scaled by $10^{-2}$, energy dist by $10^{-1}$.
>
> |Method|Config|W1 dist.|KS dist.|Energy dist.|MMD|
> |---|---|---|---|---|---|
> |SyNG-D|2|1.29 ± 0.09|5.29 ± 0.39|2.93 ± 0.24|6.11 ± 0.45|
> | |3|1.09 ± 0.10|4.43 ± 0.46|2.43 ± 0.27|5.08 ± 0.53|
> | |4|0.92 ± 0.08|3.76 ± 0.39|2.03 ± 0.23|4.26 ± 0.46|
> | |5|0.77 ± 0.08|3.02 ± 0.40|1.62 ± 0.23|3.36 ± 0.50|
> | |6|**0.64 ± 0.09**|**2.34 ± 0.46**|**1.28 ± 0.25**|**2.50 ± 0.62**|
> |SyNG-D(MLP)|2|2.07 ± 0.10|8.40 ± 0.39|4.89 ± 0.25|9.62 ± 0.45|
> | |3|1.88 ± 0.10|7.54 ± 0.39|4.43 ± 0.25|8.66 ± 0.44|
> | |4|2.00 ± 0.11|8.21 ± 0.40|4.79 ± 0.26|9.35 ± 0.46|
> | |5|**1.21 ± 0.10**|**5.06 ± 0.42**|**2.83 ± 0.25**|**5.74 ± 0.48**|
> | |6|1.55 ± 0.09|6.62 ± 0.37|3.73 ± 0.23|7.49 ± 0.42|
> |SyNG-R|2|1.32 ± 0.09|5.51 ± 0.41|3.03 ± 0.25|6.36 ± 0.47|
> | |3|1.22 ± 0.09|5.14 ± 0.41|2.82 ± 0.25|5.90 ± 0.48|
> | |4|1.10 ± 0.09|4.74 ± 0.40|2.55 ± 0.24|5.39 ± 0.47|
> | |5|1.00 ± 0.09|4.37 ± 0.38|2.33 ± 0.24|4.93 ± 0.47|
> | |6|**0.94 ± 0.09**|**4.21 ± 0.37**|**2.21 ± 0.23**|**4.69 ± 0.46**|
> |VGAE|2|2.44 ± 0.01|14.01 ± 0.04|6.51 ± 0.02|**17.28 ± 0.04**|
> | |3|2.46 ± 0.01|14.10 ± 0.04|6.56 ± 0.02|17.55 ± 0.04|
> | |4|2.47 ± 0.01|14.15 ± 0.04|6.58 ± 0.02|17.63 ± 0.04|
> | |5|2.48 ± 0.01|14.16 ± 0.04|6.60 ± 0.02|17.49 ± 0.04|
> | |6|2.51 ± 0.01|14.36 ± 0.04|6.71 ± 0.02|17.92 ± 0.03|
> | |16|**2.42 ± 0.01**|**14.00 ± 0.04**|**6.49 ± 0.02**|17.58 ± 0.04|
> |GraphMaker|-|**2.94 ± 0.01**|**16.01 ± 0.04**|**7.63 ± 0.02**|**20.70 ± 0.03**|
> |ER|-|**3.83 ± 0.01**|**20.01 ± 0.03**|**10.21 ± 0.02**|**24.53 ± 0.03**|
> |BTER|-|**1.75 ± 0.01**|**7.03 ± 0.04**|**3.96 ± 0.02**|**8.26 ± 0.04**|
> |mKPGM|-|**2.06 ± 0.01**|**9.14 ± 0.04**|**4.77 ± 0.02**|**9.26 ± 0.04**|
>
> **Table 3.** yelp dataset. Values for Clus scaled by $10^{-2}$, values for Tri scaled by $10^{-4}$.
>
> |Method|Config|Clus RMSE|Clus MAE|Clus Bias|Tri RMSE|Tri MAE|Tri Bias|
> |---|---|---|---|---|---|---|---|
> |SyNG-D|2|2.56|2.52|-2.52|**1.24**|**1.08**|**-1.05**|
> | |3|2.65|2.61|-2.61|1.71|1.57|-1.56|
> | |4|2.01|1.96|-1.96|1.59|1.47|-1.46|
> | |5|1.83|1.77|-1.77|1.82|1.72|-1.71|
> | |6|**1.77**|**1.70**|**-1.70**|2.00|1.89|-1.88|
> |SyNG-D(MLP)|2|2.35|2.31|-2.31|**0.74**|**0.61**|**-0.22**|
> | |3|1.05|0.96|-0.96|1.40|1.18|1.15|
> | |4|0.75|0.63|0.58|3.33|3.18|3.18|
> | |5|0.81|0.71|-0.68|0.98|0.84|-0.72|
> | |6|**0.65**|**0.53**|**0.47**|1.44|1.26|1.23|
> |SyNG-R|2|2.79|2.76|-2.76|1.49|1.35|-1.34|
> | |3|2.40|2.36|-2.36|1.33|1.17|-1.15|
> | |4|1.56|1.50|-1.50|1.01|0.85|-0.73|
> | |5|0.99|0.90|-0.89|0.83|0.69|-0.43|
> | |6|**0.76**|**0.65**|**-0.61**|**0.78**|**0.64**|**-0.30**|
> |VGAE|2|10.36|10.36|-10.36|**7.36**|**7.36**|**-7.36**|
> | |3|10.55|10.55|-10.55|7.45|7.45|-7.45|
> | |4|10.56|10.56|-10.56|7.46|7.46|-7.46|
> | |5|10.77|10.77|-10.77|7.49|7.49|-7.49|
> | |6|11.26|11.26|-11.26|7.67|7.67|-7.67|
> | |16|**10.25**|**10.25**|**-10.25**|7.41|7.41|-7.41|
> |GraphMaker|-|**15.52**|**15.52**|**-15.52**|**8.82**|**8.82**|**-8.82**|
> |ER|-|**14.48**|**14.48**|**-14.48**|**8.12**|**8.12**|**-8.12**|
> |BTER|-|**4.45**|**4.45**|**-4.45**|**2.27**|**2.27**|**-2.27**|
> |mKPGM|-|**15.98**|**15.98**|**-15.98**|**9.35**|**9.35**|**-9.35**|

---

> > ### Author Response · Authors · 2025-11-25
> >
> > **Table 4.** Polblogs dataset. Degree centrality. All entries scaled by $10^{-2}$.
> >
> > |Method|Config|W1 dist.|KS dist.|Energy dist.|MMD|
> > |---|---|---|---|---|---|
> > |SyNG-D|2|**0.18 ± 0.08**|**4.53 ± 0.97**|**0.01 ± 0.01**|**0.97 ± 1.44**|
> > | |3|0.22 ± 0.11|5.02 ± 1.34|**0.01 ± 0.01**|1.95 ± 2.15|
> > | |4|0.28 ± 0.13|5.74 ± 1.64|0.02 ± 0.02|3.18 ± 2.50|
> > | |5|0.44 ± 0.14|8.29 ± 2.04|0.04 ± 0.02|6.49 ± 2.44|
> > | |6|0.52 ± 0.15|9.98 ± 2.11|0.06 ± 0.03|8.31 ± 2.50|
> > |SyNG-D(MLP)|2|0.20 ± 0.07|9.59 ± 1.21|**0.01 ± 0.01**|1.47 ± 1.71|
> > | |3|0.24 ± 0.10|9.81 ± 1.09|**0.01 ± 0.01**|3.05 ± 2.25|
> > | |4|0.33 ± 0.15|10.06 ± 1.49|0.02 ± 0.02|3.20 ± 2.26|
> > | |5|0.23 ± 0.07|**8.99 ± 0.93**|**0.01 ± 0.01**|2.79 ± 1.77|
> > | |6|**0.19 ± 0.08**|9.16 ± 1.35|**0.01 ± 0.01**|**1.41 ± 1.73**|
> > |SyNG-R|2|**0.18 ± 0.10**|**4.47 ± 0.92**|**0.01 ± 0.01**|**0.92 ± 1.45**|
> > | |3|**0.18 ± 0.10**|4.49 ± 1.11|**0.01 ± 0.01**|0.93 ± 1.50|
> > | |4|**0.18 ± 0.10**|4.57 ± 1.23|**0.01 ± 0.01**|0.94 ± 1.50|
> > | |5|**0.18 ± 0.10**|4.68 ± 1.30|**0.01 ± 0.01**|1.06 ± 1.62|
> > | |6|**0.18 ± 0.10**|4.88 ± 1.49|**0.01 ± 0.01**|1.12 ± 1.67|
> > |VGAE|2|0.93 ± 0.01|35.38 ± 0.49|0.24 ± 0.01|35.42 ± 0.61|
> > | |3|0.97 ± 0.01|36.24 ± 0.53|0.26 ± 0.01|37.16 ± 0.51|
> > | |4|0.94 ± 0.01|35.65 ± 0.52|0.24 ± 0.01|36.02 ± 0.60|
> > | |5|**0.92 ± 0.01**|**35.31 ± 0.51**|**0.23 ± 0.01**|**35.33 ± 0.64**|
> > | |6|0.97 ± 0.01|36.35 ± 0.53|0.26 ± 0.01|37.42 ± 0.57|
> > | |16|0.98 ± 0.01|36.56 ± 0.49|0.26 ± 0.01|37.82 ± 0.56|
> > |GRAN|128|**0.56 ± 0.19**|**15.88 ± 2.70**|**0.08 ± 0.05**|**13.86 ± 2.01**|
> > | |256|2.25 ± 0.31|35.58 ± 3.15|0.64 ± 0.15|33.78 ± 3.61|
> > | |512|9.30 ± 0.67|63.37 ± 2.38|6.36 ± 0.72|67.41 ± 2.69|
> > |EDGE|-|**0.06 ± 0.00**|**4.91 ± 0.44**|**0.00 ± 0.00**|**0.00 ± 0.00**|
> > |GraphMaker|-|**1.72 ± 0.01**|**49.48 ± 0.60**|**0.82 ± 0.01**|**74.15 ± 0.75**|
> > |ER|-|**1.83 ± 0.01**|**56.70 ± 0.56**|**1.04 ± 0.01**|**79.24 ± 0.59**|
> > |BTER|-|**0.06 ± 0.01**|**5.69 ± 0.62**|**0.00 ± 0.00**|**0.00 ± 0.00**|
> > |mKPGM|-|**1.57 ± 0.01**|**33.09 ± 0.37**|**0.60 ± 0.01**|**50.67 ± 0.90**|
> >
> > **Table 5.** Polblogs dataset. Eigenvalues. W1 unscaled, KS, Energy, MMD scaled by $10^{-2}$.
> >
> > |Method|Config|W1 dist.|KS dist.|Energy dist.|MMD|
> > |---|---|---|---|---|---|
> > |SyNG-D|2|0.21 ± 0.06|4.82 ± 0.94|0.92 ± 0.51|2.58 ± 1.37|
> > | |3|**0.20 ± 0.04**|5.02 ± 1.03|0.90 ± 0.38|3.15 ± 1.16|
> > | |4|**0.20 ± 0.05**|4.41 ± 0.96|**0.74 ± 0.32**|2.52 ± 1.22|
> > | |5|0.25 ± 0.07|**3.22 ± 0.77**|0.80 ± 0.46|1.51 ± 1.18|
> > | |6|0.30 ± 0.09|3.32 ± 0.89|1.19 ± 0.72|**1.43 ± 1.36**|
> > |SyNG-D(MLP)|2|0.46 ± 0.10|7.00 ± 0.89|3.36 ± 1.34|5.54 ± 1.05|
> > | |3|**0.37 ± 0.09**|7.01 ± 0.76|**2.51 ± 1.04**|**5.27 ± 0.90**|
> > | |4|0.65 ± 0.11|8.90 ± 0.90|6.88 ± 1.98|8.17 ± 1.11|
> > | |5|0.38 ± 0.10|7.00 ± 0.89|2.81 ± 1.18|5.57 ± 1.08|
> > | |6|0.39 ± 0.10|**6.74 ± 0.92**|2.80 ± 1.18|5.35 ± 1.12|
> > |SyNG-R|2|0.24 ± 0.08|**5.04 ± 1.10**|**1.24 ± 0.82**|**3.01 ± 1.59**|
> > | |3|0.23 ± 0.07|5.45 ± 1.02|1.33 ± 0.76|3.64 ± 1.34|
> > | |4|0.23 ± 0.07|5.58 ± 1.06|1.35 ± 0.79|3.86 ± 1.32|
> > | |5|**0.22 ± 0.07**|5.68 ± 0.99|1.34 ± 0.77|3.98 ± 1.20|
> > | |6|**0.22 ± 0.06**|5.69 ± 0.98|1.36 ± 0.73|4.15 ± 1.17|
> > |VGAE|2|**1.23 ± 0.01**|27.33 ± 0.18|42.20 ± 0.72|31.11 ± 0.20|
> > | |3|1.27 ± 0.01|27.67 ± 0.18|44.05 ± 0.82|31.57 ± 0.21|
> > | |4|1.24 ± 0.01|27.41 ± 0.18|42.65 ± 0.74|31.23 ± 0.20|
> > | |5|**1.23 ± 0.01**|**27.27 ± 0.18**|**41.88 ± 0.75**|**31.10 ± 0.22**|
> > | |6|1.27 ± 0.01|27.72 ± 0.18|44.16 ± 0.74|31.62 ± 0.19|
> > | |16|1.28 ± 0.01|27.81 ± 0.18|44.70 ± 0.77|31.80 ± 0.20|
> > |GRAN|128|**0.48 ± 0.18**|10.91 ± 1.15|**4.63 ± 2.88**|**9.04 ± 0.98**|
> > | |256|1.23 ± 0.16|**10.89 ± 1.18**|17.03 ± 4.31|9.39 ± 1.26|
> > | |512|3.52 ± 0.19|24.55 ± 1.24|122.93 ± 12.78|26.84 ± 1.41|
> > |EDGE|-|**0.28 ± 0.04**|**8.21 ± 1.17**|**1.90 ± 0.59**|**4.69 ± 1.26**|
> > |GraphMaker|-|**1.78 ± 0.01**|**32.20 ± 0.15**|**72.49 ± 0.85**|**38.28 ± 0.13**|
> > |ER|-|**2.03 ± 0.01**|**34.42 ± 0.12**|**93.20 ± 0.92**|**40.10 ± 0.11**|
> > |BTER|-|**0.41 ± 0.02**|**4.53 ± 0.40**|**1.87 ± 0.17**|**3.04 ± 0.34**|
> > |mKPGM|-|**1.31 ± 0.01**|**23.36 ± 0.22**|**31.95 ± 0.46**|**31.15 ± 0.22**|

---

> > > ### Author Response · Authors · 2025-11-25
> > >
> > > **Table 6.** Polblogs dataset. Values for Clus scaled by $10^{-2}$, values for Tri scaled by $10^{-4}$.
> > >
> > > |Method|Config|Clus RMSE|Clus MAE|Clus Bias|Tri RMSE|Tri MAE|Tri Bias|
> > > |---|---|---|---|---|---|---|---|
> > > |SyNG-D|2|1.90|1.51|1.15|0.71|**0.55**|**0.13**|
> > > | |3|**1.56**|**1.23**|0.54|**0.68**|0.56|-0.33|
> > > | |4|1.85|1.45|0.90|0.78|0.66|-0.43|
> > > | |5|1.66|1.33|**0.49**|1.09|1.00|-0.98|
> > > | |6|1.91|1.49|0.93|1.19|1.09|-1.07|
> > > |SyNG-D(MLP)|2|3.58|3.26|3.23|0.91|0.69|0.56|
> > > | |3|**2.23**|**1.88**|**1.73**|**0.59**|**0.48**|**-0.16**|
> > > | |4|5.69|5.45|5.45|1.80|1.61|1.60|
> > > | |5|5.18|4.77|4.74|1.15|0.90|0.77|
> > > | |6|3.15|2.80|2.72|0.84|0.65|0.50|
> > > |SyNG-R|2|**2.45**|**2.00**|**1.83**|**0.83**|**0.62**|**0.39**|
> > > | |3|2.68|2.23|2.12|0.85|0.64|0.44|
> > > | |4|2.88|2.44|2.37|0.87|0.66|0.49|
> > > | |5|2.91|2.50|2.44|0.89|0.68|0.51|
> > > | |6|3.13|2.74|2.69|0.92|0.71|0.55|
> > > |VGAE|2|3.72|3.71|-3.71|2.10|2.10|-2.10|
> > > | |3|4.49|4.49|-4.49|2.20|2.20|-2.20|
> > > | |4|4.03|4.02|-4.02|2.14|2.14|-2.14|
> > > | |5|**3.26**|**3.25**|**-3.25**|**2.07**|**2.07**|**-2.07**|
> > > | |6|4.87|4.87|-4.87|2.22|2.22|-2.22|
> > > | |16|4.54|4.54|-4.54|2.22|2.22|-2.22|
> > > |GRAN|128|11.42|11.38|-11.38|**1.57**|**1.49**|**-1.41**|
> > > | |256|10.82|10.78|-10.78|4.42|4.22|4.22|
> > > | |512|**0.76**|**0.60**|**0.03**|64.94|64.45|64.45|
> > > |EDGE|10000|**5.69**|**5.43**|**-5.43**|**0.79**|**0.75**|**-0.75**|
> > > |GraphMaker|-|**20.75**|**20.75**|**-20.75**|**3.27**|**3.27**|**-3.27**|
> > > |ER|-|**20.36**|**20.36**|**-20.36**|**3.22**|**3.22**|**-3.22**|
> > > |BTER|-|**5.27**|**5.27**|**-5.27**|**0.88**|**0.88**|**-0.88**|
> > > |mKPGM|-|**21.43**|**21.43**|**-21.43**|**3.32**|**3.32**|**-3.32**|
> > >
> > > **Table 7.** DBLP dataset. Degree centrality. KS dist and MMD are scaled by $10^{−1}$. W1 dist and Energy dist are scaled by $10^{−2}$.
> > >
> > > |Method|Config|W1 dist.|KS dist.|Energy dist.|MMD|
> > > |---|---|---|---|---|---|
> > > |SyNG-D|2|0.19 ± 0.08|**0.75 ± 0.17**|**0.02 ± 0.01**|**0.93 ± 0.23**|
> > > | |3|**0.18 ± 0.06**|0.91 ± 0.16|**0.02 ± 0.01**|1.03 ± 0.19|
> > > | |4|0.22 ± 0.07|1.22 ± 0.14|0.03 ± 0.01|1.29 ± 0.17|
> > > | |5|0.33 ± 0.09|1.73 ± 0.18|0.05 ± 0.02|1.78 ± 0.20|
> > > | |6|0.44 ± 0.10|2.19 ± 0.18|0.09 ± 0.02|2.22 ± 0.20|
> > > |SyNG-D(MLP)|2|**0.31 ± 0.07**|1.81 ± 0.17|**0.04 ± 0.01**|1.50 ± 0.15|
> > > | |3|0.42 ± 0.06|1.33 ± 0.13|0.07 ± 0.02|1.67 ± 0.18|
> > > | |4|0.47 ± 0.15|**1.19 ± 0.19**|0.06 ± 0.03|**1.40 ± 0.25**|
> > > | |5|0.47 ± 0.10|1.36 ± 0.14|0.07 ± 0.02|1.72 ± 0.16|
> > > | |6|0.35 ± 0.06|1.53 ± 0.18|0.06 ± 0.01|1.73 ± 0.19|
> > > |SyNG-R|2|**0.15 ± 0.07**|**0.72 ± 0.18**|**0.01 ± 0.01**|**0.75 ± 0.28**|
> > > | |3|**0.15 ± 0.07**|0.73 ± 0.18|**0.01 ± 0.01**|**0.75 ± 0.28**|
> > > | |4|**0.15 ± 0.07**|0.73 ± 0.18|**0.01 ± 0.01**|0.77 ± 0.28|
> > > | |5|0.16 ± 0.07|0.74 ± 0.18|**0.01 ± 0.01**|0.78 ± 0.27|
> > > | |6|0.16 ± 0.07|0.75 ± 0.17|**0.01 ± 0.01**|0.80 ± 0.27|
> > > |VGAE|2|0.49 ± 0.01|**2.33 ± 0.08**|0.10 ± 0.00|2.84 ± 0.10|
> > > | |3|**0.29 ± 0.00**|2.65 ± 0.07|**0.09 ± 0.00**|**2.80 ± 0.06**|
> > > | |4|0.35 ± 0.01|3.20 ± 0.08|0.12 ± 0.00|3.28 ± 0.06|
> > > | |5|0.35 ± 0.01|3.16 ± 0.08|0.12 ± 0.00|3.23 ± 0.07|
> > > | |6|0.32 ± 0.00|2.96 ± 0.07|0.11 ± 0.00|3.01 ± 0.05|
> > > | |16|0.32 ± 0.00|2.96 ± 0.08|0.10 ± 0.00|2.96 ± 0.06|
> > > |GRAN|dim=128|1.48 ± 0.25|3.78 ± 0.88|0.41 ± 0.14|5.13 ± 0.96|
> > > | |dim=256|1.25 ± 0.03|4.61 ± 0.12|0.54 ± 0.02|6.12 ± 0.14|
> > > | |dim=512|**1.06 ± 0.01**|**2.21 ± 0.08**|**0.31 ± 0.01**|**2.97 ± 0.09**|
> > > |EDGE|-|**0.23 ± 0.12**|**0.79 ± 0.11**|**0.02 ± 0.02**|**0.99 ± 0.23**|
> > > |GraphMaker|-|**1.37 ± 0.01**|**5.56 ± 0.07**|**0.68 ± 0.01**|**7.51 ± 0.07**|
> > > |ER|-|**1.57 ± 0.01**|**6.63 ± 0.07**|**1.02 ± 0.02**|**8.72 ± 0.06**|
> > > |BTER|-|**0.08 ± 0.01**|**0.56 ± 0.05**|**0.00 ± 0.00**|**0.46 ± 0.05**|
> > > |mKPGM|-|**1.13 ± 0.01**|**3.25 ± 0.08**|**0.37 ± 0.00**|**4.57 ± 0.08**|

---

> > > > ### Author Response · Authors · 2025-11-25
> > > >
> > > > **Table 8.** DBLP dataset. Eigenvalues. Values for Energy dist are scaled by $10^{-2}$, and W1 dist, KS dist, and MMD are scaled by $10^{-1}$.
> > > >
> > > > |Method|Config|W1 dist.|KS dist.|Energy dist.|MMD|
> > > > |---|---|---|---|---|---|
> > > > |SyNG-D|2|3.02 ± 0.32|**0.81 ± 0.06**|2.35 ± 0.43|0.88 ± 0.06|
> > > > | |3|2.13 ± 0.28|0.91 ± 0.07|**1.57 ± 0.21**|**0.77 ± 0.05**|
> > > > | |4|**1.73 ± 0.17**|1.06 ± 0.07|1.65 ± 0.20|**0.77 ± 0.05**|
> > > > | |5|2.56 ± 0.43|1.32 ± 0.08|2.99 ± 0.57|0.96 ± 0.08|
> > > > | |6|3.80 ± 0.46|1.63 ± 0.08|5.62 ± 0.89|1.28 ± 0.08|
> > > > |SyNG-D(MLP)|2|**2.85 ± 0.34**|1.10 ± 0.07|**2.37 ± 0.31**|**0.93 ± 0.05**|
> > > > | |3|6.09 ± 0.47|1.10 ± 0.08|8.94 ± 1.27|1.34 ± 0.07|
> > > > | |4|4.46 ± 0.41|**0.83 ± 0.05**|5.05 ± 0.87|1.12 ± 0.07|
> > > > | |5|5.75 ± 0.46|0.97 ± 0.07|7.42 ± 1.06|1.32 ± 0.06|
> > > > | |6|5.92 ± 0.45|1.03 ± 0.08|8.06 ± 1.13|1.31 ± 0.06|
> > > > |SyNG-R|2|2.89 ± 0.33|0.63 ± 0.05|2.02 ± 0.50|0.80 ± 0.07|
> > > > | |3|2.25 ± 0.33|0.58 ± 0.05|1.35 ± 0.39|0.65 ± 0.07|
> > > > | |4|1.70 ± 0.29|0.50 ± 0.06|0.83 ± 0.25|0.49 ± 0.07|
> > > > | |5|1.31 ± 0.30|0.46 ± 0.05|0.52 ± 0.18|0.37 ± 0.07|
> > > > | |6|**1.19 ± 0.32**|**0.41 ± 0.05**|**0.40 ± 0.15**|**0.27 ± 0.07**|
> > > > |VGAE|2|12.38 ± 0.16|2.35 ± 0.03|37.67 ± 1.01|2.62 ± 0.03|
> > > > | |3|4.04 ± 0.06|0.98 ± 0.03|3.47 ± 0.17|1.20 ± 0.04|
> > > > | |4|3.83 ± 0.10|0.67 ± 0.03|2.81 ± 0.18|0.68 ± 0.03|
> > > > | |5|3.76 ± 0.12|0.65 ± 0.03|2.69 ± 0.21|0.65 ± 0.03|
> > > > | |6|**3.51 ± 0.08**|**0.61 ± 0.02**|**2.30 ± 0.13**|**0.64 ± 0.03**|
> > > > | |16|3.57 ± 0.09|0.63 ± 0.03|2.48 ± 0.16|0.65 ± 0.03|
> > > > |GRAN|dim=128|**6.40 ± 1.61**|1.61 ± 0.20|14.92 ± 6.20|1.87 ± 0.22|
> > > > | |dim=256|17.55 ± 0.48|3.06 ± 0.05|75.24 ± 3.29|3.54 ± 0.06|
> > > > | |dim=512|6.95 ± 0.24|**1.49 ± 0.05**|**11.81 ± 1.00**|**1.76 ± 0.07**|
> > > > |EDGE|-|**5.50 ± 1.34**|**1.22 ± 0.25**|**9.72 ± 4.23**|**1.39 ± 0.30**|
> > > > |GraphMaker|-|**18.39 ± 0.14**|**3.32 ± 0.02**|**85.40 ± 1.14**|**3.93 ± 0.02**|
> > > > |ER|-|**22.55 ± 0.14**|**3.77 ± 0.02**|**121.43 ± 1.26**|**4.41 ± 0.02**|
> > > > |BTER|-|**9.44 ± 0.11**|**1.14 ± 0.02**|**14.67 ± 0.35**|**1.48 ± 0.03**|
> > > > |mKPGM|-|**13.13 ± 0.13**|**2.52 ± 0.02**|**43.78 ± 0.90**|**3.00 ± 0.03**|
> > > >
> > > > **Table 9.** DBLP dataset. Values for Clus scaled by $10^{-1}$, values for Tri scaled by $10^{-4}$.
> > > >
> > > > |Method|Config|Clus RMSE|Clus MAE|Clus Bias|Tri RMSE|Tri MAE|Tri Bias|
> > > > |---|---|---|---|---|---|---|---|
> > > > |SyNG-D|2|0.62|0.59|-0.59|**1.58**|**1.26**|**0.46**|
> > > > | |3|0.53|**0.49**|**-0.49**|1.61|1.37|-0.87|
> > > > | |4|0.55|0.52|-0.52|1.81|1.57|-1.39|
> > > > | |5|**0.53**|0.49|-0.49|2.34|2.13|-2.07|
> > > > | |6|0.57|0.54|-0.54|3.01|2.86|-2.86|
> > > > |SyNG-D(MLP)|2|1.40|1.36|-1.36|2.91|2.67|-2.62|
> > > > | |3|2.42|2.39|-2.39|3.74|3.64|-3.64|
> > > > | |4|**1.01**|**0.99**|**-0.99**|3.63|3.20|3.13|
> > > > | |5|1.44|1.41|-1.41|2.29|1.81|1.47|
> > > > | |6|1.71|1.69|-1.69|**1.80**|**1.54**|**-1.37**|
> > > > |SyNG-R|2|0.38|0.33|-0.33|**1.43**|**1.11**|**0.11**|
> > > > | |3|0.32|0.27|-0.26|1.44|1.12|0.19|
> > > > | |4|0.26|0.21|-0.20|1.45|1.12|0.25|
> > > > | |5|0.23|0.18|-0.15|1.46|1.13|0.28|
> > > > | |6|**0.21**|**0.17**|**-0.13**|1.46|1.13|0.31|
> > > > |VGAE|2|6.33|6.33|-6.33|6.65|6.65|-6.65|
> > > > | |3|**0.14**|**0.13**|**-0.13**|0.58|0.58|-0.58|
> > > > | |4|0.29|0.29|0.29|0.28|0.28|0.28|
> > > > | |5|0.30|0.30|0.30|0.26|0.26|0.26|
> > > > | |6|0.33|0.33|0.33|0.11|0.10|0.10|
> > > > | |16|0.31|0.31|0.31|**0.05**|**0.05**|**0.05**|
> > > > |GRAN|128|**8.76**|**8.76**|**-8.76**|**6.78**|**6.73**|**-6.73**|
> > > > | |256|8.89|8.89|-8.89|7.93|7.93|-7.93|
> > > > | |512|8.92|8.92|-8.92|7.99|7.99|-7.99|
> > > > |EDGE|10000|**1.33**|**1.09**|**-1.09**|**2.20**|**1.77**|**-1.77**|
> > > > |GraphMaker|-|**8.98**|**8.98**|**-8.98**|**7.98**|**7.98**|**-7.98**|
> > > > |ER|-|**8.94**|**8.94**|**-8.94**|**7.96**|**7.96**|**-7.96**|
> > > > |BTER|-|**6.77**|**6.77**|**-6.77**|**6.09**|**6.09**|**-6.09**|
> > > > |mKPGM|-|**9.00**|**9.00**|**-9.00**|**8.00**|**8.00**|**-8.00**|

---

> > > > > ### Author Response · Authors · 2025-11-25
> > > > >
> > > > > **Table 10.** YouTube dataset. Degree centrality. Values for W1 distance, KS distance, and Energy distance are scaled by $10^{-2}$, and MMD values are scaled by $10^{-1}$.
> > > > >
> > > > > |Method|Config|W1 dist.|KS dist.|Energy dist.|MMD|
> > > > > |---|---|---|---|---|---|
> > > > > |SyNG-D|2|**0.15 ± 0.07**|**3.76 ± 1.43**|**0.01 ± 0.01**|**0.21 ± 0.18**|
> > > > > | |3|0.23 ± 0.09|5.61 ± 1.67|**0.01 ± 0.01**|0.45 ± 0.20|
> > > > > | |4|0.32 ± 0.10|7.42 ± 1.91|0.03 ± 0.02|0.67 ± 0.21|
> > > > > | |5|0.49 ± 0.09|11.79 ± 1.78|0.07 ± 0.02|1.15 ± 0.19|
> > > > > | |6|0.58 ± 0.09|13.92 ± 1.87|0.10 ± 0.03|1.38 ± 0.20|
> > > > > |SyNG-D(MLP)|2|0.19 ± 0.05|6.05 ± 1.42|**0.01 ± 0.00**|0.42 ± 0.15|
> > > > > | |3|**0.18 ± 0.05**|5.91 ± 1.29|**0.01 ± 0.00**|0.41 ± 0.14|
> > > > > | |4|**0.18 ± 0.05**|**5.88 ± 1.51**|**0.01 ± 0.01**|**0.39 ± 0.17**|
> > > > > | |5|0.22 ± 0.05|8.00 ± 1.57|**0.01 ± 0.01**|0.63 ± 0.15|
> > > > > | |6|0.29 ± 0.08|9.82 ± 1.64|0.03 ± 0.01|0.81 ± 0.18|
> > > > > |SyNG-R|2|**0.13 ± 0.06**|**2.86 ± 1.13**|**0.00 ± 0.00**|**0.11 ± 0.14**|
> > > > > | |3|**0.13 ± 0.06**|2.91 ± 1.16|**0.00 ± 0.00**|0.12 ± 0.15|
> > > > > | |4|**0.13 ± 0.06**|2.91 ± 1.11|**0.00 ± 0.00**|0.12 ± 0.15|
> > > > > | |5|**0.13 ± 0.06**|2.98 ± 1.16|**0.00 ± 0.00**|**0.11 ± 0.15**|
> > > > > | |6|**0.13 ± 0.06**|3.00 ± 1.21|**0.00 ± 0.00**|0.12 ± 0.15|
> > > > > |VGAE|2|0.89 ± 0.01|23.74 ± 0.44|0.20 ± 0.00|3.39 ± 0.05|
> > > > > | |3|0.89 ± 0.01|23.49 ± 0.40|0.20 ± 0.00|3.34 ± 0.05|
> > > > > | |4|**0.80 ± 0.01**|**20.10 ± 0.43**|**0.15 ± 0.00**|**2.78 ± 0.04**|
> > > > > | |5|0.91 ± 0.01|24.40 ± 0.39|0.21 ± 0.00|3.50 ± 0.04|
> > > > > | |6|0.82 ± 0.01|20.33 ± 0.43|0.16 ± 0.00|2.88 ± 0.04|
> > > > > | |16|0.85 ± 0.01|21.76 ± 0.43|0.18 ± 0.00|3.10 ± 0.04|
> > > > > |GRAN|128|0.78 ± 0.38|15.38 ± 5.47|0.12 ± 0.17|1.60 ± 0.42|
> > > > > | |256|5.71 ± 0.41|57.59 ± 3.00|3.87 ± 0.48|6.20 ± 0.32|
> > > > > | |512|**0.76 ± 0.20**|**9.24 ± 2.00**|**0.08 ± 0.04**|**0.79 ± 0.17**|
> > > > > |EDGE|-|**0.41 ± 0.05**|**7.82 ± 0.86**|**0.04 ± 0.01**|**0.78 ± 0.10**|
> > > > > |GraphMaker|-|**1.35 ± 0.00**|**37.68 ± 0.47**|**0.55 ± 0.01**|**6.50 ± 0.05**|
> > > > > |ER|-|**1.41 ± 0.01**|**47.08 ± 0.44**|**0.67 ± 0.01**|**6.92 ± 0.04**|
> > > > > |BTER|-|**0.04 ± 0.01**|**1.64 ± 0.32**|**0.00 ± 0.00**|**0.00 ± 0.00**|
> > > > > |mKPGM|-|**1.06 ± 0.01**|**31.71 ± 0.39**|**0.33 ± 0.01**|**4.49 ± 0.04**|
> > > > >
> > > > > **Table 11.** YouTube dataset. Eigenvalues. W1 dist unscaled; KS dist, Energy dist, and MMD are scaled by $10^{-1}$.
> > > > >
> > > > > |Method|Config|W1 dist.|KS dist.|Energy dist.|MMD|
> > > > > |---|---|---|---|---|---|
> > > > > |SyNG-D|2|0.31 ± 0.06|0.30 ± 0.07|0.11 ± 0.05|0.26 ± 0.09|
> > > > > | |3|0.23 ± 0.04|0.20 ± 0.06|0.05 ± 0.03|0.08 ± 0.10|
> > > > > | |4|**0.19 ± 0.04**|**0.14 ± 0.04**|**0.03 ± 0.01**|**0.01 ± 0.03**|
> > > > > | |5|0.30 ± 0.08|0.20 ± 0.06|0.07 ± 0.04|0.02 ± 0.05|
> > > > > | |6|0.40 ± 0.08|0.28 ± 0.07|0.14 ± 0.07|0.11 ± 0.11|
> > > > > |SyNG-D(MLP)|2|0.30 ± 0.06|0.28 ± 0.06|0.10 ± 0.05|0.24 ± 0.09|
> > > > > | |3|0.28 ± 0.06|0.27 ± 0.07|0.09 ± 0.05|0.20 ± 0.11|
> > > > > | |4|0.19 ± 0.05|0.18 ± 0.06|0.04 ± 0.03|0.04 ± 0.07|
> > > > > | |5|**0.17 ± 0.04**|**0.14 ± 0.04**|**0.03 ± 0.02**|0.01 ± 0.04|
> > > > > | |6|0.19 ± 0.06|**0.14 ± 0.04**|**0.03 ± 0.02**|**0.00 ± 0.01**|
> > > > > |SyNG-R|2|0.43 ± 0.07|0.44 ± 0.07|0.24 ± 0.09|0.45 ± 0.09|
> > > > > | |3|0.38 ± 0.06|0.39 ± 0.07|0.19 ± 0.07|0.39 ± 0.10|
> > > > > | |4|0.33 ± 0.06|0.36 ± 0.06|0.15 ± 0.06|0.34 ± 0.09|
> > > > > | |5|0.31 ± 0.06|0.36 ± 0.06|0.14 ± 0.06|0.32 ± 0.09|
> > > > > | |6|**0.29 ± 0.05**|**0.35 ± 0.06**|**0.13 ± 0.05**|**0.31 ± 0.09**|
> > > > > |VGAE|2|1.27 ± 0.01|1.51 ± 0.01|2.35 ± 0.04|1.85 ± 0.01|
> > > > > | |3|1.26 ± 0.01|1.51 ± 0.01|2.33 ± 0.04|1.84 ± 0.01|
> > > > > | |4|**1.15 ± 0.01**|**1.41 ± 0.01**|**1.94 ± 0.03**|**1.72 ± 0.01**|
> > > > > | |5|1.29 ± 0.01|1.53 ± 0.01|2.41 ± 0.04|1.87 ± 0.01|
> > > > > | |6|1.17 ± 0.01|1.42 ± 0.01|2.02 ± 0.03|1.74 ± 0.01|
> > > > > | |16|1.21 ± 0.01|1.46 ± 0.01|2.14 ± 0.04|1.79 ± 0.01|
> > > > > |GRAN|128|0.97 ± 0.46|**0.70 ± 0.29**|1.14 ± 1.25|**0.71 ± 0.32**|
> > > > > | |256|4.00 ± 0.23|2.40 ± 0.12|14.20 ± 1.53|2.58 ± 0.14|
> > > > > | |512|**0.94 ± 0.17**|0.76 ± 0.10|**0.99 ± 0.33**|0.79 ± 0.08|
> > > > > |EDGE|-|**0.34 ± 0.04**|**0.39 ± 0.08**|**0.14 ± 0.05**|**0.44 ± 0.07**|
> > > > > |GraphMaker|-|**1.62 ± 0.01**|**1.81 ± 0.01**|**3.68 ± 0.04**|**2.31 ± 0.01**|
> > > > > |ER|-|**1.95 ± 0.01**|**2.10 ± 0.01**|**5.54 ± 0.06**|**2.60 ± 0.01**|
> > > > > |BTER|-|**0.76 ± 0.01**|**0.60 ± 0.01**|**0.57 ± 0.02**|**0.64 ± 0.01**|
> > > > > |mKPGM|-|**1.72 ± 0.01**|**1.84 ± 0.01**|**4.19 ± 0.06**|**2.24 ± 0.01**|

---

> > > > > > ### Author Response · Authors · 2025-11-25
> > > > > >
> > > > > > **Table 12.** YouTube dataset. Values for Clus scaled by $10^{-2}$, values for Tri scaled by $10^{-4}$.
> > > > > >
> > > > > > | Method | Config | Clus RMSE | Clus MAE | Clus Bias | Tri RMSE | Tri MAE | Tri Bias |
> > > > > > | --- | --- | --- | --- | --- | --- | --- | --- |
> > > > > > | SyNG-D | 2 | 2.28 | 2.15 | -2.15 | 0.97 | 0.92 | 0.92 |
> > > > > > |  | 3 | 1.69 | 1.46 | -1.41 | 0.89 | 0.84 | 0.84 |
> > > > > > |  | 4 | **1.34** | **1.08** | **-0.95** | 0.83 | 0.78 | 0.78 |
> > > > > > |  | 5 | 1.50 | 1.22 | -1.09 | 0.58 | 0.52 | 0.52 |
> > > > > > |  | 6 | 1.42 | 1.19 | -1.01 | **0.41** | **0.35** | **0.35** |
> > > > > > | SyNG-D(MLP) | 2 | **1.12** | **0.88** | **-0.46** | 1.35 | 1.30 | 1.30 |
> > > > > > |  | 3 | 1.87 | 1.63 | 1.54 | 1.47 | 1.43 | 1.43 |
> > > > > > |  | 4 | 1.58 | 1.34 | 1.15 | 1.62 | 1.58 | 1.58 |
> > > > > > |  | 5 | 2.98 | 2.83 | 2.80 | 1.40 | 1.36 | 1.36 |
> > > > > > |  | 6 | 3.13 | 2.93 | 2.88 | **1.18** | **1.14** | **1.14** |
> > > > > > | SyNG-R | 2 | 2.23 | 2.08 | -2.07 | **1.15** | **1.11** | **1.11** |
> > > > > > |  | 3 | **1.07** | **0.87** | **-0.40** | 1.38 | 1.34 | 1.34 |
> > > > > > |  | 4 | 1.20 | 0.96 | 0.66 | 1.53 | 1.49 | 1.49 |
> > > > > > |  | 5 | 1.57 | 1.30 | 1.19 | 1.61 | 1.57 | 1.57 |
> > > > > > |  | 6 | 1.84 | 1.57 | 1.52 | 1.65 | 1.61 | 1.61 |
> > > > > > | VGAE | 2 | 11.80 | 11.80 | -11.80 | 0.81 | 0.81 | -0.81 |
> > > > > > |  | 3 | 11.66 | 11.66 | -11.66 | 0.80 | 0.80 | -0.80 |
> > > > > > |  | 4 | **8.00** | **8.00** | **-8.00** | **0.56** | **0.56** | **-0.56** |
> > > > > > |  | 5 | 12.07 | 12.07 | -12.07 | 0.83 | 0.83 | -0.83 |
> > > > > > |  | 6 | 9.24 | 9.24 | -9.24 | 0.65 | 0.65 | -0.65 |
> > > > > > |  | 16 | 9.87 | 9.87 | -9.87 | 0.69 | 0.69 | -0.69 |
> > > > > > | GRAN | 128 | 11.34 | 11.32 | -11.32 | **0.85** | **0.56** | **0.54** |
> > > > > > |  | 256 | **5.30** | **5.29** | **-5.29** | 15.05 | 14.95 | 14.95 |
> > > > > > |  | 512 | 10.11 | 10.10 | -10.10 | 1.13 | 1.08 | 1.08 |
> > > > > > | EDGE | 10000 | **14.45** | **13.81** | **-13.80** | **0.78** | **0.75** | **-0.50** |
> > > > > > | GraphMaker | - | **16.71** | **16.71** | **-16.71** | **1.12** | **1.12** | **-1.12** |
> > > > > > | ER | - | **16.30** | **16.30** | **-16.30** | **1.05** | **1.05** | **-1.05** |
> > > > > > | BTER | - | **9.73** | **9.73** | **-9.73** | **0.02** | **0.02** | **-0.01** |
> > > > > > | mKPGM | - | **16.00** | **16.00** | **-16.00** | **1.03** | **1.03** | **-1.03** |

---

> > > > > > > ### Author Response · Authors · 2025-11-25
> > > > > > >
> > > > > > > **Table 13.** Graphlet results across four datasets. BTER and mKPGM are included for comparison.
> > > > > > >
> > > > > > > |Method|Config|L1 dist.| | | |L2 dist.| | | |
> > > > > > > |:------|:----:|:----:|:----:|:----:|:----:|:----:|:----:|:----:|:----:|
> > > > > > > | | |YouTube|DBLP|PolBlogs|Yelp|YouTube|DBLP|PolBlogs|Yelp|
> > > > > > > |SyNG-D|2|1.42±0.91|3.80±0.96|0.76±0.39|0.38±0.09|0.70±0.48|1.97±0.51|0.33±0.19|0.16±0.05|
> > > > > > > | |3|**1.39±0.97**|**3.03±0.98**|**0.72±0.42**|0.42±0.11|**0.68±0.51**|**1.58±0.52**|**0.32±0.21**|0.17±0.05|
> > > > > > > | |4|1.45±0.98|3.35±0.86|0.86±0.46|**0.30±0.11**|0.71±0.50|1.75±0.45|0.38±0.23|**0.13±0.05**|
> > > > > > > | |5|1.45±0.97|3.17±0.90|0.89±0.52|0.32±0.13|0.71±0.50|1.67±0.47|0.40±0.26|0.14±0.06|
> > > > > > > | |6|1.69±1.05|3.46±0.94|0.97±0.51|0.31±0.13|0.83±0.54|1.83±0.49|0.43±0.25|0.14±0.06|
> > > > > > > |SyNG-D(MLP)|2|2.08±0.98|6.92±1.11|**1.19±0.52**|0.35±0.11|1.04±0.50|3.60±0.59|**0.51±0.26**|0.14±0.05|
> > > > > > > | |3|2.55±1.03|9.51±0.86|1.23±0.52|0.37±0.19|1.24±0.50|5.04±0.48|0.55±0.25|0.17±0.10|
> > > > > > > | |4|**1.71±0.86**|**6.17±0.74**|1.47±0.38|0.34±0.13|**0.82±0.43**|**3.19±0.39**|0.61±0.18|0.13±0.05|
> > > > > > > | |5|3.02±0.96|7.22±0.86|1.41±0.55|**0.23±0.14**|1.45±0.46|3.73±0.45|0.56±0.26|**0.11±0.07**|
> > > > > > > | |6|2.68±1.14|7.86±0.73|1.37±0.74|0.27±0.11|1.28±0.55|4.08±0.39|0.60±0.38|0.11±0.05|
> > > > > > > |SyNG-R|2|1.38±1.01|1.91±0.91|**0.81±0.39**|0.39±0.11|0.67±0.53|0.99±0.51|**0.35±0.19**|0.15±0.04|
> > > > > > > | |3|**1.33±1.05**|1.60±0.89|0.83±0.38|0.33±0.11|0.65±0.53|0.83±0.49|0.36±0.18|0.13±0.04|
> > > > > > > | |4|1.34±1.07|1.31±0.80|0.86±0.38|0.23±0.12|**0.65±0.53**|0.67±0.44|0.36±0.18|0.09±0.05|
> > > > > > > | |5|1.36±1.07|**1.13±0.72**|0.87±0.39|0.18±0.11|0.65±0.52|**0.58±0.39**|0.37±0.18|0.08±0.05|
> > > > > > > | |6|1.37±1.08|1.14±0.81|0.88±0.37|**0.18±0.11**|0.66±0.53|0.58±0.44|0.37±0.18|**0.08±0.05**|
> > > > > > > |VGAE|16|6.30±0.03|**1.51±0.11**|3.37±0.08|3.93±0.01|2.92±0.01|0.80±0.07|1.62±0.04|1.64±0.00|
> > > > > > > | |2|6.53±0.04|13.69±0.01|**2.95±0.09**|**3.66±0.01**|2.97±0.01|7.39±0.01|**1.45±0.04**|**1.53±0.00**|
> > > > > > > | |3|6.50±0.04|2.54±0.13|3.21±0.08|3.85±0.01|2.96±0.01|1.14±0.09|1.54±0.04|1.61±0.00|
> > > > > > > | |4|**5.91±0.03**|1.57±0.12|3.08±0.08|3.86±0.01|**2.79±0.01**|**0.68±0.06**|1.50±0.04|1.61±0.00|
> > > > > > > | |5|6.61±0.03|1.57±0.14|3.00±0.08|3.80±0.01|3.00±0.01|0.69±0.07|1.48±0.04|1.58±0.00|
> > > > > > > | |6|6.20±0.03|1.69±0.15|3.19±0.09|4.09±0.01|2.88±0.01|0.85±0.07|1.51±0.04|1.70±0.00|
> > > > > > > |GRAN|128|1.90±0.30|18.35±0.14|**1.95±0.33**|-|0.77±0.14|9.92±0.04|**0.79±0.19**|-|
> > > > > > > | |256|2.42±0.18|17.26±0.15|3.88±0.76|-|1.07±0.10|8.49±0.53|1.73±0.37|-|
> > > > > > > | |512|**1.67±0.38**|**17.17±0.08**|3.04±0.30|-|**0.67±0.17**|**8.22±0.31**|1.40±0.16|-|
> > > > > > > |EDGE|-|**5.35±0.49**|**5.27±2.28**|**1.14±0.29**|-|**2.12±0.19**|**2.57±1.35**|**0.45±0.11**|-|
> > > > > > > |GraphMaker|-|**7.97±0.00**|**17.45±0.01**|**7.88±0.01**|**5.17±0.00**|**3.48±0.00**|**8.32±0.00**|**3.23±0.00**|**2.14±0.00**|
> > > > > > > |ER|-|**7.90±0.00**|**17.39±0.01**|**7.80±0.01**|**5.02±0.00**|**3.47±0.00**|**8.32±0.00**|**3.22±0.00**|**2.08±0.00**|
> > > > > > > |BTER|-|**2.24±0.10**|**13.84±0.01**|**2.09±0.09**|**0.73±0.01**|**0.95±0.05**|**7.54±0.00**|**0.98±0.04**|**0.30±0.01**|
> > > > > > > |mKPGM|-|**7.57±0.01**|**17.47±0.01**|**7.83±0.03**|**4.26±0.01**|**3.32±0.01**|**8.28±0.00**|**3.09±0.01**|**1.74±0.01**|

---

> ### Author Response · Authors · 2025-11-25
>
> **Your comment 8:** "Figure 2 is misleading; a network can be shown in different forms using different visualization algorithms. This should not be used for comparisons."
>
> **Response:** Thanks for this insightful comment. We agree that using visualizations for comparison can be misleading, as it depends heavily on the specific visualization algorithm. Nevertheless, visualizing the networks generated by different methods provides a straightforward presentation in the main text and has been standard practice in the literature [7,8].
>
> In this revision, we first emphasize the visualization algorithm used: the Fruchterman–Reingold force-directed layout, which is also adopted in the aforementioned literature. This force-directed layout is widely used and is generally considered informative for displaying global structural patterns such as clustering, degree concentration, and community mixing.
>
> In the empirical section of the updated manuscript, we emphasize:
>
> > In Figure 2, we visualize the YouTube network alongside synthetic networks produced by different methods using the Fruchterman-Reingold force-directed algorithm. We also provide additional visualization results obtained using alternative tools in Appendix G.
>
> We refer you to Appendix G of the updated manuscript for additional visualizations. Across different visualizations, the networks generated by SyNGLER exhibit patterns similar to those of the original network from multiple perspectives. While not a strict quantitative comparison, these visualizations provide a straightforward representation of the generated networks.
>
> ---
>
> **Your comment 9:** "As shown in Figure 3, the proposed model is applied to very small networks.
> Another issue is the comparison against deep network models. Considering the similar process to the classical method (sampling an edge with a specific probability), it should be compared against these classical algorithms.
> Also, the training process seems to explode for a large number of nodes; therefore, the paper must increase the size of the network to consider millions of nodes."
>
> **Response:** Thanks for your suggestion on including comparisons with classical network reconstruction methods on larger size networks. We are conducting such empirical study and will update the results as soon as it is finished.
>
> ---
>
> **Your comment 10:** "Table 2 is not discussed in the main paper. The metrics are not discussed in the main paper either."
>
> **Response:** Thanks for this comment. First we would like to clarify that the results in Table 2 are indeed summarized and discussed in Section 4.1 current lines 410-415, and we quote
>
> > Real-world datasets allow a fairer comparison. For each method, we select the configuration that yields the best average performance across all four metrics. The results are summarized in Table 2. EDGE and GRAN ran out of memory on the Yelp dataset on a single NVIDIA GeForce RTX 4090 ......
>
> For the metrics, we previously provided detailed introductions and definitions in Appendix C.3 of the original manuscript. Due to page limits, we now have provided a high-level summary of what they entail in the main text of the updated manuscript, and we quote the relevant part below:
>
> > **Evaluation metrics.** To evaluate the quality of the synthetic networks, we consider the distance between network statistics of the synthetic and the observed networks. Specifically, we evaluate triangle density (Tri., measuring the prevalence of triangle motifs / three-node interactions), clustering coefficient (Clus., summarizing local transitivity), eigenvalue distributions (Eig., reflecting global spectral structure), and degree centrality (DegC., describing node connectivity). For Tri. and Clus., which are single values for each network, we compute the root-mean-square error (RMSE) and bias relative to the observed network. For Eig. and DegC., which are vectors for each network, we compute the maximum mean discrepancy (MMD), the Kolmogorov-Smirnov (KS) statistic, and the energy distance. More details are in Appendix C.3.

---

> ### Author Response · Authors · 2025-11-25
>
> **Summary comment:** "In general, the main contribution has potential, but an important rewriting of the paper must be done. The main paper is not self-contained, lacking important details, which reduces the replicability and analysis of the results."
>
> **Response:** We thank you for recognizing the potential of our paper and for your many constructive comments. We have addressed all your concerns to the best of our ability. Specifically, we have refined the analysis, organization, and presentation of the proofs so that all theoretical results are now self-contained within this paper (see our responses to Comments 1, 3, and 4); we have also provided as many implementation details as possible to enhance reproducibility and facilitate further analysis of the results, with our code published in an anonymous repository at https://github.com/SyNGLER/SyNGLER. In addition, we have conducted further empirical studies with discussions to address your questions regarding classical baselines and larger scale networks. Please do not hesitate to reach out if you have any further questions; we would be happy to discuss them.
>
> ---
>
> **References**
>
> [1] Gao, Fengnan, Zongming Ma, and Hongsong Yuan. "Community detection in sparse latent space models." Journal of Machine Learning Research 23.322 (2022): 1-50.
>
> [2] Zhang, Xuefei, Gongjun Xu, and Ji Zhu. "Joint latent space models for network data with high-dimensional node variables." Biometrika 109.3 (2022): 707-720.
>
> [3] Ma, Zhuang, and Zongming Ma. "Exploration of large networks with covariates via fast and universal latent space model fitting." arXiv preprint arXiv:1705.02372 (2017).
>
> [4] Athreya, Avanti, et al. "Statistical inference on random dot product graphs: a survey." Journal of Machine Learning Research 18.226 (2018): 1-92.
>
> [5] Li, Jinming, et al. "Statistical inference on latent space models for network data." arXiv preprint arXiv:2312.06605 (2023).
>
> [6] Zhu, Yanqiao, et al. "A survey on deep graph generation: Methods and applications." Learning on Graphs Conference. PMLR, 2022.
>
>
> [7] Chen, Xiaohui, et al. "Efficient and degree-guided graph generation via discrete diffusion modeling." arXiv preprint arXiv:2305.04111 (2023).
>
> [8] Liao, Renjie, et al. "Efficient graph generation with graph recurrent attention networks." Advances in neural information processing systems 32 (2019).
>
> [9] You, Jiaxuan, et al. "Graphrnn: Generating realistic graphs with deep auto-regressive models." International conference on machine learning. PMLR, 2018.
>
> [10] ERDdS, P., and A. R&wi. "On random graphs I." Publ. math. debrecen 6.290-297 (1959): 18.
>
> [11] Chung, Fan, and Linyuan Lu. "Connected components in random graphs with given expected degree sequences." Annals of combinatorics 6.2 (2002): 125-145.
>
> [12] Holland, Paul W., Kathryn Blackmond Laskey, and Samuel Leinhardt. "Stochastic blockmodels: First steps." Social networks 5.2 (1983): 109-137.

---

> > ### Comment · Reviewer_qzcR · 2025-11-26
> >
> > I am greatly surprised by the several changes and replies to all of the comments. I will increase my score, but a final decision must be made among all the reviewers.

---

> > > ### Author Response · Authors · 2025-11-26
> > >
> > > Dear reviewer qzcR,
> > >
> > > Thank you for kindly raising our score, and thank you again for the many constructive comments that helped us significantly improve the presentation and further highlight the contributions of our paper! We are still polishing the paper and checking details before the final deadline. Please do not hesitate to contact us if you have any further questions.
> > >
> > > Best,

---

### Official Review · Reviewer_u8rm · 2025-10-31

**Soundness:** 3
**Presentation:** 4
**Contribution:** 3
**Rating:** 6
**Confidence:** 3

**Summary:**

This paper presents SyNGLER, a new framework for efficient and realistic synthetic network generation. The key idea is to separate representation learning and generative modeling: SyNGLER first fits a latent space network model to obtain low-dimensional node embeddings and degree parameters, and then trains a distribution-free generator (either via resampling or a score-based diffusion model) in the latent space. Synthetic networks are generated by sampling new embeddings from the generator and reconstructing edges through the latent model likelihood.

The authors provide theoretical guarantees on the consistency between the true and synthetic edge distributions, decomposing the KL divergence into interpretable terms related to sparsity, latent estimation, and generative modeling errors. Experiments on both simulated and real-world datasets (YouTube, DBLP, Yelp, PolBlogs) demonstrate that SyNGLER achieves superior fidelity to key network statistics (e.g., triangle density, clustering coefficient, eigenvalue and degree distributions) while significantly reducing computational cost (measured in e-FLOPs) compared to deep graph generative models such as VGAE, GRAN, and EDGE.

Overall, the paper proposes a conceptually elegant and computationally efficient method grounded in a statistical foundation.

**Strengths:**

* Good motivation and positioning: The paper clearly identifies computational and structural limitations in existing deep graph generation methods and motivates the latent-space approach convincingly.
* Clear exposition and well-organized presentation: The paper is clearly written, with the model pipeline, algorithm, and theoretical results systematically presented.
* Theoretical rigor: The decomposition of KL divergence into interpretable components and the corresponding asymptotic analysis (Theorems 3.1–3.4) provide a solid theoretical underpinning.

**Weaknesses:**

* Limited empirical comparison with latest graph generative models. The baselines are somewhat dated (VGAE, GRAN, EDGE). Including the latest graph generators (e.g., those mentioned in the Introduction) would strengthen the empirical evaluation.
* Lack of comparison with network reconstruction methods. Since SyNGLER relies on fitting a latent space model, it would be informative to compare it with classical network reconstruction techniques.
* Unclear sensitivity to latent dimension and model choice. The performance of SyNGLER may depend on the latent dimension (r) and the choice of likelihood (Bernoulli vs Gaussian). A sensitivity analysis or ablation would help clarify robustness.

**Questions:**

1. Assumption 2.1(iv) requires the network not to be too sparse. How restrictive is this in practice for real-world sparse graphs?
2. Why was XGBoost chosen for the score network instead of neural parameterization? Does it affect sample quality or training stability?
3. Could the authors include metrics such as graphlet frequency distance or spectral entropy to capture more structural nuances?

---

> ### Author Response · Authors · 2025-11-24
> **Author Response to Reviewer u8rm**
>
> Thank you for taking the time and effort to review our paper. We appreciate your recognition of the novelty and contribution of our paper and also appreciate your constructive comments and questions. Below, we address you comments and questions one by one.
>
> **W1:** "Limited empirical comparison with latest graph generative models. The baselines are somewhat dated (VGAE, GRAN, EDGE). Including the latest graph generators (e.g., those mentioned in the Introduction) would strengthen the empirical evaluation".
>
> **Response:** Thanks for your suggestion to include comparisons with more recent graph generative models. We have conducted additional experiments with a more recent work GraphMaker [1]. Below we present our updated results on the four datasets. Best values are in bold. “–” indicates OOM (out-of-memory) and best performances are in bold.
>
> **(a) YouTube**
>
> |Method|Tri ($\times10^{-4}$)|Clus ($\times10^{-2}$)|Eig ($\times10^{-2}$)|DegC ($\times10^{-2}$)|
> |---|---|---|---|---|
> |E-R|1.05|16.3|25.96|69.22|
> |VGAE|0.559|8.0|17.20|27.83|
> |GRAN|1.13|10.1|7.88|7.86|
> |EDGE|**0.785**|14.4|4.44|7.84|
> |GraphMaker|1.12|16.7|23.13|64.99|
> |SyNG-D|0.967|2.28|2.61|2.13|
> |SyNG-D (MLP)|1.62|1.58|**0.378**|3.89|
> |SyNG-R|1.11|**2.23**|4.47|**1.10**|
>
> **(b) DBLP**
>
> |Method|Tri ($\times10^{-4}$)|Clus ($\times10^{-1}$)|Eig ($\times10^{-1}$)|DegC ($\times10^{-1}$)|
> |---|---|---|---|---|
> |E-R|7.96|8.94|4.409|8.716|
> |VGAE|**0.578**|**0.136**|1.196|2.797|
> |GRAN|7.99|8.92|1.757|2.973|
> |EDGE|2.20|1.33|1.387|0.992|
> |GraphMaker|7.98|8.98|3.931|7.512|
> |SyNG-D|1.58|0.622|0.879|0.926|
> |SyNG-D (MLP)|3.63|1.01|1.116|1.401|
> |SyNG-R|1.43|0.378|**0.799**|**0.748**|
>
>
> **(c) Yelp**
>
> |Method|Tri ($\times10^{-4}$)|Clus ($\times10^{-2}$)|Eig ($\times10^{-2}$)|DegC ($\times10^{-2}$)|
> |---|---|---|---|---|
> |E-R|8.12|14.5|24.53|77.36|
> |VGAE|7.41|10.3|17.58|33.74|
> |GRAN|-|-|-|-|
> |EDGE|-|-|-|-|
> |GraphMaker|8.82|15.5|20.70|69.26|
> |SyNG-D|2.00|1.77|**2.50**|6.72|
> |SyNG-D (MLP)|**0.745**|2.35|9.62|3.26|
> |SyNG-R|0.778|**0.756**|4.69|**0.62**|
>
> **(d) PolBlogs**
>
> |Method|Tri ($\times10^{-4}$)|Clus ($\times10^{-2}$)|Eig ($\times10^{-2}$)|DegC ($\times10^{-2}$)|
> |---|---|---|---|---|
> |E-R|3.22|20.4|40.10|79.24|
> |VGAE|2.07|3.26|31.10|35.33|
> |GRAN|1.57|11.4|9.04|13.86|
> |EDGE|0.790|5.69|4.69|**0.00**|
> |GraphMaker|3.27|20.8|38.28|74.15|
> |SyNG-D|0.710|**1.90**|**2.58**|0.97|
> |SyNG-D (MLP)|**0.594**|2.23|5.27|3.05|
> |SyNG-R|0.830|2.45|3.01|0.92|
>
> We have also compared it on a newly included ML utility task.
>
> **Table.** ML utility evaluation results. “–” indicates OOM. A generated graph is regarded as having high ML utility if the value is close to 1.
>
> |Method|DBLP|PolBlogs|YouTube|Yelp|
> |---|---|---|---|---|
> |SyNG-D|**1.00±0.00**|0.99±0.00|0.99±0.00|0.99±0.00|
> |SyNG-D (MLP)|**1.00±0.00**|0.97±0.01|0.99±0.00|0.99±0.00|
> |SyNG-R|**1.00±0.00**|0.99±0.01|0.99±0.01|**1.00±0.00**|
> |GRAN|0.98±0.08|0.92±0.08|**1.00±0.00**|–|
> |EDGE|**1.00±0.00**|0.98±0.01|**1.00±0.00**|–|
> |GraphMaker|0.95±0.02|**1.00±0.00**|0.99±0.00|0.83±0.01|
>
> For more details, please refer to Section 4 of the updated manuscript.
>
> ---
>
> **W2:** Lack of comparison with network reconstruction methods. Since SyNGLER relies on fitting a latent space model, it would be informative to compare it with classical network reconstruction techniques.
>
> **Response:** Thank you for pointing out network reconstruction methods. We first note that our method encompasses a general latent space network model that covers many classical reconstruction methods, such as the Erdős–Rényi graph, the Chung–Lu graph, and more general stochastic block models. All of these models can be incorporated into our framework. We have emphasized this in the updated manuscript and quote it below:
>
> > **Remark 1.** The latent space network model in Eq. (1) covers a wide class of classical network reconstruction models given appropriate choices of link functions and parametrizations of the node embeddings, including the Erdős–Rényi graph (Erdos & Rényi, 1960), the Chung–Lu graph (Chung & Lu, 2002), and the stochastic block model and its mixed-membership variants (Holland et al., 1983; Karrer & Newman, 2011; Airoldi et al., 2008). Further discussion is provided in Appendix B.
>
> Meanwhile, we are conducting comparisons with other classical network reconstruction techniques including the BTER and mKPGM models, and will update the result as soon as it's finish.

---

> > ### Author Response · Authors · 2025-11-24
> >
> > ---
> > **W3:** "Unclear sensitivity to latent dimension and model choice. The performance of SyNGLER may depend on the latent dimension ($r$) and the choice of likelihood (Bernoulli vs. Gaussian). A sensitivity analysis or ablation would help clarify robustness."
> >
> > **Response:** We appreciate the suggestion regarding sensitivity to the latent dimension and likelihood choice. The latent dimension is indeed an important factor. We conducted a sensitivity analysis on real data to examine how the performance of SyNG-D varies with different latent dimensions (Tables 1–12 below). In general, our method performs stably across different choices of the latent dimension in terms of preserving graph statistics.
> >
> >
> > **Table 1.** Degree centralities on YouTube. W1, KS, Energy scaled by $10^{-2}$; MMD by $10^{-1}$.
> >
> > |Method|cfg|W1|KS|Energy|MMD|
> > |---|---|---|---|---|---|
> > |SyNG-D|2|**0.15±0.07**|**3.76±1.43**|**0.01±0.01**|**0.21±0.18**|
> > | |3|0.23±0.09|5.61±1.67|**0.01±0.01**|0.45±0.20|
> > | |4|0.32±0.10|7.42±1.91|0.03±0.02|0.67±0.22|
> > | |5|0.49±0.09|11.79±1.78|0.07±0.02|1.15±0.19|
> > | |6|0.58±0.09|13.92±1.87|0.10±0.03|1.38±0.20|
> > |SyNG-D (MLP)|2|0.19±0.05|6.05±1.42|**0.01±0.00**|0.42±0.15|
> > | |3|**0.18±0.05**|5.91±1.29|**0.01±0.00**|0.41±0.14|
> > | |4|**0.18±0.05**|**5.88±1.51**|**0.01±0.01**|**0.39±0.17**|
> > | |5|0.22±0.05|8.00±1.57|**0.01±0.01**|0.63±0.15|
> > | |6|0.29±0.08|9.82±1.64|0.03±0.01|0.81±0.18|
> > |GraphMaker|-|**1.35±0.00**|**37.68±0.47**|**0.55±0.01**|**6.50±0.47**|
> >
> > **Table 2.** Eigenvalues on YouTube. KS, Energy, MMD scaled by $10^{-1}$.
> >
> > |Method|cfg|W1|KS|Energy|MMD|
> > |---|---|---|---|---|---|
> > |SyNG-D|2|0.31±0.06|0.30±0.07|0.11±0.05|0.26±0.09|
> > | |3|0.23±0.04|0.20±0.06|0.05±0.03|0.08±0.10|
> > | |4|**0.19±0.04**|**0.14±0.04**|**0.03±0.02**|**0.01±0.03**|
> > | |5|0.30±0.08|0.20±0.06|0.08±0.04|0.02±0.05|
> > | |6|0.40±0.08|0.28±0.07|0.14±0.07|0.11±0.12|
> > |SyNG-D (MLP)|2|0.30±0.06|0.28±0.06|0.10±0.05|0.24±0.09|
> > | |3|0.28±0.06|0.27±0.07|0.09±0.05|0.20±0.11|
> > | |4|0.19±0.05|0.18±0.06|0.04±0.03|0.04±0.07|
> > | |5|**0.17±0.04**|**0.14±0.04**|**0.03±0.02**|0.01±0.04|
> > | |6|0.19±0.06|**0.14±0.04**|**0.03±0.03**|**0.00±0.02**|
> > |GraphMaker|-|**1.62±0.01**|**1.81±0.01**|**3.68±0.04**|**2.31±0.01**|
> >
> > **Table 3.** Clustering coefficient and triangle density on YouTube. Clust metrics scaled by $10^{-2}$; Tri metrics by $10^{-4}$.
> >
> > |Method|cfg|Clust RMSE|Clust MAE|Clust Bias|Tri RMSE|Tri MAE|Tri Bias|
> > |---|---|---|---|---|---|---|---|
> > |SyNG-D|2|2.28|2.15|-2.15|0.97|0.92|0.92|
> > | |3|1.69|1.46|-1.41|0.89|0.84|0.84|
> > | |4|**1.34**|**1.08**|**-0.95**|0.83|0.78|0.78|
> > | |5|1.50|1.22|-1.09|0.58|0.52|0.52|
> > | |6|1.42|1.19|-1.01|**0.41**|**0.35**|**0.35**|
> > |SyNG-D (MLP)|2|**1.12**|**0.88**|**-0.46**|1.35|1.30|1.30|
> > | |3|1.87|1.63|1.54|1.47|1.43|1.43|
> > | |4|1.58|1.34|1.15|1.62|1.58|1.58|
> > | |5|2.98|2.83|2.80|1.40|1.36|1.36|
> > | |6|3.13|2.93|2.88|**1.18**|**1.14**|**1.14**|
> > |GraphMaker|-|**16.71**|**16.71**|**-16.71**|**1.12**|**1.12**|**-1.12**|
> >
> > **Table 4.** Degree centralities on Yelp. W1, Energy, MMD scaled by $10^{-2}$; KS by $10^{-1}$.
> >
> > |Method|cfg|W1|KS|Energy|MMD|
> > |---|---|---|---|---|---|
> > |SyNG-D|2|**0.15±0.05**|**0.25±0.07**|**0.54±0.19**|**0.13±0.09**|
> > | |3|0.21±0.10|0.35±0.13|0.77±0.34|0.24±0.14|
> > | |4|0.24±0.10|0.45±0.11|0.98±0.33|0.35±0.11|
> > | |5|0.36±0.11|0.60±0.12|1.42±0.37|0.50±0.12|
> > | |6|0.44±0.13|0.77±0.14|1.81±0.44|0.67±0.14|
> > |SyNG-D (MLP)|2|0.32±0.12|**0.47±0.14**|1.18±0.43|**0.33±0.14**|
> > | |3|0.39±0.12|0.54±0.13|1.38±0.40|0.43±0.12|
> > | |4|0.69±0.15|0.86±0.13|2.49±0.46|0.78±0.12|
> > | |5|**0.23±0.08**|0.52±0.10|**1.01±0.26**|0.42±0.08|
> > | |6|0.30±0.09|0.48±0.11|1.19±0.33|0.45±0.10|
> > |GraphMaker|-|**2.59±0.00**|**4.14±0.02**|**10.33±0.01**|**69.26±0.15**|
> >
> > **Table 5.** Eigenvalues on Yelp. KS, MMD scaled by $10^{-2}$; Energy by $10^{-1}$.
> >
> > |Method|cfg|W1|KS|Energy|MMD|
> > |---|---|---|---|---|---|
> > |SyNG-D|2|1.29±0.09|0.53±0.04|2.93±0.24|0.61±0.05|
> > | |3|1.09±0.10|0.44±0.05|2.43±0.27|0.51±0.05|
> > | |4|0.92±0.08|0.38±0.04|2.03±0.23|0.43±0.05|
> > | |5|0.77±0.08|0.30±0.04|1.62±0.22|0.34±0.05|
> > | |6|**0.64±0.09**|**0.23±0.05**|**1.28±0.25**|**0.25±0.06**|
> > |SyNG-D (MLP)|2|2.07±0.10|0.84±0.04|4.89±0.25|0.96±0.05|
> > | |3|1.88±0.10|0.75±0.04|4.43±0.25|0.87±0.04|
> > | |4|2.00±0.11|0.82±0.04|4.79±0.26|0.94±0.05|
> > | |5|**1.21±0.10**|**0.51±0.04**|**2.83±0.25**|**0.57±0.05**|
> > | |6|1.55±0.09|0.66±0.04|3.73±0.23|0.75±0.04|
> > |GraphMaker|-|**2.94±0.01**|**1.60±0.00**|**7.63±0.02**|**2.07±0.00**|

---

> > > ### Author Response · Authors · 2025-11-24
> > >
> > > **Table 6.** Clustering coefficient and triangle density on Yelp. Clust metrics scaled by $10^{-2}$; Tri metrics by $10^{-4}$.
> > >
> > > |Method|cfg|Clust RMSE|Clust MAE|Clust Bias|Tri RMSE|Tri MAE|Tri Bias|
> > > |---|---|---|---|---|---|---|---|
> > > |SyNG-D|2|2.56|2.52|-2.52|**1.24**|**1.08**|**-1.05**|
> > > | |3|2.65|2.61|-2.61|1.71|1.57|-1.56|
> > > | |4|2.01|1.96|-1.96|1.59|1.47|-1.46|
> > > | |5|1.83|1.77|-1.77|1.82|1.72|-1.71|
> > > | |6|**1.77**|**1.70**|-1.70|2.00|1.89|-1.88|
> > > |SyNG-D (MLP)|2|2.35|2.31|-2.31|**0.74**|**0.61**|**-0.22**|
> > > | |3|1.05|0.96|-0.96|1.40|1.18|1.15|
> > > | |4|0.75|0.63|0.58|3.33|3.18|3.18|
> > > | |5|0.81|0.71|-0.68|0.98|0.84|-0.72|
> > > | |6|**0.65**|**0.53**|0.47|1.44|1.26|1.23|
> > > |GraphMaker|-|**15.52**|**15.52**|-15.52|**8.82**|**8.82**|**-8.82**|
> > >
> > > **Table 7.** Degree centralities on PolBlogs. All metrics scaled by $10^{-2}$.
> > >
> > > |Method|cfg|W1|KS|Energy|MMD|
> > > |---|---|---|---|---|---|
> > > |SyNG-D|2|**0.18±0.08**|**4.53±0.97**|**0.01±0.01**|**0.97±1.44**|
> > > | |3|0.22±0.11|5.02±1.34|**0.01±0.01**|1.95±2.15|
> > > | |4|0.28±0.13|5.74±1.64|0.02±0.02|3.18±2.50|
> > > | |5|0.44±0.14|8.29±2.04|0.04±0.02|6.49±2.44|
> > > | |6|0.52±0.15|9.98±2.11|0.06±0.03|8.31±2.50|
> > > |SyNG-D (MLP)|2|0.20±0.07|9.59±1.21|**0.01±0.01**|1.47±1.71|
> > > | |3|0.24±0.10|9.81±1.09|**0.01±0.01**|3.05±2.25|
> > > | |4|0.33±0.15|10.06±1.49|0.02±0.02|3.20±2.26|
> > > | |5|0.23±0.07|**8.99±0.93**|**0.01±0.01**|2.79±1.77|
> > > | |6|**0.19±0.08**|9.16±1.35|**0.01±0.01**|**1.41±1.73**|
> > > |GraphMaker|-|**1.72±0.01**|**49.48±0.60**|**0.82±0.01**|**74.15±0.75**|
> > >
> > > **Table 8.** Eigenvalues on PolBlogs. KS, Energy, MMD scaled by $10^{-2}$.
> > >
> > > |Method|cfg|W1|KS|Energy|MMD|
> > > |---|---|---|---|---|---|
> > > |SyNG-D|2|0.21±0.06|0.48±0.09|0.92±0.51|0.26±0.14|
> > > | |3|0.20±0.04|0.50±0.10|0.90±0.38|0.32±0.12|
> > > | |4|**0.20±0.05**|0.44±0.10|**0.74±0.32**|0.25±0.12|
> > > | |5|0.25±0.07|**0.32±0.08**|0.80±0.46|0.15±0.12|
> > > | |6|0.30±0.09|0.33±0.09|1.19±0.72|**0.14±0.14**|
> > > |SyNG-D (MLP)|2|0.46±0.10|0.70±0.09|3.36±1.34|0.55±0.11|
> > > | |3|**0.37±0.09**|0.70±0.08|**2.51±1.04**|**0.53±0.09**|
> > > | |4|0.65±0.11|0.89±0.09|6.88±1.98|0.82±0.11|
> > > | |5|0.38±0.10|0.70±0.09|2.81±1.18|0.56±0.11|
> > > | |6|0.39±0.10|**0.67±0.09**|2.80±1.18|0.54±0.11|
> > > |GraphMaker|-|**1.78±0.01**|**3.22±0.02**|**7.25±0.09**|**3.83±0.01**|
> > >
> > > **Table 9.** Clustering coefficient and triangle density on PolBlogs. Clust metrics scaled by $10^{-2}$; Tri metrics by $10^{-4}$.
> > >
> > > |Method|cfg|Clust RMSE|Clust MAE|Clust Bias|Tri RMSE|Tri MAE|Tri Bias|
> > > |---|---|---|---|---|---|---|---|
> > > |SyNG-D|2|1.90|1.51|1.15|0.71|**0.55**|**0.13**|
> > > | |3|**1.56**|**1.23**|0.54|**0.68**|0.56|-0.33|
> > > | |4|1.85|1.45|0.90|0.78|0.66|-0.43|
> > > | |5|1.66|1.33|**0.49**|1.09|1.00|-0.98|
> > > | |6|1.91|1.49|0.93|1.19|1.09|-1.07|
> > > |SyNG-D (MLP)|2|3.58|3.26|3.23|0.91|0.69|0.56|
> > > | |3|**2.23**|**1.88**|**1.73**|**0.59**|**0.48**|-0.16|
> > > | |4|5.69|5.45|5.45|1.80|1.61|1.60|
> > > | |5|5.18|4.77|4.74|1.15|0.90|0.77|
> > > | |6|3.15|2.80|2.72|0.84|0.65|0.50|
> > > |GraphMaker|-|**20.75**|**20.75**|-20.75|**3.27**|**3.27**|-3.27|
> > >
> > > **Table 10.** Degree centralities on DBLP. W1, Energy scaled by $10^{-2}$; KS, MMD by $10^{-1}$.
> > >
> > > |Method|cfg|W1|KS|Energy|MMD|
> > > |---|---|---|---|---|---|
> > > |SyNG-D|2|0.19±0.08|**0.75±0.17**|**0.02±0.01**|**0.93±0.23**|
> > > | |3|**0.18±0.06**|0.91±0.16|**0.02±0.01**|1.03±0.19|
> > > | |4|0.22±0.07|1.22±0.14|0.03±0.01|1.29±0.17|
> > > | |5|0.33±0.09|1.74±0.18|0.05±0.02|1.78±0.20|
> > > | |6|0.44±0.10|2.19±0.18|0.09±0.02|2.22±0.20|
> > > |SyNG-D (MLP)|2|**0.31±0.07**|1.81±0.17|**0.04±0.01**|**1.50±0.16**|
> > > | |3|0.42±0.06|1.33±0.13|0.07±0.02|1.67±0.18|
> > > | |4|0.47±0.15|**1.19±0.19**|0.06±0.03|**1.40±0.25**|
> > > | |5|0.47±0.10|1.36±0.14|0.07±0.02|1.72±0.16|
> > > | |6|0.35±0.06|1.53±0.17|0.06±0.01|1.73±0.19|
> > > |GraphMaker|-|**1.37±0.01**|**5.56±0.07**|**0.68±0.01**|**7.51±0.07**|
> > >
> > > **Table 11.** Eigenvalues on DBLP. W1, KS, MMD scaled by $10^{-1}$; Energy by $10^{-2}$.
> > >
> > > |Method|cfg|W1|KS|Energy|MMD|
> > > |---|---|---|---|---|---|
> > > |SyNG-D|2|3.02±0.32|**0.81±0.06**|2.35±0.43|0.88±0.06|
> > > | |3|2.13±0.28|0.91±0.07|**1.57±0.21**|**0.77±0.05**|
> > > | |4|**1.73±0.17**|1.06±0.07|1.65±0.20|**0.77±0.05**|
> > > | |5|2.56±0.43|1.32±0.08|2.99±0.57|0.96±0.08|
> > > | |6|3.80±0.46|1.63±0.09|5.62±0.89|1.28±0.08|
> > > |SyNG-D (MLP)|2|**2.85±0.34**|1.10±0.07|**2.37±0.31**|**0.93±0.05**|
> > > | |3|6.09±0.47|1.10±0.08|8.94±1.27|1.35±0.07|
> > > | |4|4.46±0.41|**0.83±0.05**|5.05±0.87|1.12±0.07|
> > > | |5|5.75±0.46|0.97±0.07|7.42±1.06|1.32±0.06|
> > > | |6|5.92±0.45|1.03±0.08|8.06±1.13|1.31±0.06|
> > > |GraphMaker|-|**18.39±0.14**|**3.32±0.02**|**85.40±1.14**|**3.93±0.02**|

---

> > > > ### Author Response · Authors · 2025-11-24
> > > >
> > > > **Table 12.** Clustering coefficient and triangle density on DBLP. Clust metrics scaled by $10^{-1}$; Tri metrics by $10^{-4}$.
> > > >
> > > > |Method|cfg|Clust RMSE|Clust MAE|Clust Bias|Tri RMSE|Tri MAE|Tri Bias|
> > > > |---|---|---|---|---|---|---|---|
> > > > |SyNG-D|2|0.62|0.59|-0.59|**1.58**|**1.26**|**0.46**|
> > > > | |3|**0.53**|**0.49**|-0.49|1.61|1.37|-0.87|
> > > > | |4|0.55|0.52|-0.52|1.81|1.57|-1.39|
> > > > | |5|**0.53**|**0.49**|-0.49|2.34|2.13|-2.07|
> > > > | |6|0.57|0.54|-0.54|3.01|2.86|-2.86|
> > > > |SyNG-D (MLP)|2|1.40|1.36|-1.36|2.91|2.67|-2.62|
> > > > | |3|2.42|2.39|-2.39|3.74|3.64|-3.64|
> > > > | |4|**1.01**|**0.99**|-0.99|3.63|3.20|3.13|
> > > > | |5|1.44|1.41|-1.41|2.29|1.81|1.47|
> > > > | |6|1.71|1.69|-1.69|**1.80**|**1.54**|**-1.37**|
> > > > |GraphMaker|-|**8.98**|**8.98**|-8.98|**7.98**|**7.98**|-7.98|
> > > >
> > > > Regarding the choice of likelihood, since most of our experiments are performed on graph data, the binary logistic link is a natural way to model the observations. Here we also consider a linear link function (Gaussian) for comparison. Given the estimated link probability matrix $\hat P$, we evaluate
> > > > $$
> > > > \mathrm{ERR}(\hat P)=\frac{2}{n(n-1)}\sum_{i<j}(\hat P_{ij}-A_{ij})^2,\qquad
> > > > \mathrm{corr}(\hat P,A)=\frac{\sum_{i<j}(\hat P_{ij}-\bar{\hat P})(A_{ij}-\bar{A})}{\sqrt{\sum_{i<j}(\hat P_{ij}-\bar{\hat P})^2}\sqrt{\sum_{i<j}(A_{ij}-\bar{A})^2}}.
> > > > $$
> > > >
> > > > Our results below show that the logistic link function yields a better estimation of the link probability matrix than the linear link function for binary graph data. In particular, to achieve comparable (though still worse) performance with a linear link, one needs a much larger latent dimension (e.g., $r=100$) compared with the logistic link (e.g., $r=5$), while the linear link with the same $r=5$ performs much worse.
> > > >
> > > > **Table 13.** Link prediction performance of LSMs with logistic vs. linear link.
> > > >
> > > > |Model|Estimation Error|Correlation|AUC|
> > > > |---|---:|---:|---:|
> > > > |Logistic link ($r=5$)|**0.001579**|**0.7389**|**0.9982**|
> > > > |Linear link ($r=100$)|0.002069|0.6176|0.9823|
> > > > |Linear link ($r=100$)|0.003048|0.2664|0.8559|
> > > >
> > > > For general network datasets, the link distribution can be chosen flexibly based on the support and spread of the edge observations to better model the response.
> > > >
> > > > **Q1:** Assumption 2.1(iv) requires the network not to be too sparse. How restrictive is this in practice for real-world sparse graphs?
> > > >
> > > > **Response:** Thank you for your insightful question regarding network sparsity. We have conducted refined analysis and improved the result; now the sparsity regime aligns with existing literature. We have updated the result and corresponding discussion in the manuscript, and we quote below:
> > > >
> > > > >**Assumption 3.1 (Network sparsity)** The global edge sparsity follows that $w_n=\exp(\rho_n^*)=\Omega(\log{n}/n)$.
> > > > >
> > > > > Assumption 3.1 states that the edge density is bounded below by  $\Omega(\log n / n)$, and accordingly the expected node degrees are at least of order $\log n$ as $n$ grows. Such a sparsity regime is consistent with the network analysis literature;
> > > > > see, for example, Athreya et al. (2018) and Ma et al. (2020)
> > > >
> > > > ---
> > > >
> > > > **Q2:** Why was XGBoost chosen for the score network instead of neural parameterization? Does it affect sample quality or training stability?
> > > >
> > > > **Response:**  Thank you for this question regarding our choice of XGBoost for the score network. The XGBoost-based score fitting method was proposed in ForestDiffusion [1] to enable efficient score estimation for tabular data, which fits the setting of latent node embeddings. Empirically, we found that XGBoost-based score estimation is much faster to train than MLP-based score estimation and allows efficient CPU-based training. Regarding generation performance, we compared MLP-based score estimation with XGBoost-based score estimation on real data and found that the XGBoost-based approach yields better performance, as shown in Tables 1–12 above.
> > > >
> > > > **Q3:** Could the authors include metrics such as graphlet frequency distance or spectral entropy to capture more structural nuances?
> > > >
> > > > **Response:** We thank the reviewer for the insightful suggestion. Graphlet-based metrics and spectral-entropy measures indeed capture valuable local structural information. In response to this comment, we have additionally computed **4-node Graphlet Frequency Distance (GFD)** for all baselines to provide a complementary evaluation of structural nuances between original network and generated networks.
> > > >
> > > > To compute the 4-node GFD, we followed a standard ORCA-based pipeline[2]. For each graph, we enumerated all **node orbits** associated with 3-node and 4-node graphlets to obtain the graphlet orbit count vector, which we normalize into a graphlet frequency distribution. For two graphs $G_1$ and $G_2$, we define the distance between their graphlet frequency vectors using either the $L_1$ or $L_2$ norm:
> > > > $$
> > > > \text{GFD}(G_1, G_2) = \|F(G_1) - F(G_2)\|.
> > > > $$

---

> > > > > ### Author Response · Authors · 2025-11-24
> > > > >
> > > > > This metric captures fine-grained local connectivity patterns beyond global statistics. Table 14 reports 4-node GFD for all datasets and baselines; SyNGLER consistently attains smaller GFD than competing methods, supporting the structural robustness of our generated networks.
> > > > >
> > > > > **Table 14.** 4-node Graphlet Frequency Distance (GFD) across datasets and baselines (all entries ×$10^{-1}$).
> > > > >
> > > > > |Method|cfg|L1-YT|L1-DBLP|L1-PB|L1-Yelp|L2-YT|L2-DBLP|L2-PB|L2-Yelp|
> > > > > |---|---|---|---|---|---|---|---|---|---|
> > > > > |SyNG-D|2|1.42±0.91|3.80±0.96|0.76±0.39|0.38±0.09|0.70±0.48|1.97±0.51|0.33±0.19|0.16±0.05|
> > > > > | |3|**1.39±0.97**|**3.03±0.98**|**0.72±0.42**|0.42±0.11|**0.68±0.51**|**1.58±0.52**|**0.32±0.21**|0.17±0.05|
> > > > > | |4|1.45±0.98|3.35±0.86|0.86±0.46|**0.30±0.11**|0.71±0.50|1.75±0.45|0.38±0.23|**0.13±0.05**|
> > > > > | |5|1.45±0.97|3.17±0.90|0.89±0.52|0.32±0.13|0.71±0.50|1.67±0.47|0.40±0.26|0.14±0.06|
> > > > > | |6|1.69±1.05|3.46±0.94|0.97±0.51|0.31±0.13|0.83±0.54|1.83±0.49|0.43±0.25|0.14±0.06|
> > > > > |SyNG-D (MLP)|2|2.08±0.98|6.92±1.11|**1.19±0.52**|0.35±0.11|1.04±0.50|3.60±0.59|**0.51±0.26**|0.14±0.05|
> > > > > | |3|2.55±1.03|9.51±0.86|1.23±0.52|0.37±0.19|1.24±0.50|5.04±0.48|0.55±0.25|0.17±0.10|
> > > > > | |4|**1.71±0.86**|**6.17±0.74**|1.47±0.38|0.34±0.13|**0.82±0.43**|**3.19±0.39**|0.61±0.18|0.13±0.05|
> > > > > | |5|3.02±0.96|7.22±0.86|1.41±0.55|**0.23±0.14**|1.45±0.46|3.73±0.45|0.56±0.26|**0.11±0.07**|
> > > > > | |6|2.68±1.14|7.86±0.73|1.37±0.74|0.27±0.11|1.28±0.55|4.08±0.39|0.60±0.38|0.11±0.05|
> > > > > |SyNG-R|2|1.38±1.01|1.91±0.91|**0.81±0.39**|0.39±0.11|0.67±0.53|0.99±0.51|**0.35±0.19**|0.15±0.04|
> > > > > | |3|**1.33±1.05**|1.60±0.89|0.83±0.38|0.33±0.11|0.65±0.53|0.83±0.49|0.36±0.18|0.13±0.04|
> > > > > | |4|1.34±1.07|1.31±0.80|0.86±0.38|0.23±0.12|**0.65±0.53**|0.67±0.44|0.36±0.18|0.09±0.05|
> > > > > | |5|1.36±1.07|**1.13±0.72**|0.87±0.39|0.18±0.11|0.65±0.52|**0.58±0.39**|0.37±0.18|0.08±0.05|
> > > > > | |6|1.37±1.08|1.14±0.81|0.88±0.37|**0.18±0.11**|0.66±0.53|0.58±0.44|0.37±0.18|**0.08±0.05**|
> > > > > |VGAE|16|6.30±0.03|**1.51±0.11**|3.37±0.08|3.93±0.01|2.92±0.01|0.80±0.07|1.62±0.04|1.64±0.00|
> > > > > | |2|6.53±0.04|13.69±0.01|**2.95±0.09**|**3.66±0.01**|2.97±0.01|7.39±0.01|**1.45±0.04**|**1.53±0.00**|
> > > > > | |3|6.50±0.04|2.54±0.13|3.21±0.08|3.85±0.01|2.96±0.01|1.14±0.09|1.54±0.04|1.61±0.00|
> > > > > | |4|**5.91±0.03**|1.57±0.12|3.08±0.08|3.86±0.01|**2.79±0.01**|**0.68±0.06**|1.50±0.04|1.61±0.00|
> > > > > | |5|6.61±0.03|1.57±0.14|3.00±0.08|3.80±0.01|3.00±0.01|0.69±0.07|1.48±0.04|1.58±0.00|
> > > > > | |6|6.20±0.03|1.69±0.15|3.19±0.09|4.09±0.01|2.88±0.01|0.85±0.07|1.51±0.04|1.70±0.00|
> > > > > |GRAN|128|1.90±0.30|18.35±0.14|**1.95±0.33**|–|0.77±0.14|9.92±0.04|**0.79±0.19**|–|
> > > > > | |256|2.42±0.18|17.26±0.15|3.88±0.76|–|1.07±0.10|8.49±0.53|1.73±0.37|–|
> > > > > | |512|**1.67±0.38**|**17.17±0.08**|3.04±0.30|–|**0.67±0.17**|**8.22±0.31**|1.40±0.16|–|
> > > > > |EDGE|-|**5.35±0.49**|**5.27±2.28**|**1.14±0.29**|–|**2.12±0.19**|**2.57±1.35**|**0.45±0.11**|–|
> > > > > |GraphMaker|-|**7.97±0.00**|**17.45±0.01**|**7.88±0.01**|**5.17±0.00**|**3.48±0.00**|**8.32±0.00**|**3.23±0.00**|**2.14±0.00**|
> > > > > |ER|-|**7.90±0.00**|**17.39±0.01**|**7.80±0.01**|**5.02±0.00**|**3.47±0.00**|**8.32±0.00**|**3.22±0.00**|**2.08±0.00**|
> > > > >
> > > > > We have incorporated most of the additional discussions and results into the manuscript and will continue updating it until the final version is completed.
> > > > >
> > > > > [1] Jolicoeur-Martineau et al. (2024). *Generating and imputing tabular data via diffusion and flow-based gradient-boosted trees*. AISTATS.
> > > > >
> > > > > [2] Hočevar & Demšar (2014). *A combinatorial approach to graphlet counting*. Bioinformatics.

---

### Official Review · Reviewer_DqnC · 2025-11-01

**Soundness:** 3
**Presentation:** 2
**Contribution:** 2
**Rating:** 2
**Confidence:** 3

**Summary:**

This paper focuses on the topic of synthetic network generation, aiming to address a key challenge: how to generate graphs that preserve the structural properties of a real-world network while remaining computationally efficient. To tackle this, the authors propose SyNGLER, a two-stage framework. First, a latent space network model is used to learn low-dimensional node embeddings, then, a distribution-free generator is trained in this latent space to produce new node embeddings, from which synthetic graphs are generated. The paper provides theoretical guarantees via KL divergence decomposition and consistency analysis, and empirically shows on synthetic and real datasets that SyNGLER better preserves structural statistics while being more computationally efficient than existing methods.

**Strengths:**

1. The paper addresses an important problem — generating synthetic graphs that preserve structural properties while being computationally efficient.
2. Provides theoretical analysis.
3. The experiments cover both synthetic and multiple real-world networks (YouTube, DBLP, Yelp, PolBlogs), and evaluate diverse structural metrics (degree, clustering, eigenvalues, triangle density)

**Weaknesses:**

1. While synthetic graph generation is indeed a meaningful and important research direction, this paper focuses only on generating non-attributed graphs that replicate structural statistics (e.g., degree, clustering, spectrum) of a real network. First of all, in modern applications, most graphs are attributed, and the interaction between attributes and structure is often essential to the task, how could the proposed SyNGLER benefit such graphs remain undiscussed. Moreover, the paper does not demonstrate or validate any downstream usage, so it remains unclear what practical purpose these synthetic structure-only graphs serve beyond matching statistical properties.

2. Would be better if code could be provided.

**Questions:**

I was curious to see what would be the practical impact of the SyNGLER generated graphs, and how to validate these practical impacts.

1. For the practical impact, the method only focuses on generating network structure (adjacency), without modeling node or edge attributes. In most of the real-world applications, graphs are attributed, and structure alone is insufficient. Ignoring attributes may lead to significant information loss, and it is unclear how the generated synthetic graphs could be used for tasks that rely on feature–structure interactions (e.g., GNN training, attributed diffusion). I wonder if the authors has any empirical or insights on how SyNGLER could be extended to attributed graphs?

2. For validating the practical impact, the paper demonstrates that the generated networks match several structural statistics of the input graph. However, structural similarity alone does not guarantee that synthetic networks can serve as valid surrogates for scientific analysis or downstream decision-making. How can we validate functional equivalence of the input graph and the synthetic graph?

---

> ### Author Response · Authors · 2025-11-23
> **Author Response to Reviewer DqnC**
>
> Thank you for taking the time and effort to review our paper. We appreciate your recognition of the importance of the problem studied and of our theoretical and empirical analysis, as well as your critiques on the extension to attributed graphs and on the practical impact of the SyNGLER generated graphs, which have motivated us to conduct further study and have significantly improved the paper. Below, we address your questions one by one.
>
> **Q1/W1. Regarding extension to attributed graphs**
>
> **Response:** Thanks for your insightful question regarding attributed graphs.
> This is indeed an important direction that broadens the applicability of our framework to a wider range of real-world problems, as many graph modeling tasks, such as training graph neural networks, take into consideration both the graph structure and the node attributes. In this revision, we have included a subsection titled “SyNGLER-Attr: Synthetic Network Generation via Latent Embedding Reconstruction for Attributed Networks,” which we quote below:
>
> > In the main body of this work, we study the task of generating networks when only network structures are observed and needed in downstream tasks, using the vanilla SyNGLER. Meanwhile, generating attributed networks that preserve feature-network interactions is also an important problem. The classical VGAE (Kipf & Welling, 2016) approaches this problem in the setting of fixed attributes, where the attributes of the generated graph remain the same as those of the input graph. Using continuous diffusion models, Jo et al. (2022) developed a pipeline that successfully generates attributed graphs with new features, although its design and training are tailored to multiple small-scale graphs. Li et al. (2023) uses a message-passing neural network (MPNN) as the encoder and is able to efficiently generate attributed graphs from a single large observation. In our paper, to address this task, we introduce SyNGLER-Attr, a generalization of SyNGLER for generating attributed networks.
> > Specifically, SyNGLER-Attr jointly embeds and reconstructs the latent embeddings of the network and its associated attributes, allowing the synthetic data to preserve the joint latent structures of the network and its attributes. Due to page limits, we present the detailed algorithmic description and implementation, along with an empirical study of SyNGLER-Attr to assess its utility in downstream machine learning tasks in Appendix E.
>
> Below, we introduce SyNGLER-Attr and present empirical results.
>
> We first clarify the setting of generating attributed graphs. In this task, we are given a single large attributed graph with both an adjacency matrix $A \in \mathbb{R}^{n \times n}$ and an attribute matrix $Y \in \mathbb{R}^{n \times p}$, where $n$ is the number of nodes and $p$ is the number of attributes. Our goal is to generate a new graph $\tilde{A}$ and a new attribute matrix $\tilde{Y}$ that preserve the structural and attribute properties of the original graph. In other words, we want the joint distribution of $(\tilde{A}, \tilde{Y})$ to be close to the joint distribution of $(A, Y)$. This setting has been considered in the literature on attributed graph generation, such as [2, 3].
>
>
> SyNGLER-Attr achieves this by jointly embedding the network and attributes into a latent space, motivated by [1]. Consider an attributed network with binary edges.
> Suppose that $A \in \{0,1\}^{n \times n}$ is the adjacency matrix of the network and $Y \in \mathbb{R}^{n \times p}$ is the attribute matrix.
> We embed the network into an embedding matrix $Z = (Z_1, Z_2) \in \mathbb{R}^{n \times (d_1 + d_2)}$, where $Z_1$ captures network latent factors and $Z_2$ captures attribute-specific latent factors.
> Concretely, we model the network and the attributes as follows:
> $$
> A_{ij} \stackrel{\text{ind.}}{\sim} \text{Bernoulli}(P_{ij}) \quad \text{with} \quad P = g(\alpha 1_n^\top + 1_n \alpha^\top + Z_1 Z_1^\top), \tag{1}
> $$
> $$
> Y = 1_n \mu^\top + Z_1 \Lambda_1^\top + Z_2 \Lambda_2^\top + \mathcal{E},
> \qquad \epsilon_i \stackrel{\text{i.i.d.}}{\sim} \mathcal{N}(0, \Psi), \tag{2}
> $$
> where $\Lambda_1$ and $\Lambda_2$ are the loadings for $Z_1$ and $Z_2$, respectively, and $\Psi$ is a diagonal noise covariance matrix.
>
> Given this model, we first estimate $Z_1$ from the adjacency matrix $A$ using the latent space model described in the paper.
> We then estimate $\Lambda_1$ by regressing the centered attributes on $Z_1$, and compute the residuals to estimate $Z_2$ and $\Lambda_2$ via factor analysis.
> As a result, we obtain the embedding matrix $Z=(Z_1,Z_2)$ and the loadings $\Lambda$ for both network and attribute factors, which can be used for generation.
> For generation, we learn the distribution of the rows of $Z$ and sample new embeddings from this learned distribution.
> Substituting the generated embeddings into the above models with the estimated loadings, we can then generate a new adjacency matrix and a new attribute matrix.

---

> ### Author Response · Authors · 2025-11-23
>
> Our algorithm is summarized in the following steps:
>
> **Algorithm 1: SyNGLER-Attr: Synthetic Network Generation via Latent Embedding Reconstruction for Attributed Network**
>
> **Input:**  Adjacency matrix $A \in$ {0,1}$^{n\times n}$, Attribute matrix $Y \in \mathbb{R}^{n\times p}$.
> 1. Maximum likelihood estimation using the latent space likelihood model $(A_{ij} \mid z_{1i},z_{1j},\alpha_i,\alpha_j,\rho)
> \sim \text{Bernoulli}\big(
> g(z_{1i}^\top z_{1j}+\alpha_i+\alpha_j+\rho)
> \big)$ to obtain $(\hat Z_1,\hat\alpha,\hat\rho)\in\mathbb{R}^{n\times d_1}\times\mathbb{R}^n\times\mathbb{R}$.
> 2. Regress $Y$ on $Z_1$ under the model $Y =  \mathbf{1}_n\mu^T+Z_1 \Lambda_1^\top+ R$, where R denotes the residual matrix.
>
> 3. Conduct an eigenvalue ratio test on the residual matrix $R$ to determine the dimension of the additional latent variable $Z_2$. Fit $R = Z_2\Lambda_2^\top + E, E_{i\cdot}\sim N(0,\Phi)$ where $\Phi$ is diagonal to obtain$(\hat Z_2,\hat\Lambda_2,\hat\Phi).$
>
> 4. Train a sampler using $\hat{Z} = (\hat{Z_1}, \hat{Z_2})$ and $\hat{\alpha}$, i.e.,  $\text{Sampler} = \mathrm{GenModel}\big(\{(\hat z_i,\hat\alpha_i)\}_{i=1}^n\big).$
>
>
> 5. Sample new latent embeddings:$(\tilde z_{1i}^{\top}, \tilde{z}_{2i}^{\top},\tilde\alpha_i)^{\top} \sim \text{Sampler}.$
>
> 6. With sampled latent variables $\tilde z_i =\begin{pmatrix}
> \tilde z_{1i}^{\top} \\
> \tilde z_{2i}^{\top}
> \end{pmatrix}^{\top} \in \mathbb{R}^{d_1+d_2},\tilde{\alpha}$, generate $A$ via $\tilde A_{ij} = \tilde A_{ji}\sim p(\cdot\,\big|\,\tilde z_{1i}^\top\tilde z_{1j} + \tilde\alpha_i + \tilde\alpha_j + \hat\rho)$, and generate attributes via $\tilde Y = \mathbf{1}_n\hat\mu^\top + \tilde Z_1\hat\Lambda_1^\top + \tilde Z_2\hat\Lambda_2^\top + N(0,\hat{\Phi}).$
>
> **Output:** Generated network $\tilde A$ and generated attributes $\tilde Y$.
>
> In this revision, we were also aware of an important reference [2], which introduces GraphMaker, a method that can generate both attributed and non-attributed networks. We have now included this reference in the Introduction, where we write:
>
> > A recent work Li et al. (2023) uses a message-passing neural network (MPNN) as the encoder and is
> able to efficiently generate networks from a single large observation.
>
> We have also included GraphMaker as a baseline method in our numerical study for both attributed and non-attributed network generation tasks in the updated manuscript.
>
> Below, we present experiments on the Cora dataset, an attributed network dataset, in Table 1 and Table 2. In both tables, SyNGLER-Attr captures the network structure and the attribute distribution, verifying the effectiveness of our method for generating attributed graphs.
>
> In particular, SyNGLER-Attr is able to generate graphs with structural statistics that are closer to those of the original graph and with comparable numerical statistics, as compared to GraphMaker [2], as shown in Table 1. We also compare the generated attributes with the original attributes in terms of KS distance and MMD, and find that SyNGLER-Attr achieves better performance than GraphMaker, as shown in Table 2.
>
> In addition, we include an ML-utility evaluation with a link prediction task, where we train two GAE models on the generated graph and the original graph and compare their performance on the test split of the original graph. The details of this ML-utility evaluation are described in the response to Q2 below.
>
> **Table 1:** Generation performance of SyNG-Attr and GraphMaker on the Cora dataset.
>
> |Category|Metric|SyNG-Attr|GraphMaker|
> |---|---|---:|---:|
> |**Structural Metrics**|Triangle Density (RMSE)|$3.60e^{-5}$|$\mathbf{1.00e^{-6}}$|
> | |Clustering Coefficient (RMSE)|$\mathbf{5.38e^{-2}}$|$7.72e^{-2}$|
> | |Eigenvalues (MMD)|$\mathbf{6.80e^{-1}}$|$1.03$|
> | |Degree Centrality (MMD)|$2.61e^{-1}$|$\mathbf{2.36e^{-2}}$|
> |**ML Utility (Link Prediction Task)**|Mean Ratio|$0.98\pm0.01$|$\mathbf{1.00\pm0.00}$|
>
> **Table 2:** KS and MMD distance for the generated attributes of SyNG-Attr and GraphMaker on the Cora dataset.
>
> |Method|KS|MMD|
> |---|---|---|
> |SyNG-Attr|**0.1398**|**0.1676**|
> |GraphMaker|0.1400|0.1678|
>
> ---
>
> **W.2. Regarding providing the codes**
>
> **A** Thank you for the suggestion. We have published our codebase in the anonymous repository at https://github.com/SyNGLER/SyNGLER.
>
> ---
>
> **Q.2 Regarding validating functional equivalence of the input graph and the synthetic graph**
> **Response:** Thank you for your constructive comment on validating the functional equivalence of the input graph and the synthetic graph.
> We strongly agree that validating the practical utility of the generated graphs, as you suggested, is an important aspect of evaluation, in addition to comparing graph statistics that are commonly used in the literature. Based on your suggestion, we have conducted additional experiments following the setting in [2] to evaluate the utility of the generated graph with machine learning models.

---

> ### Author Response · Authors · 2025-11-23
>
> This experiment quantitatively assesses whether the ML model trained on the generated graph can achieve similar performance as the model trained on the original graph. Below, we introduce the detailed pipeline and then provide the corresponding experimental results.
>
> Suppose that $A$ is a network adjacency matrix. We adopt a one-layer GCN-based GAE model with parameter $W_1\in R^{d_0\times d_1}, W_2 \in R^{d_1 \times d_2}$. Suppose that  $X = \phi(A) \in R^{n\times d_0}$ is a feature map (and we use the node degrees as the feature in our experiments).
> We define the following encoder model.
> Suppose that $\tilde A = D^{-1/2} A D^{-1/2}$ is the normalized adjacency matrix where $D = \text{diag}(A\mathbf{1})$ is the degree matrix, then we can define the GCN encoder as
> $$
>  \text{GCN}(W,A) = \tilde A \mathsf{ReLU}(\tilde A \phi (A) W_1) W_2,
> $$
> whose outputs an embedding matrix $Z\in \mathbb{R}^{n\times d_2}$.
> We then predict the link probability with $\sigma(ZZ^\top)$.
>
> **Training:** Let $M$ be a symmetric mask matrix in $\mathbb{R}^{n\times n}$ where the non-zero entries indicate the training edge split.
> We only consider outputs over $A\circ M$ (entry-wise multiplication), i.e.
> $$
> Z^M = \text{GCN}(W,A\circ M), \\
> L(W_1, W_2) = \sum_{(i,j)\in M} -A_{ij}{Z_i^M}^\top  Z_j^M + \log(1+ \exp \{ {Z_i^M}^\top Z_j^M\}).
> $$
> We then optimize this loss function to obtain a trained model with parameter $\hat W$.
>
> **Comparison:** We use the generated graph $A'$ to execute the training procedure above and obtain $\check W$ (we use check to emphasize that it is trained on the generated graph).
> Then we evaluate two models, namely $\text{GCN}(\hat W,A\circ M)$ and $\text{GCN}(\check W,A\circ M)$ on the test split of the original graphs.
>  The evaluation here should take the original masked graph as input, i.e., we predict the edges in the test split with $\sigma(\check{Z}\check{Z}^\top)$, where
> $$
> \check{Z} = \text{GCN}(\check W, A \circ M),
> $$
> Then we evaluate the prediction quality of the AUC in the test split:
> $$
> \text{AUC}(\check W) = \frac{\sum_{(i,j) \notin M} \mathbf{1}[\sigma(\check{Z}_i^\top \check{Z}_j) > \sigma(\hat{Z}_i^\top \hat{Z}_j)]}{n(n-1)/2 - |M|}.
> $$
> Similarly, we can compute $\text{AUC}(\hat W)$ with the same formula.
> Then we can compute the ratio $\text{AUC}(\check W)/\text{AUC}(\hat W)$ to compare the two models.
> As suggested by [2], if this ratio appears to be close to one, the generated graph can potentially be used for training downstream ML models.
> Our experimental results show that our method indeed produces generated graphs that are practically useful.
> The detailed results are reported in the tables below.
>
> **Table 3.** ML utility of SyNG-D, SyNG-R, EDGE, GRAN, and GraphMaker on four datasets. “–”: OOM.
>
> |method|cfg|DBLP|PolBlogs|YouTube|Yelp|
> |---|---|---|---|---|---|
> |SyNG-D|2|**1.00±0.00**|0.98±0.01|0.94±0.02|0.98±0.00|
> |SyNG-D|3|**1.00±0.00**|0.98±0.01|0.98±0.01|**0.99±0.00**|
> |SyNG-D|4|**1.00±0.00**|**0.99±0.01**|0.98±0.01|**0.99±0.00**|
> |SyNG-D|5|**1.00±0.00**|**0.99±0.01**|**0.99±0.01**|**0.99±0.00**|
> |SyNG-D|6|**1.00±0.00**|**0.99±0.00**|**0.99±0.01**|**0.99±0.00**|
> |SyNG-D (MLP)|2|**1.00±0.00**|0.98±0.01|0.90±0.02|0.98±0.00|
> |SyNG-D (MLP)|3|**1.00±0.00**|**0.99±0.01**|0.98±0.01|**0.99±0.00**|
> |SyNG-D (MLP)|4|**1.00±0.00**|0.96±0.01|0.96±0.01|**0.99±0.00**|
> |SyNG-D (MLP)|5|**1.00±0.00**|0.97±0.01|**0.99±0.00**|**0.99±0.00**|
> |SyNG-D (MLP)|6|**1.00±0.00**|0.98±0.01|0.98±0.01|**0.99±0.00**|
> |SyNG-R|2|**1.00±0.00**|0.98±0.01|0.96±0.02|0.98±0.00|
> |SyNG-R|3|**1.00±0.00**|0.98±0.01|0.98±0.01|0.99±0.00|
> |SyNG-R|4|**1.00±0.00**|0.98±0.01|0.98±0.01|0.99±0.00|
> |SyNG-R|5|**1.00±0.00**|0.98±0.01|**0.99±0.01**|**1.00±0.00**|
> |SyNG-R|6|**1.00±0.00**|**0.99±0.01**|**0.99±0.01**|**1.00±0.00**|
> |EDGE|-|**1.00±0.00**|**0.98±0.01**|**1.00±0.00**|–|
> |GRAN|128|**0.98±0.08**|0.92±0.08|**1.00±0.00**|–|
> |GRAN|256|0.75±0.00|**1.04±0.00**|0.99±0.00|–|
> |GRAN|512|0.83±0.06|**1.04±0.00**|0.98±0.02|–|
> |GraphMaker|-|**0.95±0.02**|**1.00±0.00**|**0.99±0.00**|**0.83±0.01**|
>
> In summary,  SyNGLER can generate graphs that yields a ML model with performance comparable to those trained on the original graphs. This validates the practical utility of the generated graphs in terms of training ML models.
>
> We have updated the experimental results and discussions in Section 4 and Appendix F of the revised manuscript, and kindly refer you there for details and more updated results.
>
> **References**
>
> [1] Zhang, Xuefei, Gongjun Xu, and Ji Zhu. "Joint latent space models for network data with high-dimensional node variables." Biometrika 109.3 (2022): 707-720.
>
> [2] Li, Mufei, et al. "Graphmaker: Can diffusion models generate large attributed graphs?." arXiv preprint arXiv:2310.13833 (2023).
>
> [3] Jo, Jaehyeong, Seul Lee, and Sung Ju Hwang. "Score-based generative modeling of graphs via the system of stochastic differential equations." International conference on machine learning. PMLR, 2022.

---

### Official Review · Reviewer_VEKk · 2025-11-02

**Soundness:** 3
**Presentation:** 3
**Contribution:** 3
**Rating:** 8
**Confidence:** 3

**Summary:**

SyNGLER is an efficient framework that generates synthetic networks via latent embedding reconstruction. It learns low-dimensional node embeddings and trains a lightweight generator in the latent space. This approach reduces computational cost and preserves key network structural properties, such as sparsity and degree heterogeneity.

**Strengths:**

The SyNGLER framework efficiently generates synthetic networks using latent embedding reconstruction. It significantly reduces computational cost and preserves key structural properties like sparsity and degree heterogeneity. It also provides theoretical consistency guarantees.

**Weaknesses:**

The SyNGLER framework needs extensions for conditional generation and support for complex network types like directed, dynamic, or multilayer networks. The resampling variant (SyNG-R) risks exposing sensitive information and creating unrealistic duplicate node embeddings. Future development must include rigorous privacy-preserving mechanisms.

**Questions:**

Why was the XGBoost-based score approximation chosen for SyNG-D's generator?

---

> ### Author Response · Authors · 2025-11-24
> **Author Response to Reviewer VEKk**
>
> Thank you for taking the time and effort to review our paper, and thank you for your appreciation and recognition of the efficiency and effectiveness of our method. Below, we respond to your questions one by one.
>
> **W:** "The SyNGLER framework needs extensions for conditional generation and support for complex network types like directed, dynamic, or multilayer networks. The resampling variant (SyNG-R) risks exposing sensitive information and creating unrealistic duplicate node embeddings. Future development must include rigorous privacy-preserving mechanisms."
>
> **Response:** Thank you for this forward-looking comment. Our current work focuses on unconditional generation for large, static, undirected networks, and we agree that extending SyNGLER to conditional generation and more complex network types is an important direction for future work.
>
> From a modeling perspective, the latent space formulation we use is flexible enough to accommodate these extensions. Conditional generation can be achieved by augmenting the latent embeddings with conditioning variables (e.g., node features or global covariates) and training a conditional version of the generative model in latent space. Directed networks can be handled by introducing separate in-/out-embeddings and degree parameters; multilayer networks by adding layer-specific latent effects on top of global embeddings; and dynamic networks by letting the latent embeddings evolve over time (e.g., via a temporal prior or dynamical model). These extensions are natural, but their properties and implementation require careful study, which go beyond the scope of this paper and we leave for future work.
>
> Regarding SyNG-R and privacy, we view the resampling-based variant as a simple option for scenarios where privacy is not the primary constraint (e.g., internal benchmarking or simulation studies). In our implementation, another main recommended configurations is SyNG-D, which samples new embeddings from a learned continuous latent distribution rather than copying existing nodes. This avoids directly reusing existing node embeddings that may carry sensitive latent characteristics. We fully agree, however, that rigorous privacy-preserving mechanisms are crucial for privacy-sensitive applications and lie beyond the scope of the current work. We view this as an important future avenue and plan to investigate formal privacy guarantees for SyNGLER in subsequent work.
>
> Meanwhile, in this revision, we have extended SyNGLER for static, undirected attributed graph, to make the current study on the static and undirected case more complete. Below, we introduce the extension named SyNGLER-Attr and present the main results on this aspect.
>
> We first clarify the setting of generating attributed graphs. In this task, we are given a single large attributed graph with both an adjacency matrix $A \in \mathbb{R}^{n \times n}$ and an attribute matrix $Y \in \mathbb{R}^{n \times p}$, where $n$ is the number of nodes and $p$ is the number of attributes. Our goal is to generate a new graph $\tilde{A}$ and a new attribute matrix $\tilde{Y}$ that preserve the structural and attribute properties of the original graph. In other words, we want the joint distribution of $(\tilde{A}, \tilde{Y})$ to be close to the joint distribution of $(A, Y)$.
>
> SyNGLER-Attr achieves this by jointly embedding the network and attributes into a latent space. Consider an attributed network with binary edges.
> Suppose that $A \in \{0,1\}^{n \times n}$ is the adjacency matrix of the network and $Y \in \mathbb{R}^{n \times p}$ is the attribute matrix.
> We embed the network into an embedding matrix $Z = (Z_1, Z_2) \in \mathbb{R}^{n \times (d_1 + d_2)}$, where $Z_1$ captures network latent factors and $Z_2$ captures attribute-specific latent factors.
> Concretely, we model the network and the attributes as follows:
> $$
> A_{ij} \stackrel{\text{ind.}}{\sim} \text{Bernoulli}(P_{ij}) \quad \text{with} \quad P = g(\alpha 1_n^\top + 1_n \alpha^\top + Z_1 Z_1^\top), \tag{1}
> $$
> $$
> Y = 1_n \mu^\top + Z_1 \Lambda_1^\top + Z_2 \Lambda_2^\top + \mathcal{E},
> \qquad \epsilon_i \stackrel{\text{i.i.d.}}{\sim} \mathcal{N}(0, \Psi), \tag{2}
> $$
> where $\Lambda_1$ and $\Lambda_2$ are the loadings for $Z_1$ and $Z_2$, respectively, and $\Psi$ is a diagonal noise covariance matrix.

---

> > ### Author Response · Authors · 2025-11-24
> >
> > Given this model, we first estimate $Z_1$ from the adjacency matrix $A$ using the latent space model described in the paper. We then estimate $\Lambda_1$ by regressing the centered attributes on $Z_1$, and compute the residuals to estimate $Z_2$ and $\Lambda_2$ via factor analysis. As a result, we obtain the embedding matrix $Z=(Z_1,Z_2)$ and the loadings $\Lambda$ for both network and attribute factors, which can be used for generation. For generation, we learn the distribution of the rows of $Z$ and sample new embeddings from this learned distribution. Substituting the generated embeddings into the above models with the estimated loadings, we then generate a new adjacency matrix and a new attribute matrix.
> >
> > The algorithm is summarized in the following steps:
> >
> > **Algorithm 1: SyNGLER-Attr: Synthetic Network Generation via Latent Embedding Reconstruction for Attributed Network**
> >
> > **Input:**  Adjacency matrix $A \in$ {0,1}$^{n\times n}$, Attribute matrix $Y \in \mathbb{R}^{n\times p}$.
> > 1. Maximum likelihood estimation using the latent space likelihood model $(A_{ij} \mid z_{1i},z_{1j},\alpha_i,\alpha_j,\rho)
> > \sim \text{Bernoulli}\big(
> > g(z_{1i}^\top z_{1j}+\alpha_i+\alpha_j+\rho)
> > \big)$ to obtain $(\hat Z_1,\hat\alpha,\hat\rho)\in\mathbb{R}^{n\times d_1}\times\mathbb{R}^n\times\mathbb{R}$.
> > 2. Regress $Y$ on $Z_1$ under the model $Y =  \mathbf{1}_n\mu^T+Z_1 \Lambda_1^\top+ R$, where R denotes the residual matrix.
> >
> > 3. Conduct an eigenvalue ratio test on the residual matrix $R$ to determine the dimension of the additional latent variable $Z_2$. Fit $R = Z_2\Lambda_2^\top + E, E_{i\cdot}\sim N(0,\Phi)$ where $\Phi$ is diagonal to obtain$(\hat Z_2,\hat\Lambda_2,\hat\Phi).$
> >
> > 4. Train a sampler using $\hat{Z} = (\hat{Z_1}, \hat{Z_2})$ and $\hat{\alpha}$, i.e.,  $\text{Sampler} = \mathrm{GenModel}\big(\{(\hat z_i,\hat\alpha_i)\}_{i=1}^n\big).$
> >
> >
> > 5. Sample new latent embeddings:$(\tilde z_{1i}^{\top}, \tilde{z}_{2i}^{\top},\tilde\alpha_i)^{\top} \sim \text{Sampler}.$
> >
> > 6. With sampled latent variables $\tilde z_i =\begin{pmatrix}
> > \tilde z_{1i}^{\top} \\
> > \tilde z_{2i}^{\top}
> > \end{pmatrix}^{\top} \in \mathbb{R}^{d_1+d_2},\tilde{\alpha}$, generate $A$ via $\tilde A_{ij} = \tilde A_{ji}\sim p(\cdot\,\big|\,\tilde z_{1i}^\top\tilde z_{1j} + \tilde\alpha_i + \tilde\alpha_j + \hat\rho)$, and generate attributes via $\tilde Y = \mathbf{1}_n\hat\mu^\top + \tilde Z_1\hat\Lambda_1^\top + \tilde Z_2\hat\Lambda_2^\top + N(0,\hat{\Phi}).$
> >
> > **Output:** Generated network $\tilde A$ and generated attributes $\tilde Y$.
> >
> > Below, we present experiments on the Cora dataset, an attributed network dataset, in Table 1 and Table 2. We compare SyNGLER-Attr with GraphMaker [1], which also produces synthetic attributed networks.  In both tables, SyNGLER-Attr captures the network structure and the attribute distribution, verifying the effectiveness of our method for generating attributed graphs.
> >
> > In particular, SyNGLER-Attr is able to generate graphs with structural statistics that are closer to those of the original graph and with comparable numerical statistics, as compared to GraphMaker [1], as shown in Table 1. We also compare the generated attributes with the original attributes in terms of KS distance and MMD, and find that SyNGLER-Attr achieves better performance than GraphMaker, as shown in Table 2.
> >
> > In addition, we include an ML-utility evaluation with a link prediction task, where we train two GAE models on the generated graph and the original graph and compare their performance on the test split of the original graph. For more details, we kindly refer you to the updated Section 4 of our manuscript.
> >
> > **Table 1:** Generation performance of SyNG-Attr and GraphMaker on the Cora dataset.
> >
> > |Category|Metric|SyNG-Attr|GraphMaker|
> > |---|---|---:|---:|
> > |**Structural Metrics**|Triangle Density (RMSE)|$3.60e^{-5}$|$\mathbf{1.00e^{-6}}$|
> > | |Clustering Coefficient (RMSE)|$\mathbf{5.38e^{-2}}$|$7.72e^{-2}$|
> > | |Eigenvalues (MMD)|$\mathbf{6.80e^{-1}}$|$1.03$|
> > | |Degree Centrality (MMD)|$2.61e^{-1}$|$\mathbf{2.36e^{-2}}$|
> > |**ML Utility (Link Prediction Task)**|Mean Ratio|$0.98\pm0.01$|$\mathbf{1.00\pm0.00}$|
> >
> > **Table 2:** KS and MMD distance for the generated attributes of SyNG-Attr and GraphMaker on the Cora dataset.
> >
> > |Method|KS|MMD|
> > |---|---|---|
> > |SyNG-Attr|**0.1398**|**0.1676**|
> > |GraphMaker|0.1400|0.1678|
> >
> > **Reference**
> >
> > [1] Li, Mufei, et al. "Graphmaker: Can diffusion models generate large attributed graphs?." arXiv preprint arXiv:2310.13833 (2023).

---

> > > ### Author Response · Authors · 2025-11-24
> > >
> > > **Q:** Why was the XGBoost-based score approximation chosen for SyNG-D's generator?
> > >
> > > **Response:** Thank you for this question regarding our choice of XGBoost for the score network. The XGBoost-based score fitting method was proposed in ForestDiffusion [1] to enable efficient score estimation for tabular data, which fits the structure of latent node embeddings. Empirically, we found that XGBoost-based score estimation is much more efficient than MLP-based score estimation and allows faster CPU-based training. Regarding generation performance, we compared MLP-based and XGBoost-based score estimation on real data and found that the XGBoost-based approach generally yields better performance, as shown in Tables 1–12 below. We will include a more detailed comparison between MLP-based and XGBoost-based score estimation in the revised version.
> > >
> > > **Table 1.** Generation performance (degree centralities) of SyNG-D on the YouTube dataset. Values for W1 distance, KS distance, and Energy distance are scaled by $10^{-2}$, and MMD values are scaled by $10^{-1}$. The best performance in each method is highlighted in bold font.
> > > | Method | cfg | W1 dist | KS dist | Energy dist | MMD |
> > > |:-------|:---:|:--------|:--------|:------------|:----|
> > > | SyNG-D | 2 | **0.15 ± 0.07** | **3.76 ± 1.43** | **0.01 ± 0.01** | **0.21 ± 0.18** |
> > > |        | 3 | 0.23 ± 0.09 | 5.61 ± 1.67 | **0.01 ± 0.01** | 0.45 ± 0.20 |
> > > |        | 4 | 0.32 ± 0.10 | 7.42 ± 1.91 | 0.03 ± 0.02 | 0.67 ± 0.22 |
> > > |        | 5 | 0.49 ± 0.09 | 11.79 ± 1.78 | 0.07 ± 0.02 | 1.15 ± 0.19 |
> > > |        | 6 | 0.58 ± 0.09 | 13.92 ± 1.87 | 0.10 ± 0.03 | 1.38 ± 0.20 |
> > > | SyNG-D (MLP) | 2 | 0.19 ± 0.05 | 6.05 ± 1.42 | **0.01 ± 0.00** | 0.42 ± 0.15 |
> > > |        | 3 | **0.18 ± 0.05** | 5.91 ± 1.29 | **0.01 ± 0.00** | 0.41 ± 0.14 |
> > > |        | 4 | **0.18 ± 0.05** | **5.88 ± 1.51** | **0.01 ± 0.01** | **0.39 ± 0.17** |
> > > |        | 5 | 0.22 ± 0.05 | 8.00 ± 1.57 | **0.01 ± 0.01** | 0.63 ± 0.15 |
> > > |        | 6 | 0.29 ± 0.08 | 9.82 ± 1.64 | 0.03 ± 0.01 | 0.81 ± 0.18 |
> > >
> > > **Table 2.** Generation performance (eigenvalues) of SyNG-D on the YouTube dataset. Values for KS distance, Energy distance, and MMD are scaled by $10^{-1}$. The best performance in each method is highlighted in bold font.
> > > | Method | cfg | W1 dist | KS dist | Energy dist | MMD |
> > > |:-------|:---:|:--------|:--------|:------------|:----|
> > > | SyNG-D | 2 | 0.31 ± 0.06 | 0.30 ± 0.07 | 0.11 ± 0.05 | 0.26 ± 0.09 |
> > > |        | 3 | 0.23 ± 0.04 | 0.20 ± 0.06 | 0.05 ± 0.03 | 0.08 ± 0.10 |
> > > |        | 4 | **0.19 ± 0.04** | **0.14 ± 0.04** | **0.03 ± 0.02** | **0.01 ± 0.03** |
> > > |        | 5 | 0.30 ± 0.08 | 0.20 ± 0.06 | 0.08 ± 0.04 | 0.02 ± 0.05 |
> > > |        | 6 | 0.40 ± 0.08 | 0.28 ± 0.07 | 0.14 ± 0.07 | 0.11 ± 0.12 |
> > > | SyNG-D (MLP) | 2 | 0.30 ± 0.06 | 0.28 ± 0.06 | 0.10 ± 0.05 | 0.24 ± 0.09 |
> > > |        | 3 | 0.28 ± 0.06 | 0.27 ± 0.07 | 0.09 ± 0.05 | 0.20 ± 0.11 |
> > > |        | 4 | 0.19 ± 0.05 | 0.18 ± 0.06 | 0.04 ± 0.03 | 0.04 ± 0.07 |
> > > |        | 5 | **0.17 ± 0.04** | 0.14 ± 0.04 | **0.03 ± 0.02** | 0.01 ± 0.04 |
> > > |        | 6 | 0.19 ± 0.06 | **0.14 ± 0.04** | 0.03 ± 0.03 | **0.00 ± 0.02** |
> > >
> > > **Table 3.** Generation performance (clustering coefficient and triangle density) of SyNG-D on the YouTube dataset. Values for Clust RMSE, Clust MAE, and Clust Bias are scaled by $10^{-2}$, and Tri RMSE, Tri MAE, and Tri Bias are scaled by $10^{-4}$. The best performance in each method is highlighted in bold font.
> > > | Method | cfg | Clust RMSE | Clust MAE | Clust Bias | Tri RMSE | Tri MAE | Tri Bias |
> > > |:-------|:---:|:-----------|:----------|:-----------|:---------|:--------|:---------|
> > > | SyNG-D | 2 | 2.28 | 2.15 | -2.15 | 0.97 | 0.92 | 0.92 |
> > > |        | 3 | 1.69 | 1.46 | -1.41 | 0.89 | 0.84 | 0.84 |
> > > |        | 4 | **1.34** | **1.08** | **-0.95** | 0.83 | 0.78 | 0.78 |
> > > |        | 5 | 1.50 | 1.22 | -1.09 | 0.58 | 0.52 | 0.52 |
> > > |        | 6 | 1.42 | 1.19 | -1.01 | **0.41** | **0.35** | **0.35** |
> > > | SyNG-D (MLP) | 2 | **1.12** | **0.88** | **-0.46** | 1.35 | 1.30 | 1.30 |
> > > |        | 3 | 1.87 | 1.63 | 1.54 | 1.47 | 1.43 | 1.43 |
> > > |        | 4 | 1.58 | 1.34 | 1.15 | 1.62 | 1.58 | 1.58 |
> > > |        | 5 | 2.98 | 2.83 | 2.80 | 1.40 | 1.36 | 1.36 |
> > > |        | 6 | 3.13 | 2.93 | 2.88 | **1.18** | **1.14** | **1.14** |

---

> > > > ### Author Response · Authors · 2025-11-24
> > > >
> > > > **Table 4.** Generation performance (degree centralities) of SyNG-D on the Yelp dataset. Values for W1 dist, Energy dist, and MMD are scaled by $10^{-2}$, and KS dist is scaled by $10^{-1}$. The best performance in each method is highlighted in bold font.
> > > > | Method | cfg | W1 dist | KS dist | Energy dist | MMD |
> > > > |:-------|:---:|:--------|:--------|:------------|:----|
> > > > | SyNG-D | 2 | **0.15 ± 0.05** | **0.25 ± 0.07** | **0.54 ± 0.19** | **0.13 ± 0.09** |
> > > > |        | 3 | 0.21 ± 0.10 | 0.35 ± 0.13 | 0.77 ± 0.34 | 0.24 ± 0.14 |
> > > > |        | 4 | 0.24 ± 0.10 | 0.45 ± 0.11 | 0.98 ± 0.33 | 0.35 ± 0.11 |
> > > > |        | 5 | 0.36 ± 0.11 | 0.60 ± 0.12 | 1.42 ± 0.37 | 0.50 ± 0.12 |
> > > > |        | 6 | 0.44 ± 0.13 | 0.77 ± 0.14 | 1.81 ± 0.44 | 0.67 ± 0.14 |
> > > > | SyNG-D (MLP) | 2 | 0.32 ± 0.12 | **0.47 ± 0.14** | 1.18 ± 0.43 | **0.33 ± 0.14** |
> > > > |        | 3 | 0.39 ± 0.12 | 0.54 ± 0.13 | 1.38 ± 0.40 | 0.43 ± 0.12 |
> > > > |        | 4 | 0.69 ± 0.15 | 0.86 ± 0.13 | 2.49 ± 0.46 | 0.78 ± 0.12 |
> > > > |        | 5 | **0.23 ± 0.08** | 0.52 ± 0.10 | **1.01 ± 0.26** | 0.42 ± 0.08 |
> > > > |        | 6 | 0.30 ± 0.09 | 0.48 ± 0.11 | 1.19 ± 0.33 | 0.45 ± 0.10 |
> > > >
> > > > **Table 5.** Generation performance (eigenvalues) of SyNG-D on the Yelp dataset. Values for KS dist and MMD are scaled by $10^{-2}$, Energy dist is scaled by $10^{-1}$, and W1 dist is unscaled. The best performance in each method is highlighted in bold font.
> > > > | Method | cfg | W1 dist | KS dist | Energy dist | MMD |
> > > > |:-------|:---:|:--------|:--------|:------------|:----|
> > > > | SyNG-D | 2 | 1.29 ± 0.09 | 0.53 ± 0.04 | 2.93 ± 0.24 | 0.61 ± 0.05 |
> > > > |        | 3 | 1.09 ± 0.10 | 0.44 ± 0.05 | 2.43 ± 0.27 | 0.51 ± 0.05 |
> > > > |        | 4 | 0.92 ± 0.08 | 0.38 ± 0.04 | 2.03 ± 0.23 | 0.43 ± 0.05 |
> > > > |        | 5 | 0.77 ± 0.08 | 0.30 ± 0.04 | 1.62 ± 0.22 | 0.34 ± 0.05 |
> > > > |        | 6 | **0.64 ± 0.09** | **0.23 ± 0.05** | **1.28 ± 0.25** | **0.25 ± 0.06** |
> > > > | SyNG-D (MLP) | 2 | 2.07 ± 0.10 | 0.84 ± 0.04 | 4.89 ± 0.25 | 0.96 ± 0.05 |
> > > > |        | 3 | 1.88 ± 0.10 | 0.75 ± 0.04 | 4.43 ± 0.25 | 0.87 ± 0.04 |
> > > > |        | 4 | 2.00 ± 0.11 | 0.82 ± 0.04 | 4.79 ± 0.26 | 0.94 ± 0.05 |
> > > > |        | 5 | **1.21 ± 0.10** | **0.51 ± 0.04** | **2.83 ± 0.25** | **0.57 ± 0.05** |
> > > > |        | 6 | 1.55 ± 0.09 | 0.66 ± 0.04 | 3.73 ± 0.23 | 0.75 ± 0.04 |
> > > >
> > > > **Table 6.** Generation performance (clustering coefficient and triangle density) of SyNG-D on the Yelp dataset. Values for Clust RMSE, Clust MAE, and Clust Bias are scaled by $10^{-2}$, and Tri RMSE, Tri MAE, and Tri Bias are scaled by $10^{-4}$. The best performance in each method is highlighted in bold font.
> > > > | Method | param | Clust RMSE | Clust MAE | Clust Bias | Tri RMSE | Tri MAE | Tri Bias |
> > > > |:-------|:-----:|:-----------|:----------|:-----------|:---------|:--------|:---------|
> > > > | SyNG-D | 2 | 2.56 | 2.52 | -2.52 | **1.24** | **1.08** | **-1.05** |
> > > > |        | 3 | 2.65 | 2.61 | -2.61 | 1.71 | 1.57 | -1.56 |
> > > > |        | 4 | 2.01 | 1.96 | -1.96 | 1.59 | 1.47 | -1.46 |
> > > > |        | 5 | 1.83 | 1.77 | -1.77 | 1.82 | 1.72 | -1.71 |
> > > > |        | 6 | **1.77** | **1.70** | **-1.70** | 2.00 | 1.89 | -1.88 |
> > > > | SyNG-D (MLP) | 2 | 2.35 | 2.31 | -2.31 | **0.74** | **0.61** | **-0.22** |
> > > > |        | 3 | 1.05 | 0.96 | -0.96 | 1.40 | 1.18 | 1.15 |
> > > > |        | 4 | 0.75 | 0.63 | 0.58 | 3.33 | 3.18 | 3.18 |
> > > > |        | 5 | 0.81 | 0.71 | -0.68 | 0.98 | 0.84 | -0.72 |
> > > > |        | 6 | **0.65** | **0.53** | **0.47** | 1.44 | 1.26 | 1.23 |
> > > >
> > > > **Table 7.** Generation performance (degree centralities) of SyNG-D on the PolBlogs dataset. All metrics (W1 dist, KS dist, Energy dist, and MMD) are scaled by $10^{-2}$. The best performance in each method is highlighted in bold font.
> > > > | Method | cfg | W1 dist | KS dist | Energy dist | MMD |
> > > > |:-------|:---:|:--------|:--------|:------------|:----|
> > > > | SyNG-D | 2 | **0.18 ± 0.08** | **4.53 ± 0.97** | **0.01 ± 0.01** | **0.97 ± 1.44** |
> > > > |        | 3 | 0.22 ± 0.11 | 5.02 ± 1.34 | **0.01 ± 0.01** | 1.95 ± 2.15 |
> > > > |        | 4 | 0.28 ± 0.13 | 5.74 ± 1.64 | 0.02 ± 0.02 | 3.18 ± 2.50 |
> > > > |        | 5 | 0.44 ± 0.14 | 8.29 ± 2.04 | 0.04 ± 0.02 | 6.49 ± 2.44 |
> > > > |        | 6 | 0.52 ± 0.15 | 9.98 ± 2.11 | 0.06 ± 0.03 | 8.31 ± 2.50 |
> > > > | SyNG-D (MLP) | 2 | 0.20 ± 0.07 | 9.59 ± 1.21 | **0.01 ± 0.01** | 1.47 ± 1.71 |
> > > > |        | 3 | 0.24 ± 0.10 | 9.81 ± 1.09 | **0.01 ± 0.01** | 3.05 ± 2.25 |
> > > > |        | 4 | 0.33 ± 0.15 | 10.06 ± 1.49 | 0.02 ± 0.02 | 3.20 ± 2.26 |
> > > > |        | 5 | 0.23 ± 0.07 | **8.99 ± 0.93** | **0.01 ± 0.01** | 2.79 ± 1.77 |
> > > > |        | 6 | **0.19 ± 0.08** | 9.16 ± 1.35 | **0.01 ± 0.01** | **1.41 ± 1.73** |

---

> > > > > ### Author Response · Authors · 2025-11-24
> > > > >
> > > > > **Table 8.** Generation performance (eigenvalues) of SyNG-D on the PolBlogs dataset. Values for KS dist, Energy dist, and MMD are scaled by $10^{-2}$. All values are rounded to two decimal places. The best performance in each method is highlighted in bold font.
> > > > > | Method | cfg | W1 dist | KS dist | Energy dist | MMD |
> > > > > |:-------|:---:|:--------|:--------|:------------|:----|
> > > > > | SyNG-D | 2 | 0.21 ± 0.06 | 0.48 ± 0.09 | 0.92 ± 0.51 | 0.26 ± 0.14 |
> > > > > |        | 3 | 0.20 ± 0.04 | 0.50 ± 0.10 | 0.90 ± 0.38 | 0.32 ± 0.12 |
> > > > > |        | 4 | **0.20 ± 0.05** | 0.44 ± 0.10 | **0.74 ± 0.32** | 0.25 ± 0.12 |
> > > > > |        | 5 | 0.25 ± 0.07 | **0.32 ± 0.08** | 0.80 ± 0.46 | 0.15 ± 0.12 |
> > > > > |        | 6 | 0.30 ± 0.09 | 0.33 ± 0.09 | 1.19 ± 0.72 | **0.14 ± 0.14** |
> > > > > | SyNG-D (MLP) | 2 | 0.46 ± 0.10 | 0.70 ± 0.09 | 3.36 ± 1.34 | 0.55 ± 0.11 |
> > > > > |        | 3 | **0.37 ± 0.09** | 0.70 ± 0.08 | **2.51 ± 1.04** | **0.53 ± 0.09** |
> > > > > |        | 4 | 0.65 ± 0.11 | 0.89 ± 0.09 | 6.88 ± 1.98 | 0.82 ± 0.11 |
> > > > > |        | 5 | 0.38 ± 0.10 | 0.70 ± 0.09 | 2.81 ± 1.18 | 0.56 ± 0.11 |
> > > > > |        | 6 | 0.39 ± 0.10 | **0.67 ± 0.09** | 2.80 ± 1.18 | 0.54 ± 0.11 |
> > > > >
> > > > > **Table 9.** Generation performance (clustering coefficient and triangle density) of SyNG-D on the PolBlogs dataset. Values for Clust RMSE, Clust MAE, and Clust Bias are scaled by $10^{-2}$, and Tri RMSE, Tri MAE, and Tri Bias are scaled by $10^{-4}$. The best performance in each method is highlighted in bold font.
> > > > > | Method | param | Clust RMSE | Clust MAE | Clust Bias | Tri RMSE | Tri MAE | Tri Bias |
> > > > > |:-------|:-----:|:-----------|:----------|:-----------|:---------|:--------|:---------|
> > > > > | SyNG-D | 2 | 1.90 | 1.51 | 1.15 | 0.71 | **0.55** | **0.13** |
> > > > > |        | 3 | **1.56** | **1.23** | 0.54 | **0.68** | 0.56 | -0.33 |
> > > > > |        | 4 | 1.85 | 1.45 | 0.90 | 0.78 | 0.66 | -0.43 |
> > > > > |        | 5 | 1.66 | 1.33 | **0.49** | 1.09 | 1.00 | -0.98 |
> > > > > |        | 6 | 1.91 | 1.49 | 0.93 | 1.19 | 1.09 | -1.07 |
> > > > > | SyNG-D (MLP) | 2 | 3.58 | 3.26 | 3.23 | 0.91 | 0.69 | 0.56 |
> > > > > |        | 3 | **2.23** | **1.88** | **1.73** | **0.59** | **0.48** | **-0.16** |
> > > > > |        | 4 | 5.69 | 5.45 | 5.45 | 1.80 | 1.61 | 1.60 |
> > > > > |        | 5 | 5.18 | 4.77 | 4.74 | 1.15 | 0.90 | 0.77 |
> > > > > |        | 6 | 3.15 | 2.80 | 2.72 | 0.84 | 0.65 | 0.50 |
> > > > >
> > > > > **Table 10.** Generation performance (degree centralities) of SyNG-D on the DBLP dataset. Values for W1 dist and Energy dist are scaled by $10^{-2}$, and KS dist and MMD are scaled by $10^{-1}$. The best performance in each method is highlighted in bold font.
> > > > > | Method | cfg | W1 dist | KS dist | Energy dist | MMD |
> > > > > |:-------|:---:|:--------|:--------|:------------|:----|
> > > > > | SyNG-D | 2 | 0.19 ± 0.08 | **0.75 ± 0.17** | **0.02 ± 0.01** | **0.93 ± 0.23** |
> > > > > |        | 3 | **0.18 ± 0.06** | 0.91 ± 0.16 | **0.02 ± 0.01** | 1.03 ± 0.19 |
> > > > > |        | 4 | 0.22 ± 0.07 | 1.22 ± 0.14 | 0.03 ± 0.01 | 1.29 ± 0.17 |
> > > > > |        | 5 | 0.33 ± 0.09 | 1.74 ± 0.18 | 0.05 ± 0.02 | 1.78 ± 0.20 |
> > > > > |        | 6 | 0.44 ± 0.10 | 2.19 ± 0.18 | 0.09 ± 0.02 | 2.22 ± 0.20 |
> > > > > | SyNG-D (MLP) | 2 | **0.31 ± 0.07** | 1.81 ± 0.17 | **0.04 ± 0.01** | **1.50 ± 0.16** |
> > > > > |        | 3 | 0.42 ± 0.06 | 1.33 ± 0.13 | 0.07 ± 0.02 | 1.67 ± 0.18 |
> > > > > |        | 4 | 0.47 ± 0.15 | **1.19 ± 0.19** | 0.06 ± 0.03 | **1.40 ± 0.25** |
> > > > > |        | 5 | 0.47 ± 0.10 | 1.36 ± 0.14 | 0.07 ± 0.02 | 1.72 ± 0.16 |
> > > > > |        | 6 | 0.35 ± 0.06 | 1.53 ± 0.17 | 0.06 ± 0.01 | 1.73 ± 0.19 |
> > > > >
> > > > > **Table 11.** Generation performance (eigenvalues) of SyNG-D on the DBLP dataset. Values for W1 dist, KS dist, and MMD are scaled by $10^{-1}$, and Energy dist is scaled by $10^{-2}$. The best performance in each method is highlighted in bold font.
> > > > > | Method | cfg | W1 dist | KS dist | Energy dist | MMD |
> > > > > |:-------|:---:|:--------|:--------|:------------|:----|
> > > > > | SyNG-D | 2 | 3.02 ± 0.32 | **0.81 ± 0.06** | 2.35 ± 0.43 | 0.88 ± 0.06 |
> > > > > |        | 3 | 2.13 ± 0.28 | 0.91 ± 0.07 | **1.57 ± 0.21** | **0.77 ± 0.05** |
> > > > > |        | 4 | **1.73 ± 0.17** | 1.06 ± 0.07 | 1.65 ± 0.20 | **0.77 ± 0.05** |
> > > > > |        | 5 | 2.56 ± 0.43 | 1.32 ± 0.08 | 2.99 ± 0.57 | 0.96 ± 0.08 |
> > > > > |        | 6 | 3.80 ± 0.46 | 1.63 ± 0.09 | 5.62 ± 0.89 | 1.28 ± 0.08 |
> > > > > | SyNG-D (MLP) | 2 | **2.85 ± 0.34** | 1.10 ± 0.07 | **2.37 ± 0.31** | **0.93 ± 0.05** |
> > > > > |        | 3 | 6.09 ± 0.47 | 1.10 ± 0.08 | 8.94 ± 1.27 | 1.35 ± 0.07 |
> > > > > |        | 4 | 4.46 ± 0.41 | **0.83 ± 0.05** | 5.05 ± 0.87 | 1.12 ± 0.07 |
> > > > > |        | 5 | 5.75 ± 0.46 | 0.97 ± 0.07 | 7.42 ± 1.06 | 1.32 ± 0.06 |
> > > > > |        | 6 | 5.92 ± 0.45 | 1.03 ± 0.08 | 8.06 ± 1.13 | 1.31 ± 0.06 |

---

> > > > > > ### Author Response · Authors · 2025-11-24
> > > > > >
> > > > > > **Table 14.** Generation performance (clustering coefficient and triangle density) of SyNG-D on the DBLP dataset. Values for Clust RMSE, Clust MAE, and Clust Bias are scaled by $10^{-1}$, and Tri RMSE, Tri MAE, and Tri Bias are scaled by $10^{-4}$. The best performance in each method is highlighted in bold font.
> > > > > > | Method | param | Clust RMSE | Clust MAE | Clust Bias | Tri RMSE | Tri MAE | Tri Bias |
> > > > > > |:-------|:-----:|:-----------|:----------|:-----------|:---------|:--------|:---------|
> > > > > > | SyNG-D | 2 | 0.62 | 0.59 | -0.59 | **1.58** | **1.26** | **0.46** |
> > > > > > |        | 3 | **0.53** | **0.49** | **-0.49** | 1.61 | 1.37 | -0.87 |
> > > > > > |        | 4 | 0.55 | 0.52 | -0.52 | 1.81 | 1.57 | -1.39 |
> > > > > > |        | 5 | **0.53** | **0.49** | **-0.49** | 2.34 | 2.13 | -2.07 |
> > > > > > |        | 6 | 0.57 | 0.54 | -0.54 | 3.01 | 2.86 | -2.86 |
> > > > > > | SyNG-D (MLP) | 2 | 1.40 | 1.36 | -1.36 | 2.91 | 2.67 | -2.62 |
> > > > > > |        | 3 | 2.42 | 2.39 | -2.39 | 3.74 | 3.64 | -3.64 |
> > > > > > |        | 4 | **1.01** | **0.99** | **-0.99** | 3.63 | 3.20 | 3.13 |
> > > > > > |        | 5 | 1.44 | 1.41 | -1.41 | 2.29 | 1.81 | 1.47 |
> > > > > > |        | 6 | 1.71 | 1.69 | -1.69 | **1.80** | **1.54** | **-1.37** |
> > > > > >
> > > > > > [1] Jolicoeur-Martineau, Alexia, Kilian Fatras, and Tal Kachman. “Generating and imputing tabular data via diffusion and flow-based gradient-boosted trees.” *International Conference on Artificial Intelligence and Statistics*. PMLR, 2024.

---

### Author Response · Authors · 2025-12-03
**Comment to Area Chair**

Dear Area Chair,

We are writing regarding the recent incident involving “leaked reviewer/AC identities.” We understand the seriousness of this issue and that it has created significant additional workload for you. Below, we briefly summarize the status of our submission for your convenience; further details are provided in our point-by-point rebuttal.

Our original scores were “**8, 6, 2, 2**”. Two reviewers provided positive comments acknowledging our novelty and contribution. The other two reviewers offered very specific and constructive comments, which we have carefully addressed point by point.

**Before the incident occurred, one of the reviewers who initially gave a score of 2 had already raised their score from 2 to 6**. They responded to our rebuttal, stating:

> **I am greatly surprised by the several changes and replies to all of the comments.**

**The remaining three reviewers did not have the opportunity to respond before the discussion phase closed.** In particular, **the other reviewer who initially gave a score of 2 had specific concerns** regarding extending our method to attributed graphs and evaluating it using downstream tasks. **We have fully addressed these concerns in our rebuttal**. For the reviewers with scores of 6 and 8, we have also addressed all of their concerns and questions in our rebuttal and have substantially revised the paper to reflect these improvements. We believe these changes significantly enhance both the quality and the contribution of our work.

We sincerely thank you for your time and effort in handling our paper, and we hope that our efforts during the rebuttal phase can be taken into consideration.

Best regards,

---

### Meta-Review · Area_Chair_zDHL · 2026-01-06

**Summary:**

The paper propose, SyNGLER, to efficiently generate graph samples that preserves characteristics including sparsity or degree heterogeneity. SyNGLER consist of two-stage including latent space modelling for low-dimensional embedding and a generation stage trained on the latent. KL divergence are discussed and consistency analysis are provided. Empirical simulation and experiments on real datasets are conducted to demonstrate SyNGLER performance on preserved structures and network statistics which shown to be more computationally efficient than existing methods.

**Reviewer Concerns:**

Generation of non-attribute graphs. The point has been addressed in an extension to SyNGLER-Attr to have joint input of both adjacency and attribute.

Including code for reproducibility. Code repo provided in the discussion.

Lack of comparison with existing graph generation. Additional experiments are included while as the reviewer pointed out, GRAN, EDGE comparison are added to the Table 3 presentation with ML utility as a measure of performances.

Paper clarity and replicability are generally low; which are not explicitly addressed.

**Reviewer Scores:**

There is a raise in score in reviewer qzcR as claimed in response.

---

### Decision · Program_Chairs · 2026-01-26

Reject